# Understanding challenges to the interpretation of disaggregated evaluations of algorithmic fairness

**Stephen R. Pfohl**
Google Research
spfohl@google.com

**Natalie Harris**
Google Research

**Chirag Nagpal**
Google Research

**David Madras**
Google DeepMind

**Vishwali Mhasawade**
New York University

**Olawale Salaudeen**
Massachusetts Institute of Technology

**Awa Dieng**
Google DeepMind

**Shannon Sequeira**
Google Research

**Santiago Arciniegas**
The Hospital for Sick Children

**Lillian Sung**
The Hospital for Sick Children

**Nnamdi Ezeanochie**
Google

**Heather Cole-Lewis**
Google

**Katherine Heller**
Google Research

**Sanmi Koyejo**
Stanford University

**Alexander D'Amour**
Google DeepMind

## Abstract

Disaggregated evaluation across subgroups is critical for assessing the fairness of machine learning models, but its uncritical use can mislead practitioners. We show that equal performance across subgroups is an unreliable measure of fairness when data are representative of the relevant populations but reflective of real-world disparities. Furthermore, when data are not representative due to selection bias, both disaggregated evaluation and alternative approaches based on conditional independence testing may be invalid without explicit assumptions regarding the bias mechanism. We use causal graphical models to characterize fairness properties and metric stability across subgroups under different data generating processes. Our framework suggests complementing disaggregated evaluations with explicit causal assumptions and analysis to control for confounding and distribution shift, including conditional independence testing and weighted performance estimation. These findings have broad implications for how practitioners design and interpret model assessments given the ubiquity of disaggregated evaluation.

## 1 Introduction

A significant body of work uses disaggregated evaluation of machine learning models across subgroups (*e.g.*, by race, ethnicity, or gender) to assess algorithmic fairness properties [1]. In this paradigm, differences in a performance metric (*e.g.*, accuracy, sensitivity, specificity, positive predictive value) or other statistical property (*e.g.* calibration or the distribution of predictions or covariates) across subgroups are taken as evidence of a fairness violation. In healthcare contexts, for example, this approach has been applied to a variety of settings to evaluate machine learning models across patient subgroups [2–4], including prognostic models of cardiovascular disease risk [5–7] as well as diagnostic classifiers for medical images [8, 9].

39th Conference on Neural Information Processing Systems (NeurIPS 2025).

We assume a setting where the policy and fairness goal is consistent with accurate prediction for all subgroups of interest based on historical data, which can be justified in some circumstances [6, 7, 10–12]. However, we emphasize that fairness is not an inherent property of a model, but rather related to the effect on outcomes that a policy leveraging the model has in a deployment context [13], and even accurate models can introduce harm and exacerbate disparities [14, 15].

In this work, we investigate ways that disaggregated evaluation of the performance of predictive models across subgroups can be misleading. A central issue is that models that maximize predictive performance for each subgroup do not generally attain equal performance across subgroups. This follows because the optimal value of a performance metric typically changes under distribution shift [16] and data distributions tend to differ across subgroups in contexts where disparities are present (*e.g.*, under differential exposure to social and structural determinants of health (SDOH) [17, 18]). We argue that disaggregated evaluation can thus mislead because differences in model performance across subgroups do not necessarily imply misestimation for any subgroup, nor an unfair or inequitable intervention or policy [6, 7, 12, 19]. As a consequence, disaggregated evaluation does not necessarily allow for identification of subgroups for which further data collection or model scaling would be beneficial. Furthermore, algorithmic strategies to develop models that attain equal performance (*e.g.*, constrained optimization or post-processing [2, 20, 21]) can directly introduce harm without inducing a contextually-meaningful notion of fairness [6, 7, 12, 19].

We emphasize that the above considerations apply in settings that are *well-specified*, that is, when the prediction task is well-formulated on the basis of a specific use case and target population, and a large and high-quality dataset representative of that target population is available for model development and evaluation [22–24]. If this is not the case, *e.g.*, if the observed data are misrepresentative of the intended target population, a model that predicts outcomes well in the observed data may not generalize to the intended target population, potentially directly introducing fairness-related harms when the structure of the misgeneralization is systematic across subgroups. For example, under various forms of bias affecting problem formulation, data collection, and measurement, accurate prediction in the observed data implies structured forms of misestimation in the target population that cannot be detected in the observed data without knowledge of the structure of the bias [22, 23, 25–29]. In this work, while we primarily operate in the ideal, well-specified setting detailed above, we do extend our analysis to the non-ideal setting to include structured forms of selection bias [25].

We argue that building understanding of the issues discussed above is critical to the design and interpretation of disaggregated evaluations. To that end, we make several contributions, which can be summarized as follows:

- We characterize the properties of models learned and evaluated under a variety of data generating processes. We use causal graphical models of distribution shift to encode structured forms of heterogeneity across subgroups and to describe explicit forms of distribution shift through selection bias. This characterizes the fairness properties of models under various assumptions on the data generating process. This approach builds on prior works that use causal directed acyclic graphs to study algorithmic fairness and distribution shift [29–36], as well as those that study incompatibilities between different notions of fairness [37–40].

- We present theoretical results that show that in simple, prototypical cases, average performance of the optimal predictor is not expected to be the same across subgroups, but that these differences can be directly anticipated based on the causal structure of the data generating process. In some cases, performance differences can be directly explained by differences in the marginal distribution of a confounder across subgroups, potentially motivating the use of evaluation procedures that control for such confounding.

- We investigate the use of weighted evaluation procedures as a means of constructing evaluations that control for confounding due to distributional differences across subgroups, building off of Cai et al. [16]. We show how such procedures can be interpreted as a class of configurable conditional independence tests and provide guidance for the interpretation of such techniques in concert with standard disaggregated evaluations.

- We conduct experiments with synthetic and real-world data to empirically verify the properties suggested by our theoretical analysis*.

---

*Code to reproduce the experiments is available at `https://github.com/google-research/google-research/tree/master/causal_evaluation`.

## 2 Preliminaries

We consider data with covariates $X \in \mathbb{R}^n$, a binary label $Y \in \{0, 1\}$, and a categorical subgroup indicator $A \in \mathcal{A}$. We reason about models $f$ that take as input $Z \in \{X, \{X, A\}\}$ to produce scores $R = f(Z)$ that can be compared to a threshold $\tau$ to yield binary predictions $\hat{Y} = \mathbb{1}[R \geq \tau]$.

Several of our findings relate to the properties of oracle models $f^*$ that can be considered to return $\mathbb{E}[Y \mid Z]$, such that $f^*(Z) = \mathbb{E}[Y \mid Z]$. Following Mhasawade et al. [29], we define $f^*(X)$ as the *population* Bayes-optimal model that returns the conditional expectation of $Y$ given covariates $X$, such that $R^* = f^*(X) = \mathbb{E}[Y \mid X]$ and the *subgroup* Bayes-optimal model as $R_A^* = f_A^*(X, A) = \mathbb{E}[Y \mid X, A]$. The subgroup Bayes-optimal model can also be represented as a set of subgroup-specific Bayes-optimal models ($\{f_a^*\}_{a \in \mathcal{A}}$ for $f_a^*(X) = \mathbb{E}[Y \mid X, A = a]$). For arbitrary, potentially non-optimal models, we refer to models that only have access to $X$ as *subgroup-agnostic* and those that have access to $A$ as *subgroup-aware*. We refer to fitting separate models for each subgroup as a type of subgroup-aware prediction called *stratified* prediction.

Because our scope is limited to modeling binary outcomes, it follows that $\mathbb{E}[Y \mid X] = P(Y = 1 \mid X)$. This implies that the Bayes-optimal predictor $f^*(Z)$ captures all of the information that $Z$ has about $Y$, and thus $Y \perp Z \mid f^*(Z)$ and $\mathbb{E}[Y \mid f^*(Z)] = \mathbb{E}[Y \mid f^*(Z), Z]$. Furthermore, Bayes-optimal predictors are calibrated, meaning that $c(r) = r$ for all $r \in [0, 1]$ for a calibration curve $c(r) = \mathbb{E}[Y \mid R = r]$. We note that calibration is necessary but not sufficient for Bayes-optimality: miscalibration implies lack of Bayes-optimality, but calibration does not imply Bayes-optimality.

We assume that the model of interest is fit and evaluated based on data drawn from a *source* distribution over $\{X, Y, A\}$ given by $P(X, Y, A)$ and evaluated on a *target* distribution indicative of the target population for which it is of interest to deploy the model. Unless stated otherwise, we assume that the source distribution matches the target distribution. When it is of interest to indicate a systematic difference between source and target distributions, we use the convention of *selection* [25], where we consider a model fit using data drawn from the selected, source population $P(X, Y, A \mid S = 1)$ and reason about the properties of those models evaluated on samples drawn from the full, target population without selection $P(X, Y, A)$.

### 2.1 Algorithmic Fairness and Robustness

It is common to reason about algorithmic fairness through disaggregated evaluation of model performance over subgroups of the population or otherwise testing for some form of independence or conditional independence involving subgroup membership. For disaggregated evaluation, we consider metrics $m : \mathbb{R} \times \mathcal{Y} \to \mathbb{R}$ that are decomposable (*i.e.*, can be computed at an instance-level and aggregated), where the induced fairness condition is given by $\mathbb{E}[m_j(Y, R) \mid A = a] = \mathbb{E}[m_j(Y, R)]$ for some fixed set of subgroups $\{a\}_{a \in \mathcal{A}}$ and metrics $\{m_j\}_{j=1}^M$.

It is also popular to operationalize fairness with statistical criteria corresponding to independence or conditional independence. For example, this includes demographic parity ($R \perp A$) [41, 42], separation ($R \perp A \mid Y$) [20], equalized odds ($\hat{Y} \perp A \mid Y$) [20], sufficiency ($Y \perp A \mid R$) [37], and predictive parity ($Y \perp A \mid \hat{Y} = 1$) [39]. Some of these criteria can be directly interpreted as conditions of equal performance across subgroups. For example, separation is a sufficient condition for equalized odds, which corresponds to a condition where both true positive error rates (also known as sensitivity or recall) and false positive error rates (also known as 1-specificity) are equal across subgroups. Predictive parity likewise corresponds to a case where the positive predictive value (also known as precision) of the model is equal across subgroups. In contrast, sufficiency is more naturally described as a condition of equal calibration curves, rather than as an equal performance condition.

Other works have framed fairness as a form of robustness over subgroups [43, 44], conceptualized in terms of worst-case performance across subgroups, with an optimization objective formulated to maximize worst-case subgroup performance. This perspective is aligned with that implicitly taken in disaggregated evaluation, in that some deviation in model performance across subgroups is taken as evidence of disparities in model quality across subgroups.

A number of prior works have documented theoretical and empirical incompatibilities and trade-offs between the notions of fairness outlined above [2, 37–40]. Particularly relevant to this work are the findings of Liu et al. [37], where it is shown that, in settings where the prevalence of the label

**Table 1: Causal graphs encoding assumptions regarding heterogeneity across subgroups**. $X$ indicates covariates, $Y$ a binary label, $A$ subgroup membership, and $S$ selection.

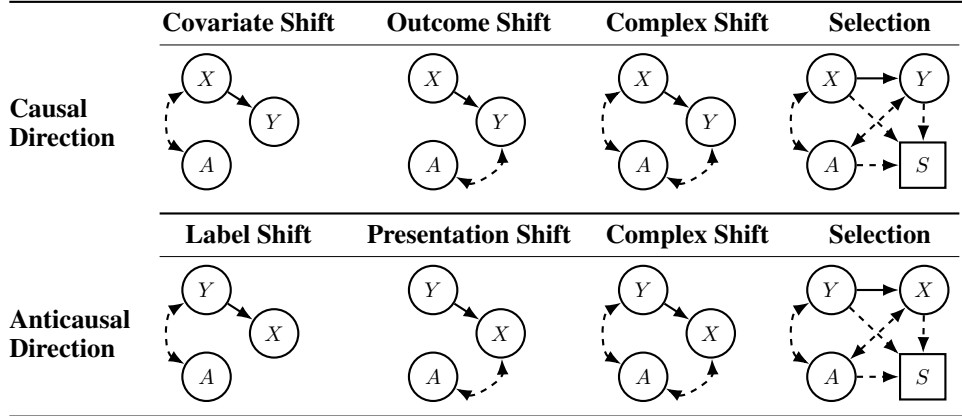

differs across subgroups (*i.e.*, $A \not\perp Y$), fitting a high-quality predictive model for each subgroup implies satisfaction of calibration and sufficiency, but violation of separation and demographic parity, in the sense that empirical risk minimization with a covariate set that encodes $A$ implies some non-zero lower bound on separation and demographic parity error while minimizing an upper bound on calibration and sufficiency error. Other related works [7, 11, 12] show that when allocation on the basis of a predictor is unconstrained and the utility function has a certain structure (*e.g.*, if the benefit of the intervention increases as a function of the risk of the outcome in the absence of the intervention), maximizing prediction performance for each subgroup is consistent with maximizing the utility of the allocation for each subgroup.

## 3 Understanding fairness through causal models of distribution shift

In this section, we provide the basis for understanding the fairness properties that can be expected for models that fit the data well under a variety of assumptions on the data generating process. These properties described here serve as context for the core theoretical results regarding the stability of performance metrics and approach to controlled evaluation across subgroups that are described subsequently in Section 4. In Section 3.1 we detail the data generating processes of interest that we study in this work. We do so through the specification of a collection of causal directed acyclic graphs that incorporate subgroup structure to describe distributional differences across subgroups. In Section 3.2, we characterize the conditional independence properties of subgroup-aware and subgroup-agnostic Bayes-optimal models fit to data drawn from each of the data generating processes of interest.

### 3.1 Describing subgroup heterogeneity with causal directed acyclic graphs

We define a collection of simple, prototypical data generating processes that we use to study the performance and fairness properties of models under varying assumptions regarding causal structure and heterogeneity across subgroups. For that purpose, we use causal directed acyclic graphs (DAGs) [45]. These graphs involve $X$, $Y$, and $A$ and are analogous to those used to describe distribution shift mechanisms in prior works [30–35]. We consider graphs in both the *causal* and *anticausal* directions, that is, when $X$ is a direct parent of $Y$ and when $Y$ is a direct parent of $X$ [46]. While we include $A$ in these graphs to describe the role of heterogeneity across subgroups, we do not consider $A$ to be a direct cause of either $X$ or $Y$. Rather, we use bidirected edges to describe cases where an unobserved confounder that influences $X$ or $Y$ varies in distribution across subgroups [47].

The causal settings that we study are presented in Table 1. In brief, in both the causal and anticausal directions, we consider two simple mechanisms of heterogeneity across subgroups, a complex shift mechanism that combines the two simpler ones, and a configurable graph that incorporates a selection node to describe distribution shift between a source and target distribution. To describe selection mechanisms, we augment each of the causal graphs described thus far with a square selection node $S$.

The set of variables with direct arrows into $S$ are considered those that affect the selection process. If the set of variables with direct edges into $S$ are deemed Pa$(S)$, the distribution $P(S \mid X, Y, A)$ can be simplified to $P(S \mid \text{Pa}(S))$.

For the remainder of this section, we restrict ourself to a setting without selection. In the causal direction, the two simple cases are *covariate shift* and *outcome shift* across subgroups. Informally, covariate shift captures the assumption that an outcome $Y$ is generated from covariates $X$ with the same mechanism for all subgroups; outcome shift captures the assumption that the mechanism differs. For example, if $X$ is a set of clinical covariates (*e.g.*, prior health conditions, lab test results, etc.) and $Y$ is a downstream clinical outcome (*e.g.*, cardiovascular event in the next ten years), the two settings correspond to different assumptions on whether the probability of the clinical outcome differs across subgroups after holding the covariate values constant. In this example, the choice to represent the edges that involve $A$ as bidirected corresponds to an assumption that differences in the distribution of covariates and outcomes across subgroups are not inherent properties of the subgroups but are rather a consequence of differential exposure to unobserved SDOH variables.

More formally, under covariate shift, $P(X) \neq P(X \mid A)$ but $P(Y \mid X) = P(Y \mid X, A)$, encoding the invariance $Y \perp A \mid X$. In other words, the distribution of $X$ differs across subgroups but the conditional distribution $Y \mid X$ is unchanged. Under outcome shift, we assume that $P(X) = P(X \mid A)$, but there exists some unobserved confounder that affects $Y$ and is not independent of $A$, such that $P(Y \mid X) \neq P(Y \mid X, A)$. The complex shift case combines the two settings such that $P(X) \neq P(X \mid A)$ and $P(Y \mid X) \neq P(Y \mid X, A)$.

In the anticausal direction, the two simple cases are *label shift* and *presentation shift*. An example of a plausible anticausal setting is a case where it is of interest to identify a condition $Y$ (*e.g.*, pneumonia) from a chest X-ray [8, 9]. Informally, label shift captures the assumption that conditioned on a class of $Y$, the distributions of covariates $X$ are identical across subgroups, while presentation shift allows this relationship to change and holds the prevalence of $Y$ constant. For example, this could correspond to a setting where it is assumed that the distribution of chest X-rays given the presence/absence of pneumonia is the same across subgroups of differing condition prevalence. Formally, in the label shift case, the prevalence of $Y$ differs across subgroups, but $P(X \mid Y)$ is stable across subgroups (*i.e.*, $P(X \mid Y) = P(X \mid Y, A)$). In the presentation shift case, the prevalence of $Y$ is the same across subgroups, but $P(X \mid Y)$ differs across subgroups (*i.e.*, $P(X \mid Y) \neq P(X \mid Y, A)$). This could occur, for example, if there were systematic differences in the imaging technology used across subgroups. As in the causal direction, a complex anticausal shift can be constructed if we let $P(Y) \neq P(Y \mid A)$ and $P(X \mid Y) \neq P(X \mid Y, A)$.

## 3.2 The effect of causal structure on the properties of Bayes-optimal models

In this section, we describe the conditional independence properties of Bayes-optimal models learned and evaluated on data drawn from each of the DAGs described in Table 1. We do so to establish the properties that are expected to be satisfied for models that fit the data well, as a foundation for the investigation into the stability of performance metrics across subgroups in Section 4. To derive the key results, we consider the conditional independencies among $X$, $Y$, and $A$ that are directly implied by each of the causal graphical structures. This allows us to identify cases where the structure of the causal graph implies that the Bayes-optimal predictor satisfies the sufficiency or separation criteria. We note that we focus on sufficient conditions where Bayes-optimality is sufficient to imply the fairness criteria of interest, which are weaker than impossibility results [37–40].

To evaluate the conditions of interest, we use the conditional independencies implied by the relevant DAG and consider scores $R$ as a deterministic function of $Z \in \{X, \{X, A\}\}$. For conditional independence statements of the form $R \perp V_1 \mid V_2$, for any other variables $V_1$ and $V_2$, we consider $R$ as a deterministic function of $Z$ and use the result that $Z \perp V_1 \mid V_2$ implies $R \perp V_1 \mid V_2$ ([48], Lemma 4.2). For Bayes-optimal models, we additionally have the constraint that $Y \perp Z \mid R$ when $R = f^*(Z)$ is Bayes-optimal.

We present the key results in Supplementary Table B1. We find that while the sufficiency fairness criterion must be satisfied by a subgroup Bayes-optimal predictor regardless of the causal graph, population Bayes-optimal prediction implies sufficiency only when $Y \perp A \mid X$ (*i.e.*, in the covariate shift setting) or in the case of extreme subgroup separability (Section 3.2.1). The condition $Y \perp A \mid X$ is further central to understanding the properties of evaluations that control for the distribution of

**Table 2: Stability of model performance metrics across subgroups**. ✓ indicates cases in which prediction is sufficient to induce $\{R, Y\} \perp A \mid V$; ✗ indicates otherwise. $f$ indicates prediction with an arbitrary model. $f^*$ and $f_A^*$ indicate the population and subgroup Bayes-optimal models.

| Setting | Setting $Z \in \{X, \{X, A\}\}$ | Model | $\{R, Y\} \perp A \mid V$ $V =$ {} | $X$ | $Y$ | $R$ |
|---|---|---|---|---|---|---|
| Covariate shift | $X$ | $f^*$ | ✗ | ✓ | ✗ | ✓ |
| | | $f$ | ✗ | ✓ | ✗ | ✗ |
| | $\{X, A\}$ | $f_A^*$ | ✗ | ✓ | ✗ | ✓ |
| | | $f$ | ✗ | ✗ | ✗ | ✗ |
| Label shift | $X$ | $f^*$ | ✗ | ✗ | ✓ | ✗ |
| | | $f$ | ✗ | ✗ | ✓ | ✗ |
| | $\{X, A\}$ | $f_A^*$ | ✗ | ✗ | ✗ | ✓ |
| | | $f$ | ✗ | ✗ | ✗ | ✗ |
| Other $Y \not\perp A \mid X$ | $X$ | $f^*$ | ✗ | ✗ | ✗ | ✗ |
| $\Big($ Outcome Shift | | $f$ | ✗ | ✗ | ✗ | ✗ |
| Presentation Shift | $\{X, A\}$ | $f_A^*$ | ✗ | ✗ | ✗ | ✓ |
| Complex Shift $\Big)$ | | $f$ | ✗ | ✗ | ✗ | ✗ |

$X$ (Section 4.2) and the potential for subgroup-aware modeling to improve over subgroup-agnostic modeling. In particular, when the causal graph implies $Y \perp A \mid X$, the population Bayes-optimal predictor is also subgroup Bayes-optimal and there is no expected benefit to subgroup-aware modeling. In contrast, when $Y \not\perp A \mid X$, the population Bayes-optimal predictor is not generally subgroup Bayes-optimal, and subgroup-aware prediction may improve performance.

Analysis of the separation criterion serves as an interesting special case for our general findings regarding the stability of performance metrics across subgroups, given the connection between separation and equalized odds (*i.e.*, the control of true positive and false positive error rates across subgroups). We find that the separation criterion is implied only by the label shift graph and then only when prediction is subgroup-agnostic, regardless of whether the model is Bayes-optimal or arbitrary. This is consistent with prior findings [37] showing that models that fit the data well for subgroups satisfy sufficiency but do not generally satisfy the separation and equalized odds criteria.

In Supplementary Section A.2, we extend this analysis to settings with selection bias, enumerating the conditions analogous to those described thus far for each of the base graphs augmented with a selection mechanism that depends on combinations of $X$, $Y$, and $A$ (Supplementary Table B2). For brevity, we defer a summary of the results to Supplementary Section A.2.

### 3.2.1 Subgroup Separability

We highlight the properties of a special case where the distribution of covariates are separable across subgroups [9, 49]. In this setting, there is little overlap in the distribution of $X$ for any pair of subgroups $a_i$ and $a_j$, such that the ratio $\frac{P(X|A=a_i)}{P(X|A=a_j)}$ is large for any $X$ where $P(X \mid A = a_i)$ is non-trivial. This implies that each region of $X$ with non-trivial density is associated with exactly one subgroup, and $A$ can be predicted with high accuracy from $X$.

Under subgroup separability, we observe that the population and subgroup Bayes-optimal models behave similarly, as if $Y \perp A \mid X$, regardless of the underlying causal structure that generated the data. To see this, consider that $\mathbb{E}[Y \mid X] = \sum_{a \in \mathcal{A}} \mathbb{E}[Y \mid X, A = a] P(A = a \mid X)$ [49]. As such, for $X$ where $P(X \mid A) \gg 0$, we have that $P(A \mid X) \approx 1$ and $\mathbb{E}[Y \mid X] \approx \mathbb{E}[Y \mid X, A]$. We then have that, as in the case of covariate shift, the population Bayes-optimal predictor satisfies sufficiency and control for $X$ is enough to explain differences in performance for any performance metric.

## 4  Understanding disaggregated evaluations of algorithmic fairness

We now present the core results regarding the stability of performance metrics across subgroups for both optimal and arbitrary subgroup-aware and subgroup-agnostic models. We argue that distri-

butional differences across subgroups directly contribute to model performance differences across subgroups, constituting a form of confounding, distinct from the effects of estimation error. Critically, even Bayes-optimal models do not generally achieve equal performance across subgroups despite zero estimation error. However, we show in Section 4.1 that the stability of performance metrics across subgroups can be understood as a consequence of causal structure. Finally, in Section 4.2, we show how this characterization motivates an approach to controlled evaluation across subgroups.

## 4.1 Performance metric stability

To characterize the stability of model performance across subgroups for each of the causal settings discussed thus far, we rely on a sufficient condition for equal average performance across subgroups for arbitrary performance metrics. Specifically, we note that $\{R, Y\} \perp A$ is sufficient to induce $\mathbb{E}[m(R, Y) \mid A] = \mathbb{E}[m(R, Y)]$ for any performance metric $m$.

A key result is that in *none* of the settings discussed thus far does $\{R, Y\} \perp A$ hold (Table 2), implying that in general we should not expect equal performance across subgroups for an arbitrary performance metric. This simple observation regarding instability of model performance across subgroups can be enriched if we consider the extent to which differences in model performance metrics across subgroups can be explained by causal structure. We do so here through characterization of the condition $\{R, Y\} \perp A \mid V$ for a control variable $V \in \{X, Y, R\}$ for each of the causal settings discussed, focusing on cases where selection bias is not present. The condition $\{R, Y\} \perp A \mid V$ then serves as a sufficient condition for $\mathbb{E}[m(R, Y) \mid V, A] = \mathbb{E}[m(R, Y) \mid V]$. We then say that differences in performance across subgroups are explained by $V$ if we have $\mathbb{E}[m(R, Y) \mid A] \neq \mathbb{E}[m(R, Y)]$, but $\{R, Y\} \perp A \mid V$ and $V \not\perp A$.

In Table 2, we summarize whether control for $V \in \{X, Y, R\}$ is sufficient to induce $\{R, Y\} \perp A \mid V$ based on the conditional independencies implied by each of the graphical settings discussed. These results are relevant to both Bayes-optimal and arbitrary models, with some differences depending on whether optimality is achieved. For models that are subgroup-agnostic, we find that $X$ explains performance differences only for the covariate shift graph, and $Y$ explains performance differences only for the label shift graph. However, these relationships no longer hold if the model is subgroup-aware because now $\{R, Y\} \perp A \mid V$ no longer necessarily holds. As an exception, in the covariate shift graph, subgroup Bayes-optimality implies equal performance conditioned on $X$.

In graphs where $Y \not\perp A \mid X$, performance differences are expected in general and neither $X$ nor $Y$ alone explains those differences. However, note that when $V = R$, $\{R, Y\} \perp A \mid V$ reduces to the sufficiency fairness criterion. This implies that for any model satisfying sufficiency, differences in performance can be explained by differences in the distribution of the score $R$.

## 4.2 Controlled evaluation

The observation that performance differences across subgroups may be attributed to specific distributional differences across subgroups suggests that evaluations that control for this source of confounding may be constructed through balancing of the confounding variable $V$ in comparisons of model performance across subgroups or between subgroups and the overall population. Here, we present an approach to doing so, building off of Cai et al. [16], and discuss the interpretation of such evaluations for algorithmic fairness. Note that such evaluations do not require an assumption of Bayes-optimality, even if the interpretation can depend on whether the model is optimal. In Supplementary Table B3, we detail the set of implications and conclusions that can be drawn from controlled evaluations in a setting where the causal graph is unknown under various assumptions (*i.e.*, the choice of control and covariate variable sets and whether the model is Bayes-optimal).

We consider a controlled comparison of each subgroup with the overall population. Specifically, we compare the performance of a model for subgroup $A = a$ with the performance $M_a$ on the weighted aggregate distribution $\mathcal{Q} := \int P(R, Y \mid V = v) P(V = v \mid A = a) \mathrm{d}v$, corresponding to sampling $V$ from the subgroup distribution $P(V \mid A = a)$ and $\{R, Y\}$ from the aggregate population distribution $P(R, Y \mid V) = \int P(R, Y \mid V, A = a') P(A = a' \mid V) \mathrm{d}a'$. Concretely, $M_a$ can be computed via a weighted average in the aggregate population with weights $w \propto P(A = a \mid V)$:

$$M_a := \int \frac{P(A = a \mid v)}{P(A = a)} m(R, Y) P(R, Y \mid v) P(v) \mathrm{d}v = \int w * m(R, Y) P(R, Y \mid v) P(v) \mathrm{d}v.$$

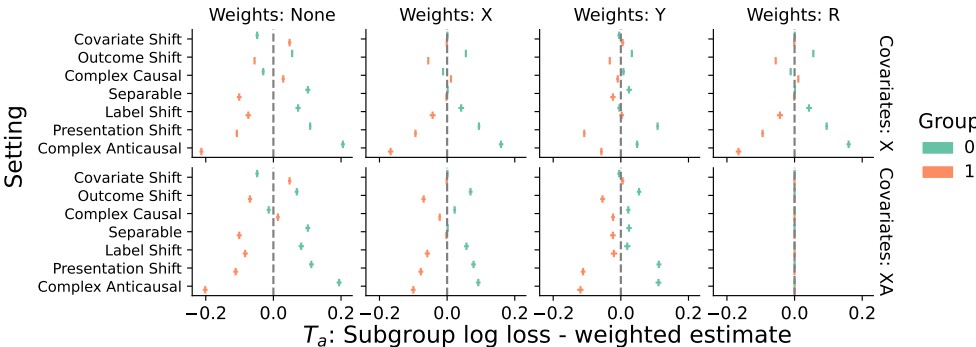

**Figure 1: Controlled evaluation for confounding across subgroups**. Plotted are the statistics $T_a$ with 95% confidence intervals, corresponding to differences between unweighted disaggregated performance with population performance weighted to match the distribution of $X$, $Y$, or $R$ on the subgroups. The first row corresponds to subgroup-agnostic prediction and the second row corresponds to subgroup-aware prediction where $A$ is used as a covariate.

We then define $T_a := \mathbb{E}[m(R, Y) \mid A = a] - M_a$ as the difference between the performance for subgroup $A = a$ and $M_a$. In Supplementary Section A.1, we provide additional details regarding this approach and describe alternative configurations of the weights, including the approach of Cai et al. [16] that maps the data to a region of shared overlap between a pair of distributions. Pseudo-code for calculating $T_a$ in a setting where $P(A = a \mid V)$ is given or estimated is given in Algorithms 1 and 2, respectively.

This form of controlled evaluation is perhaps best understood as a form of conditional independence testing. Consider that if $\{R, Y\} \perp A \mid V$, the distribution that $M_a$ is computed with respect to is identical to the subgroup distribution, and thus $T_a$ must be zero. It follows that a hypothesis test for whether $T_a$ differs from zero corresponds to a conditional independence test against the null hypothesis that $\{R, Y\} \perp A \mid V$. A few specific configurations of $V$ and $Z$ correspond to notable conditional independence tests. For example, when $V = X$ and $Z = X$, the test corresponds to a test against the null hypothesis $Y \perp A \mid X$. Furthermore, if $V = R$, regardless of whether $Z$ is $X$ or $\{X, A\}$, the test corresponds to a test for sufficiency, as $\{R, Y\} \perp A \mid R$ simplifies to $Y \perp A \mid R$.

For the causal settings we study in this work, $T_a$ is expected to be zero when $V$ is chosen to be a variable marked with ✓ in Table 2. Therefore, in a setting where the causal structure is not known, observing that $T_a$ is non-zero provides evidence against the set of assumptions and conditions that imply $\{R, Y\} \perp A \mid V$. For example, observing that $T_a$ for a subgroup-agnostic model is non-zero after controlling for $X$ provides evidence against the hypothesis that the data were generated under the covariate shift graph, regardless of whether the model is Bayes-optimal or arbitrary.

Viewing controlled evaluation as a form of conditional independence testing clarifies some challenges related to interpretation. For example, as is shown in Table 2, any subgroup-agnostic predictor satisfies $\{R, Y\} \perp A \mid X$ in a covariate shift setting. The implication is that a weighted evaluation procedure for $X$ in a covariate shift setting does not distinguish between differences in model performance that would remain under Bayes-optimality from those that follow from poor fit of the model (*e.g.*, due to underrepresentation or model misspecification) for some values of $X$ of differing prevalence across subgroups. Similarly, an evaluation that controls for $R$ inherits the same limitations as a test for sufficiency, in that it is possible for a poorly-fit model to satisfy sufficiency [11, 19].

## 5 Experiments

We conduct a simulation study and experiment with real-world tabular data. The purpose of these experiments is to empirically verify the properties discussed in Section 3. We briefly describe the design of the experiments here and defer additional details to Supplementary Sections A.3 and A.4. For the simulation study, we generate data corresponding to each of the settings described in Table 1. For the real-world data experiments, we follow Ding et al. [50] to derive prediction tasks from the American Community Survey (ACS) Public Use Microdata Sample (PUMS) provided by the

U.S. Census Bureau [51]. We use the 'ACSIncome' (prediction of whether an individual's income exceeds $50,000$) and 'ACSPublicCoverage' (prediction of whether an individual is enrolled in a public health insurance plan) task definitions [50], stratifying the data by race and ethnicity (cohort characteristics provided in Supplementary Table B4).

The properties we assess mirror those discussed in Section 3: we assess properties related to (1) the informativeness of subgroup membership (Section 3.2), (2) calibration and sufficiency (Section 3.2), and (3) controlled evaluation (Section 4.2). To assess properties with respect to the informativeness of subgroup membership, we assess differences in performance between subgroup-aware and subgroup-agnostic models (as in Liu et al. [52]). In the main text, we report results for subgroup-aware models that leverage $A$ as a covariate and include additional results with stratified models in the supplementary material. To assess calibration and sufficiency violation, we visualize calibration curves for each subgroup and conduct evaluations that control for $R$. To verify empirical consistency with the properties presented in Table 2 and Supplementary Table B3, we compute $M_a$ and $T_a$ to control for $X, Y$, or $R$ in the evaluation of model performance.

**Simulation study results:** The results of the simulation study generally coincide with those that are expected based on the theoretical analysis. We find that subgroup-aware models generally improve predictive performance overall and for subgroups when $Y \not\perp A \mid X$, *i.e.*, in all settings except for covariate shift, with the exception of the case where subgroups are separable based on $X$ (Supplementary Figure B1). Specifically, we observe improvements overall and for subgroups in log-loss and non-negative changes in AUC-ROC. We note that sensitivity and specificity do not strictly improve; rather an increase in one is often paired with a reduction in the other. However, we observe that net benefit [53], a decision-theoretic metric that combines sensitivity and specificity based on an assumed tradeoff between true positives and false negatives, improves like the log-loss.

The effect of subgroup-aware prediction on calibration and sufficiency mirrors the effects on overall model performance (Supplementary Figure B2). In cases where $Y \not\perp A \mid X$ and prediction is subgroup-agnostic, we observe miscalibration for subgroups and sufficiency violation. In all cases, subgroup-aware prediction results in calibrated models that satisfy sufficiency. As expected, there is no difference in calibration between subgroup-agnostic and subgroup-aware models when there is minimal overlap in $X$ across subgroups.

We conduct a small experiment to verify a subset of the properties expected under selection. We adapt the complex causal shift setting and introduce three selection mechanisms corresponding to cases where the selection mechanism depends on $X, Y$, or $\{Y, A\}$. Further methodological details are provided in Supplementary Section A.3.1. We visualize calibration curves in Supplementary Figure B3, finding that they correspond to the properties presented in Supplementary Table B2.

We find the results of controlled evaluation are generally consistent with those presented in Table 2. In general, we find that only for the covariate shift graph or when subgroups are separable, control for $X$ is sufficient to control for differences in the average log loss between the population and subgroups for subgroup-agnostic models, consistent with the interpretation of this form of the controlled evaluation as a conditional independence test for $Y \perp A \mid X$ (Figure 1). This same pattern holds for other metrics (Supplementary Figures B5–B10). We further find that performance is not generally stable after control for $Y$, except in the label shift setting, and then only when prediction is subgroup-agnostic. Consistent with the interpretation of control for $R$ as corresponding to a test for sufficiency, we note that the statistic $T_a$ takes on a value not statistically significantly different from zero in cases where calibration and sufficiency are satisfied and a non-zero value otherwise. For completeness, we report the absolute values of the performance of each of the models of interest in each setting for each subgroup as well as the weighted population estimates $M_a$ (Supplementary Figures B11–B17). For comparison, we further apply the approach of Cai et al. [16] (Supplementary Figures B18–B24), finding that the results are largely qualitatively consistent with ours.

**Results on ACS PUMS:** For the experiments with ACS PUMS, we report the change in performance attained through subgroup-aware prediction (Supplementary Figure B25), visualize calibration curves (Supplementary Figure B26), and apply the approach to controlled evaluation (Supplementary Figures B27–B40). While the structure of the causal graph underlying these data is not available, our approach can be used for conditional independence testing and attribution of performance differences to distribution shifts. We generally observe greater evidence against the hypothesis $Y \perp A \mid X$ for the 'ACSPublicCoverage' task than we do for the 'ACSIncome' task. Specifically, for 'ACSPublic-

Coverage', we find that subgroup-aware prediction outperforms subgroup-agnostic prediction for nearly all race/ethnicity subgroups (Supplementary Figure B25) and that differences in performance between subgroups and the aggregate population are not generally explained by control for $X$; for 'ACSIncome', improvements in performance through subgroup-aware prediction are more minor, and differences in performance are explained to a greater degree by $X$ (Supplementary Figure B27). Furthermore, while the extent of miscalibration appears to be minor for both tasks (Supplementary Figure B26), we do observe evidence of sufficiency violation for subgroup-agnostic prediction for both tasks, affecting the "Other" subgroup for 'ACSIncome' and nearly all subgroups for 'ACSPublicCoverage'. Stratified prediction, but not subgroup-aware prediction with a race/ethnicity covariate mitigates the sufficiency violation for the "Other" subgroup for the 'ACSIncome' task, potentially as a result of suboptimality due to relative underrepresentation of this subgroup in the data.

## 6   Discussion

Our work highlights the challenges of interpretation of disaggregated evaluations over subgroups, consistent with findings of prior work studying related challenges for evaluation under distribution shift (*e.g.*, Cai et al. [16] and Liu et al. [52]) and incompatibility among fairness criteria [6, 7, 12, 19, 37–39]. Through vignettes corresponding to prototypical data generating processes represented as causal DAGs, we show how performance differences across subgroups can arise as a consequence of conditional dependencies implied by causal structure. This perspective further allows for understanding the properties of evaluations designed to control for confounding across subgroups, as well as the effects that subgroup-aware prediction and subgroup separability have on the properties of models.

Our findings suggest that disaggregated evaluations may be enriched if combined with controlled procedures that effectively match the distribution of a set of control variables across subgroups. We interpret such evaluations as a form of conditional independence test that, when rejected, provides evidence towards falsification of assumptions consistent with the conditional independence criteria of interest (*e.g.*, causal assumptions). More broadly, controlled evaluation allows for assessment of hypotheses related to causal structure and aids in the assessment of model fit and fairness violation. We provide further guidance as to the interpretation of such evaluations in Supplementary Table B3.

Our work has several limitations that pose opportunities for future work. As our scope was limited to the analysis of binary outcomes, our findings may not generalize to continuous or multi-class outcomes. Furthermore, while the dependence of several of the findings on an assumption of Bayes-optimality clarifies that the challenges related to the interpretation of disaggregated evaluations remain even under optimal prediction, it may limit the extent to which those findings are relevant in practical settings where Bayes-optimality cannot be assumed nor directly tested for. Relatedly, our analysis of controlled evaluations assumed access to an oracle weight model $P(A \mid V)$, which must be estimated in practice. There is a need for future work to conduct further theoretical and empirical characterization of the role that estimation error has on the properties that we study here, *e.g.*, by casting the evaluation procedure within a semiparametric estimation framework that considers the models for $Y$ and $A$ as imperfectly estimated nuisance functions [54, 55].

We offer a concrete conceptual reorientation of analytic practice for algorithmic fairness. First, it is important to understand differences in optimal model performance across subgroups as a consequence of distributional differences that may be caused by disparities. Observing model performance differences thus motivates deeper investigation to understand the causes of distributional differences and to disambiguate such differences from observational biases (*e.g.*, selection bias) and estimation error. Second, we argue that fairness (as well as related concepts such as equity or justice) is a downstream property of a policy or intervention that may leverage a predictive model [13, 56–58]. Subgroup performance differences are then relevant to fairness to the extent that they reflect a contextually-meaningful notion of fairness in the context of the policy. Our results show that *if* it is of interest to model well outcomes that may be disparate across subgroups, we should not in general expect parity in model performance across subgroups. However, we emphasize that the assumptions underpinning this setting should not be taken for granted, and that there are several unresolved normative questions outside of the scope of this work pertaining to the justification for the prediction of disparate outcomes across populations and control for proxies of disparity in evaluations. Designing effective evaluation procedures that are grounded in understanding of both the societal context contributing to inequities and the capacity for interventions and policies that incorporate predictive models to promote equity and fairness goals is a critical area of future work.

## Acknowledgments and Disclosure of Funding

This study was funded by Google LLC and/or subsidiary thereof (Google). SRP, NH, CN, DM, AwD, SS, NE, HC-L, KH, and AlD are or were employees of Google and may own stock as a part of a standard compensation package. SK acknowledges support by NSF 2046795 and 2205329, IES R305C240046, ARPA-H, the MacArthur Foundation, Schmidt Sciences, Stanford HAI, and Google.

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

## Supplementary Material

## A    Supplementary methods

### A.1    Weighting approaches to controlled evaluation

The approach to weighting model performance metrics to control for distributional differences across subgroups builds on a general approach to estimating model performance under distribution shift. For a source distribution $\mathcal{P}$ and target distribution $\mathcal{Q}$ over $\{R, Y\}$, performance on $\mathcal{Q}$ can be estimated using data from $\mathcal{P}$ with appropriate weights. Formally, this is $\mathbb{E}_{\mathcal{Q}}[m(Y, R)] = \mathbb{E}_{\mathcal{P}}[w * m(Y, R)]$ for $w \propto \frac{P_{\mathcal{Q}}(R,Y)}{P_{\mathcal{P}}(R,Y)}$, assuming positivity $(P_{\mathcal{Q}}(R, Y) > 0 \rightarrow P_{\mathcal{P}}(R, Y) > 0)$. From some mixture distribution over $\mathcal{P}$ and $\mathcal{Q}$, where examples from $\mathcal{Q}$ are indexed by $D = 1$ and those from $\mathcal{P}$ by $D = 0$, we note that the weights can be reformulated as $w \propto \frac{P(R,Y|D=1)}{P(R,Y|D=0)} \propto P(D = 1 \mid R, Y)$.

As described in Section 4.2, we construct controlled evaluations by setting the distribution of a variable $V$ to a reference distribution. We consider a source distribution $\mathcal{P}$ and target distribution $\mathcal{Q}$ over $V$ such that we are computing the performance in $\mathcal{P}$ after fixing the marginal to $P_{\mathcal{Q}}(V)$. The resulting expectation is given by $\int w * m(R, Y) P_{\mathcal{P}}(R, Y \mid v) P_{\mathcal{P}}(v) dv$ for weights $w \propto \frac{P_{\mathcal{Q}}(V)}{P_{\mathcal{P}}(V)} \propto P(D = 1 \mid V)$, where $D$ indicates the identity of the distribution, as before.

In this work, we primarily consider setting $\mathcal{P} = P(R, Y)$ and $\mathcal{Q} = P(R, Y \mid A = a)$ with weights $w \propto P(A = a \mid V)$. This yields a mapping of the marginal population distribution $P(V)$ onto the subgroup distribution $P(V \mid A = a)$, implicitly fixing the conditional distribution of $P(R, Y \mid V)$ to that of the aggregate population. It follows that if $\{R, Y\} \perp A \mid V$, then $P_{\mathcal{P}}(R, Y \mid v) = P_{\mathcal{Q}}(R, Y \mid v)$ and thus weighted performance $M_a := \int w * m(R, Y) P(R, Y \mid v) P(v) dv$ in $\mathcal{P}$ is equal to the marginal performance $\mathbb{E}[m(R, Y) \mid A = a]$ in $\mathcal{Q}$. If weighted performance in $\mathcal{P}$ is not equal to the marginal performance in $\mathcal{Q}$, then $\{R, Y\} \not\perp A \mid V$.

It is possible to construct alternative weight configurations that can be used for the same purpose as the approach that we present. For example, if the distributions are set such that $\mathcal{P} = P(R, Y \mid A = a)$ and $\mathcal{Q} = P(R, Y)$, then the weights are given by $w \propto \frac{1}{P(A=a|V)}$, corresponding to a mapping of the subgroup $A = a$ onto the population distribution, analogous to approaches to inverse propensity score weighting for treatment effect estimation [59]. Alternatively, if we consider a pair of subgroups $A = a$ and $A = a'$ where $\mathcal{P} = P(R, Y \mid A = a)$ and $\mathcal{Q} = P(R, Y \mid A = a')$, the weights are given by $w \propto \frac{P(A=a'|V)}{P(A=a|V)}$. However, both of these formulations are susceptible to returning unstable and high-variance estimates when small values in the denominator induce extreme weights. We note that our approach is not susceptible to this issue given that $P(A = a \mid V)$ is bounded between 0 and 1.

Cai et al. [16] presented an alternative weighting strategy for pairwise comparisons motivated to address the extreme weights issue by considering mapping the data to a *shared distribution* of sufficient overlap between the distributions of covariates for a pair of subgroups. This approach considers a target distribution $\mathcal{Q} \propto \frac{P(V|A=a)P(V|A=a')}{P(V|A=a)+P(V|A=a')}$ for two source distributions $\mathcal{P}_a$ and $\mathcal{P}_{a'}$. This has the effect of defining a target density that takes on a value of zero in cases where the control variable has zero support in either of the subgroup distributions. The weights for this approach are given by $w \propto \frac{P(A=a'|V)}{P(A=a)P(A=a'|V)+P(A=a')P(A=a|V)}$ for $A = a$ and $w \propto \frac{P(A=a|V)}{P(A=a)P(A=a'|V)+P(A=a')P(A=a|V)}$ for $A = a'$.

### A.2    Properties under selection

In this section, we consider fairness properties under explicit distribution shift through the lens of selection bias [25]. We consider models fit using data drawn from the selected, *source* population $P(X, Y, A \mid S = 1)$ and reason about the properties of those models evaluated on samples drawn from the full, *target* population without selection $P(X, Y, A)$. Graphically, we represent the selection variable $S$ with a square node to indicate conditioning on selection in the observed data, where a directed edge in to $S$ indicates dependence of the selection mechanism on the originating node (Table 1). This formulation allows for us to anticipate the fairness properties of models under selection based on the structure of the causal graph and the connectivity of the selection node (Supplementary Table B1).

**Algorithm 1: Calculation of the weighted metric difference $T_a$ with oracle access to $P(A = a \mid V)$.** This procedure computes the difference between the mean of a metric function $m$ evaluated on a subgroup $A = a$ and the weighted metric $M_a$ computed on the aggregate data. The weighted metric $M_a$ corresponds to evaluation of $m$ with respect a distribution where $V \sim P(V \mid A = a)$ and $Y \sim P(Y \mid V)$. See Section 4.2 and Supplementary Section A.1 for further details

**Data:**

    Dataset $\mathcal{D} = \{(X_i, Y_i, A_i, R_i)\}_{i=1}^N$
    Target subgroup $a \in \mathcal{A}$
    Metric function $m : \mathcal{Y} \times \mathcal{R} \to \mathbb{R}$
    Selection function `get_v` that selects $V$ from $\{X, Y, R\}$
    Function $g : \mathcal{V} \to [0, 1]$, that returns $P(A = a \mid V)$

**Result:** The statistic $T_a$

**begin**

```
//Get weights for all N data points
```
    **for** $i = 1$ **to** $N$ **do**
        $v_i \leftarrow$ `get_v`$(X_i, Y_i, R_i)$         `//Extract features V for instance i`
        $w_i \leftarrow g(v_i)$                  `//Apply model g to get weight` $w_i$
    **end**
```
//Calculate weighted and subgroup means
```
    $M_a \leftarrow \frac{\sum_{i=1}^N w_i \cdot m(Y_i, R_i)}{\sum_{i=1}^N w_i}$         `//Weighted mean metric`
    $M_{\text{mean}} \leftarrow \frac{\sum_{i=1}^N \mathbb{1}(A_i=a) \cdot m(Y_i, R_i)}{\sum_{i=1}^N \mathbb{1}(A_i=a)}$     `//Mean metric on subgroup A=a`
```
//Compute the final statistic
```
    $T_a \leftarrow M_{\text{mean}} - M_a$
    **return** $T_a$

**end**

.

Intuitively, the model learned in the selected population generalizes to the full population if $\mathbb{E}[Y \mid Z, S = 1] = \mathbb{E}[Y \mid Z]$, for a set of predictor variables $Z$. To reason about this, we consider the sufficient condition that $S \perp Y \mid Z$, which implies that

$$\frac{P(Y \mid Z)}{P(Y \mid Z, S = 1)} = \frac{P(S = 1 \mid Z)}{P(S = 1 \mid Z, Y)} = 1. \tag{1}$$

The implication is that a Bayes-optimal model will generalize under selection if the variables used as predictors d-separate the selection node $S$ from $Y$. For example, in a graph where the subgroup covariate shift assumption holds and selection depends on $X$ and $A$, a Bayes-optimal model fit in the selected population using only $X$ generalizes to all subgroups in the full population. However, following the results described previously, average performance will not be stable under selection, generally, or across subgroups in the selected or full population.

For cases where the graph structure does not imply that a model learned in the selected population is subgroup Bayes-optimal in the full target population, we reason about whether sufficiency is implied by reasoning about whether each of the ratios in the following identity are equal to 1:

$$\underbrace{\frac{P(Y \mid R, A)}{P(Y \mid R)}}_{Y \perp A \mid R} = \underbrace{\frac{P(Y \mid S = 1, R, A)}{P(Y \mid S = 1, R)}}_{Y \perp A \mid R, S=1} * \underbrace{\frac{P(S = 1 \mid R, A)}{P(S = 1 \mid R)}}_{S \perp A \mid R} * \underbrace{\frac{P(S = 1 \mid Y, R)}{P(S = 1 \mid Y, R, A)}}_{S \perp A \mid Y, R} \tag{2}$$

In other words, if the model satisfies sufficiency in the selected population ($Y \perp A \mid R$) and the selection node is d-separated from $A$ given $R$ and $\{R, Y\}$, then the model satisfies sufficiency in the full population. This allows us to, for example, claim that a population Bayes-optimal model satisfies sufficiency in the subgroup covariate shift graph if selection depends only on $Y$, such that the model is miscalibrated to the same extent for all subgroups. We note that equation (2) permits a re-ordering

**Algorithm 2: Calculation of the weighted metric difference $T_a$ with cross-fitting.** This procedure computes the difference between the mean of a metric function $m$ evaluated on a subgroup $A = a$ and the weighted metric $M_a$ computed on the aggregate data. The weighted metric $M_a$ corresponds to evaluation of $m$ with respect a distribution where $V \sim P(V \mid A = a)$ and $Y \sim P(Y \mid V)$. See Section 4.2 and Supplementary Section A.1 for further details.

**Data:**

 Dataset $\mathcal{D} = \{(X_i, Y_i, A_i, R_i)\}_{i=1}^N$
 Target subgroup $a \in \mathcal{A}$
 Metric function $m : \mathcal{Y} \times \mathcal{R} \to \mathbb{R}$
 Number of folds $K$
 Function `get_foldid` that maps data indices $i \in \{1, \dots, N\}$ to fold indices $j \in \{1, \dots, K\}$
 Selection function `get_v` that selects $V$ from $\{X, Y, R\}$
 Algorithm `Alg` for estimating $P(A = a \mid V)$

**Result:** The statistic $T_a$

**begin**

 Split data $\mathcal{D}$ into $K$ folds using `get_foldid`: $\{\mathcal{D}_j\}_{j=1}^K$
 `//Train K models for` $P(A = a \mid V)$`, one for each fold`
 **for** $j = 1$ **to** $K$ **do**
  $\mathcal{D}_{\text{train}} \leftarrow \bigcup_{m \neq j} \mathcal{D}_m$        `//Use K-1 folds for training`
  $g_j \leftarrow \texttt{Alg}(\mathcal{D}_{\text{train}})$         `//Train model` $g_j$
 **end**
 `//Get held-out predictions for all N data points`
 **for** $i = 1$ **to** $N$ **do**
  $v_i \leftarrow \texttt{get\_v}(X_i, Y_i, R_i)$    `//Extract features V for instance i`
  $j \leftarrow \texttt{get\_foldid}(i)$     `//Get the fold index for instance i`
  $w_i \leftarrow g_j(v_i)$       `//Apply model` $g_j$ `to get weight` $w_i$
 **end**
 `//Calculate weighted and subgroup means`
 $M_a \leftarrow \dfrac{\sum_{i=1}^N w_i \cdot m(Y_i, R_i)}{\sum_{i=1}^N w_i}$      `//Weighted mean metric`
 $M_{\text{mean}} \leftarrow \dfrac{\sum_{i=1}^N \mathbb{1}(A_i = a) \cdot m(Y_i, R_i)}{\sum_{i=1}^N \mathbb{1}(A_i = a)}$   `//Mean metric on subgroup A=a`
 `//Compute the final statistic`
 $T_a \leftarrow M_{\text{mean}} - M_a$
 **return** $T_a$

**end**

---

that allows for reasoning about a different set of conditional independencies:

$$\underbrace{\frac{P(Y \mid R, A)}{P(Y \mid R)}}_{Y \perp A \mid R} = \underbrace{\frac{P(Y \mid S = 1, R, A)}{P(Y \mid S = 1, R)}}_{Y \perp A \mid R, S=1} * \underbrace{\frac{P(S = 1 \mid Y, R)}{P(S = 1 \mid R)}}_{S \perp Y \mid R} * \underbrace{\frac{P(S = 1 \mid R, A)}{P(S = 1 \mid Y, R, A)}}_{S \perp Y \mid R, A} \tag{3}$$

For reasoning about separation, we use analogous logic and reason about the terms in the following two identities:

$$\underbrace{\frac{P(R \mid Y, A)}{P(R \mid Y)}}_{R \perp A \mid Y} = \underbrace{\frac{P(R \mid Y, A, S = 1)}{P(R \mid Y, S = 1)}}_{R \perp A \mid Y, S=1} * \underbrace{\frac{P(S = 1 \mid Y, A)}{P(S = 1 \mid Y)}}_{S \perp A \mid Y} * \underbrace{\frac{P(S = 1 \mid Y, R)}{P(S = 1 \mid Y, R, A)}}_{S \perp A \mid Y, R}, \tag{4}$$

and

$$\underbrace{\frac{P(R \mid Y, A)}{P(R \mid Y)}}_{R \perp A \mid Y} = \underbrace{\frac{P(R \mid Y, A, S = 1)}{P(R \mid Y, S = 1)}}_{R \perp A \mid Y, S=1} * \underbrace{\frac{P(S = 1 \mid Y, R)}{P(S = 1 \mid Y)}}_{S \perp R \mid Y} * \underbrace{\frac{P(S = 1 \mid Y, A)}{P(S = 1 \mid Y, R, A)}}_{S \perp R \mid Y, A}. \tag{5}$$

To summarize the results that follow (Supplementary Table B2), a Bayes-optimal model generalizes under selection if the variables used as covariates d-separate the selection node $S$ from $Y$ in the

causal graph. For example, when selection depends on only $X$ or $A$, subgroup Bayes-optimal predictors generalize and satisfy sufficiency in the target domain, regardless of other components of the graph. When selection depends only on $Y$, subgroup Bayes-optimality implies sufficiency without calibration in the target domain. When selection depends on $Y$ and either $X$ or $A$, subgroup Bayes-optimality in the source domain does not imply sufficiency in the target domain. Furthermore, separation in the target domain is implied by Bayes-optimal prediction only in the label shift graph when the selection mechanism does not depend on $A$, and then only when the model does not depend on $A$.

## A.3 Simulation study

We conduct a simulation study to verify the properties studied in this work. We construct data generating processes satisfying each of the settings studied in this work. The data generating processes are provided in Supplementary Section A.3.1. For each data generating process, we sample 70,000 independent samples and use 50,000 for training and 20,000 as a held-out testing dataset for evaluation.

All model fitting and evaluation procedures are repeated and conducted separately for cases where prediction of $Y$ is conducted with (1) $X$ alone, (2) $X$ and an additional categorical covariate indicating subgroup membership $A$, and (3) a set of models using $X$ alone fit separately for each subgroup. For model fitting, we use the scikit-learn version [60] 1.6.1 implementation of gradient boosting classification trees (specifically, `HistGradientBoostingClassifier`) with stratified five-fold cross-validation, with a hyperparameter grid over the maximum number of leaf nodes in $\{10, 25, 50\}$, refitting the model over the training data using the hyperparameter setting with the minimum average log-loss over the held-out cross-validation folds. The refit model is then used to make predictions on the held-out testing data.

For the experiment involving selection, we modify the complex causal shift graph with three selection mechanisms (see Supplementary Section A.3.1), and for each data generating process, sample 50,000 samples conditioned on $S = 1$ for training and 20,000 samples from the full population (*i.e.*, without selection). We repeat the same training procedure described above.

To get estimates of $P(A \mid V)$ for use in weighted estimation of model performance, we fit models to predict $A$ from $V$. When $V$ is $X$ or $Y$, the fitting procedure is identical to that used for fitting the models for $Y$, in that we conduct cross-validation with gradient boosting trees on the training data and make predictions with the resulting model on the testing dataset. For cases where $V = R$, we instead conduct a nested cross-validation procedure using only the testing data, similar to cross-fitting [55]. Here, we use an outer stratified five-fold cross-validation partition of the testing data, which, for each outer fold, conducts an inner stratified five-fold cross-validation procedure that returns a model used to make held-out predictions on the corresponding held-out outer fold. Metrics are then computed on the full test set.

For evaluation, we compute unweighted and weighted performance estimates for the log-loss, area under the receiver operating characteristic curve (AUC-ROC), sensitivity (recall), specificity, precision, and net benefit [53]. We compute sensitivity, specificity, and precision using a threshold of 0.5. For net benefit, we use the parametrization presented in Pfohl et al. [7], where the preference trade-off is encoded by a choice of threshold. We use a threshold of 0.5 for both the classifier decision threshold and the preference trade-off threshold. To generate confidence intervals, we use the percentile bootstrap with 10,000 bootstrap samples of the testing data. For weighted metrics, the un-normalized sample weights are treated as fixed based on the result of the procedure described above and sampled alongside the data elements. The resulting samples are then used for weighted computation of metrics.

To generate calibration curves, we quantile-discretize the range of scores $R$ into ten bins and take the empirical mean of $Y$ for the data in each bin. We compute confidence intervals for each bin separately using the Wilson Score Interval Method [61] with the implementation provided by the Statsmodels package version 0.12.1 [62].

The simulation study was conducted on machines with 32 CPUs and 32 GB of RAM. Computing the bootstrap confidence intervals for each of the settings and metrics was the most significant contributor to the overall run time, taking approximately two hours per setting (*i.e.*, approximately 14 hours for the seven settings considered in the primary analyses).

### A.3.1 Data generating processes

**Causal-direction data generating processes** This description encompasses the covariate shift, outcome shift and complex causal shift settings. We consider $X$ to be univariate, $Y$ to be binary, and $A$ to be binary, taking on a value of 0 or 1. We use a binary latent variable $U$ to encode the relationship between $X$ and $A$. For the covariate shift setting, we set $\mu_0 = -2$, $\mu_1 = 0$, $\gamma_A = 1$, $\beta_{a_0} = \beta_{a_1} = 0.5$, and $\alpha_{a_0} = \alpha_{a_1} = 0$. For the outcome shift setting, we set $\mu_0 = -2$, $\mu_1 = 0$, $\gamma_A = 0$, $\beta_{a_0} = 0.5$, $\beta_{a_1} = -1$, and $\alpha_{a_0} = 0.1$, $\alpha_{a_1} = 0$. For the complex causal shift setting, the settings are identical to the outcome shift case except that $\gamma_A = 1$, which has the effect of introducing a covariate shift. To verify properties in a setting where the subgroup covariates distributions are separable, we further increase the extent of the covariate shift present in the complex causal shift case by setting $\mu_1 = 2$.

$$
\begin{aligned}
U &\sim \text{Bernoulli}(0.5) \\
X \mid U = 0 &\sim \mathcal{N}(\mu_0, 1) \\
X \mid U = 1 &\sim \mathcal{N}(\mu_1, 1) \\
A \mid U &\sim \gamma_A U + (1 - \gamma_A) * \text{Bernoulli}(0.5) \\
Y \mid A = 0 &\sim \text{Bernoulli}\left(\text{logit}^{-1}\left(\beta_{a_0} x + \alpha_{a_0}\right)\right) \\
Y \mid A = 1 &\sim \text{Bernoulli}\left(\text{logit}^{-1}\left(\beta_{a_1} x + \alpha_{a_1}\right)\right)
\end{aligned}
$$

We construct three settings with selection bias through augmentation of the complex causal shift setting. We implement three settings, corresponding to cases where the selection mechanism depends on $X$, $Y$, or $\{Y, A\}$, correspond to $S_X$, $S_Y$, and $S_{YA}$. The selected dataset is constructed by filtering the data to cases where $S = 1$.

$$
\begin{aligned}
S_X &\sim \text{Bernoulli}(-\frac{4}{25}X^2 + 1) \\
S_Y &\sim \text{Bernoulli}(0.8Y + 0.4(1 - Y)) \\
S_{YA} \mid A = 0 &\sim \text{Bernoulli}(0.5Y + 0.8(1 - Y)) \\
S_{YA} \mid A = 1 &\sim \text{Bernoulli}(0.25Y + 0.8(1 - Y))
\end{aligned}
$$

**Anticausal data generating processes** This description encompasses the label shift, presentation shift, and complex anticausal shift settings. We consider $X$ to be univariate, $Y$ to be binary, and $A$ to be binary, taking on a value of 0 or 1. For simplicity, we define this data generating process as having $A$-dependent effects, rather than using a latent variable $U$. For the label shift case, we set $\pi_{Y_0} = 0.5$, $\pi_{Y_1} = 0.1$, $\mu_{A_0 Y_0} = -1$, $\mu_{A_0 Y_1} = 1$, $\mu_{A_1 Y_0} = -1$, $\mu_{A_1 Y_1} = 1$. For the presentation shift case, we set $\pi_{Y_0} = 0.5$, $\pi_{Y_1} = 0.5$, $\mu_{A_0 Y_0} = 1$, $\mu_{A_0 Y_1} = 0$, $\mu_{A_1 Y_0} = -1$, $\mu_{A_1 Y_1} = 1$. The complex anticausal shift setting uses the same parameters as the presentation shift setting except $\pi_{Y_0} = 0.5$, $\pi_{Y_1} = 0.1$.

$$
\begin{aligned}
A &\sim \text{Bernoulli}(0.5) \\
Y &\sim \text{Bernoulli}\left(A\pi_{Y_0} + (1 - A)\pi_{Y_1}\right) \\
X \mid A, Y &\sim \mathcal{N}(\mu_{AY}, 1)
\end{aligned}
$$

### A.4 Experiments with the American Community Survey (ACS) Public Use Microdata Sample (PUMS)

As described in the main text, we follow Ding et al. [50] to define prediction tasks from ACS PUMS [51]. We use the 'ACSIncome' and 'ACSPublicCoverage' tasks definitions provided by the folktables Python package version 0.0.12 [50]. For all experiments, we use the 5-Year horizon California

'person' files, encompassing census records from 2013-2018. The 'ACSIncome' task definition is to predict whether an individual's income is greater than $50,000$ per year for adults of age 18 years or older with an income of at least $100$ that have worked greater than zero hours in the past twelve months. The 'ACSPublicCoverage' task definition is to predict whether an individual is enrolled in a public health insurance plan, restricted to individuals of age 65 years or younger making less than $30,000$ per year. The standard covariates used for the 'ACSIncome' task are age (AGEP), class of worker (COW), educational attainment (SCHL), marital status (MAR), occupation (OCCP), place of birth (POBP), relationship (RELP), usual hours worked per week past 12 month (WKHP), and race (RAC1P). The standard covariates used for the 'ACSPublicCoverage' task are AGEP, SCHL, MAR, sex, disability (DIS), employment status of parents (ESP), citizenship status (CIT), mobility status (MIG), military service (MIL), ancestry (ANC), nativity (NAT), hearing difficulty (DEAR), vision difficulty (DEYE), cognitive difficulty (DREM), income, employment status (ESR), state (ST), gave birth to child within the past 12 months (FER), and RAC1P.

To define subgroups, we create a custom combined race and ethnicity field that combines the RA1CP field with the hispanic origin flag (HISP) and groups rare categories. The combined race/ethnicity field takes on the value "Hispanic" for individuals of Hispanic origin and the value of RA1CP field otherwise. We then combine the American Indian and Alaska Native with the Native Hawaiian and Pacific Islander into a group called "Other". For modeling, we remove RAC1P from the covariate set, and use the combined race/ethnicity field for subgroup-aware prediction.

For modeling and evaluation, we replicate the procedure used in the simulation study, using gradient boosting trees for classification with the same cross-validation and hyperparameter selection procedure. As is standard for the prediction tasks proposed by Ding et al. [50], we do not directly use the person weights provided in the ACS PUMS data, which implies that the derived estimates and models may not be representative of the underlying populations.

As in the case of the simulation study, we conduct these experiments using machines with 32 CPUs and 32 GB of RAM. Computation of the bootstrap confidence intervals was the most significant contributor to the overall run time, taking approximately 14 hours per setting (*i.e.*, approximately 28 hours for the two tasks considered).

# B Supplementary figures and tables

**Supplementary Table B1: Conditional independence properties of Bayes-optimal models**. ✓ indicates conditions where Bayes-optimal prediction is a sufficient condition for the listed criteria. ✗ indicates that Bayes-optimal prediction is not a sufficient condition for the property. $f^*(Z)$ corresponds to the Bayes-optimal predictor that depends on $Z$.

| | Sufficiency $(Y \perp A \mid f^*(Z))$ | | Separation $(f^*(Z) \perp A \mid Y)$ | |
| --- | --- | --- | --- | --- |
| Setting | $Z = X$ | $Z = \{X, A\}$ | $Z = X$ | $Z = \{X, A\}$ |
| Covariate Shift | ✓ | ✓ | ✗ | ✗ |
| Outcome Shift | ✗ | ✓ | ✗ | ✗ |
| Causal Complex Shift | ✗ | ✓ | ✗ | ✗ |
| Label Shift | ✗ | ✓ | ✓ | ✗ |
| Presentation Shift | ✗ | ✓ | ✗ | ✗ |
| Anticausal Complex Shift | ✗ | ✓ | ✗ | ✗ |

**Supplementary Table B2: Properties of Bayes-optimal models under selection**. ✓ indicates cases where Bayes-optimal prediction in the selected population $P(\cdot \mid S = 1)$ is sufficient to induce the listed property in the full population $P(\cdot)$. "Other $Y \not\perp A \mid X$" indicate the outcome shift, presentation shift, and complex causal and anticausal shift graphs.

| | Sufficiency | | Subgroup calibration | | Separation | |
| --- | --- | --- | --- | --- | --- | --- |
| Setting | $Z = X$ | $Z = \{X, A\}$ | $Z = X$ | $Z = \{X, A\}$ | $Z = X$ | $Z = \{X, A\}$ |
| **Covariate shift** | | | | | | |
| $X \to S$ | ✓ | ✓ | ✓ | ✓ | ✗ | ✗ |
| $A \to S$ | ✓ | ✓ | ✓ | ✓ | ✗ | ✗ |
| $\{X, A\} \to S$ | ✓ | ✓ | ✓ | ✓ | ✗ | ✗ |
| $Y \to S$ | ✓ | ✓ | ✗ | ✗ | ✗ | ✗ |
| $\{X, Y\} \to S$ | ✗ | ✗ | ✗ | ✗ | ✗ | ✗ |
| $\{A, Y\} \to S$ | ✗ | ✗ | ✗ | ✗ | ✗ | ✗ |
| $\{X, Y, A\} \to S$ | ✗ | ✗ | ✗ | ✗ | ✗ | ✗ |
| **Label shift** | | | | | | |
| $X \to S$ | ✗ | ✓ | ✗ | ✓ | ✓ | ✗ |
| $A \to S$ | ✗ | ✓ | ✗ | ✓ | ✗ | ✗ |
| $\{X, A\} \to S$ | ✗ | ✓ | ✗ | ✓ | ✗ | ✗ |
| $Y \to S$ | ✗ | ✓ | ✗ | ✗ | ✓ | ✗ |
| $\{X, Y\} \to S$ | ✗ | ✗ | ✗ | ✗ | ✓ | ✗ |
| $\{A, Y\} \to S$ | ✗ | ✗ | ✗ | ✗ | ✗ | ✗ |
| $\{X, Y, A\} \to S$ | ✗ | ✗ | ✗ | ✗ | ✗ | ✗ |
| **Other $Y \not\perp A \mid X$** | | | | | | |
| $X \to S$ | ✗ | ✓ | ✗ | ✓ | ✗ | ✗ |
| $A \to S$ | ✗ | ✓ | ✗ | ✓ | ✗ | ✗ |
| $\{X, A\} \to S$ | ✗ | ✓ | ✗ | ✓ | ✗ | ✗ |
| $Y \to S$ | ✗ | ✓ | ✗ | ✗ | ✗ | ✗ |
| $\{X, Y\} \to S$ | ✗ | ✗ | ✗ | ✗ | ✗ | ✗ |
| $\{A, Y\} \to S$ | ✗ | ✗ | ✗ | ✗ | ✗ | ✗ |
| $\{X, Y, A\} \to S$ | ✗ | ✗ | ✗ | ✗ | ✗ | ✗ |

**Supplementary Table B3: Implications of findings of controlled evaluations**. Each row corresponds to a different setting of the control variable, covariates, test result, and assumption of Bayes-optimality. Results assume that the weight model $P(A = a \mid V)$ is correct.

| Control Variable | Covariates | Reject($T_a = 0$) | Bayes-optimal | Implications |
|---|---|---|---|---|
| $X$ | $X$ | Yes | Yes | • Subgroup-dependent error structure conditioned on $X$ is present.
• Evidence that data is not generated under covariate shift.
• Sufficiency violation may be present.
• Subgroup performances differences not explained by covariate shift.
• Possible to improve performance with subgroup-aware prediction.
• Improving prediction on the basis of $X$ alone is not possible. |
| $X$ | $X$ | Yes | No | • Same as the Bayes-optimal case, with the exception that it may be possible to improve prediction on the basis of $X$. |
| $X$ | $X$ | No | Yes | • Results consistent with (*i.e.*, cannot reject) the hypothesis of no subgroup-dependent error structure conditioned on $X$.
• Results consistent with (*i.e.*, cannot reject) the hypothesis that data generated under covariate shift.
• Results consistent with (*i.e.*, cannot reject) the hypothesis that all performance differences explained by covariate shift. |
| $X$ | $X$ | No | No | • Results consistent with (*i.e.*, cannot reject) the hypothesis of no subgroup-dependent error structure conditioned on $X$.
• Results consistent with (*i.e.*, cannot reject) the hypothesis that data generated under covariate shift.
• Cannot rule out the hypothesis that performance differences explained by systematic underperformance for some values of $X$. |

| Control Var | Covariates | Reject($T_a = 0$) | Bayes-optimal | Implications |
|---|---|---|---|---|
| $X$ | $\{X, A\}$ | Yes | Yes | • Subgroup-dependent error structure conditioned on $X$ is present.
• Evidence that data is not generated under covariate shift.
• Subgroup performances differences not explained by covariate shift. |
| $X$ | $\{X, A\}$ | Yes | No | • Subgroup-dependent error structure conditioned on $X$ is present. |
| $X$ | $\{X, A\}$ | No | Yes | • Results consistent with (*i.e.*, cannot reject) the hypothesis of no subgroup-dependent error structure conditioned on $X$.
• Results consistent with (*i.e.*, cannot reject) the hypothesis that data generated under covariate shift.
• Results consistent with (*i.e.*, cannot reject) the hypothesis that all performance differences explained by covariate shift. |
| $X$ | $\{X, A\}$ | No | No | • Results consistent with (*i.e.*, cannot reject) the hypothesis of no subgroup-dependent error structure conditioned on $X$. |
| $Y$ | $X$ | Yes | Yes | • Subgroup-dependent error structure conditioned on $Y$ is present.
• Evidence that data is not generated under label shift. |
| $Y$ | $X$ | Yes | No | • Same as the Bayes-optimal case. |

*Continued on next page*

| Control Var | Covariates | Reject($T_a = 0$) | Bayes-optimal | Implications |
|---|---|---|---|---|
| $Y$ | $X$ | No | Yes | • Results consistent with (*i.e.*, cannot reject) the hypothesis of no subgroup-dependent error structure conditioned on $Y$. 
 • Results consistent with (*i.e.*, cannot reject) the hypothesis that data generated under label shift. 
 • Results consistent with (*i.e.*, cannot reject) the hypothesis that the separation and equalized odds criteria hold. 
 • If label shift holds, properties implied by covariate shift violation may hold (*i.e.*, sufficiency violation and informativeness of subgroup membership). |
| $Y$ | $X$ | No | No | • Same as the Bayes-optimal case. |
| $Y$ | $\{X, A\}$ | Yes | Yes | • Subgroup-dependent error structure conditioned on $Y$ is present. |
| $Y$ | $\{X, A\}$ | Yes | No | • Same as the Bayes-optimal case. |
| $Y$ | $\{X, A\}$ | No | Yes | • Cannot reject hypothesis of no subgroup-dependent error structure conditioned on $Y$. |
| $Y$ | $\{X, A\}$ | No | No | • Same as the Bayes-optimal case. |
| $R$ | $X$ | Yes | Yes | • Evidence of sufficiency violation. 
 • Subgroup-aware prediction likely to improve performance. 
 • Evidence that data not generated under covariate shift. |
| $R$ | $X$ | Yes | No | • Evidence of sufficiency violation. 
 • Subgroup-aware prediction likely to improve performance. 
 • Evidence that data either not generated under covariate shift or that model underperforms for subgroups. |

| Control Var | Covariates | Reject($T_a = 0$) | Bayes-optimal | Implications |
|---|---|---|---|---|
| $R$ | $X$ | No | Yes | • Results consistent with (*i.e.*, cannot reject) the hypothesis that sufficiency is satisfied.
• Results consistent with (*i.e.*, cannot reject) the hypothesis that data generated under covariate shift. |
| $R$ | $X$ | No | No | • Results consistent with (*i.e.*, cannot reject) the hypothesis that sufficiency is satisfied. |
| $R$ | $\{X, A\}$ | Yes | Yes | • This state should not be possible since subgroup Bayes-optimality implies $T_a = 0$ under control for $R$. |
| $R$ | $\{X, A\}$ | Yes | No | • Evidence of sufficiency violation.
• Indicates that the model could be improved for one or more subgroups. |
| $R$ | $\{X, A\}$ | No | Yes | • Results consistent with (*i.e.*, cannot reject) the hypothesis that sufficiency is satisfied. |
| $R$ | $\{X, A\}$ | No | No | • Same as the Bayes-optimal case. |

**Supplementary Table B4: ACS PUMS cohort characteristics.** Shown are the number of individuals and the prevalence of the label for each race/ethnicity subgroup for the two tasks derived from the ACS PUMS data.

| | ACSIncome | | ACSPublicCoverage | |
|---|---|---|---|---|
| | Count | Prevalence | Count | Prevalence |
| Asian | 151,163 | 0.449 | 104,642 | 0.286 |
| Black | 40,764 | 0.327 | 41,090 | 0.507 |
| Hispanic | 313,007 | 0.200 | 317,534 | 0.393 |
| Multiracial | 23,781 | 0.374 | 20,734 | 0.323 |
| Other | 8,910 | 0.304 | 8,590 | 0.397 |
| White | 412,572 | 0.504 | 236,842 | 0.299 |

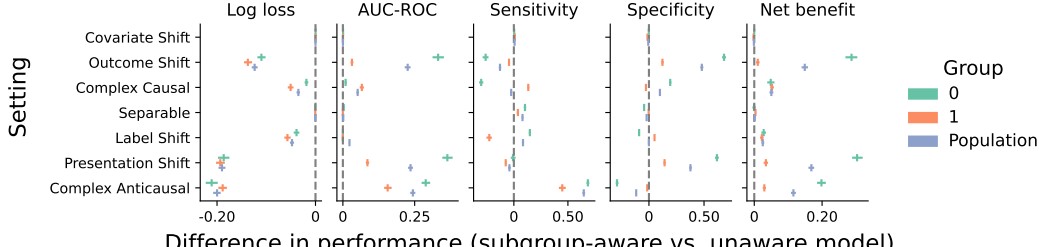

**Supplementary Figure B1: Simulation study: the effect of subgroup-aware prediction on model performance**. We report the difference in performance between models that have access to subgroup membership as an additional covariate as compared to those that do not. Plotted are average differences with 95% confidence intervals for each setting and for several performance metrics.

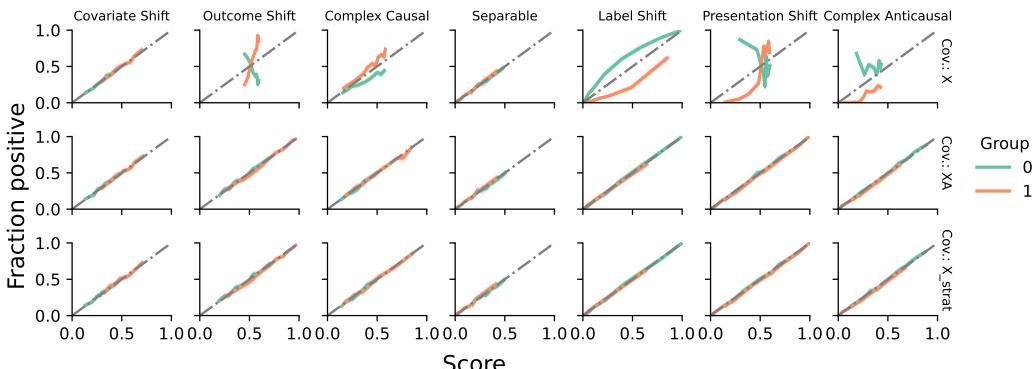

**Supplementary Figure B2: Simulation study: calibration curves**. Plotted are calibration curves for each subgroup with 95% confidence intervals. The first row corresponds to subgroup-agnostic prediction, the second row to prediction with $A$ as an additional covariate, and the third row to stratified prediction by $A$.

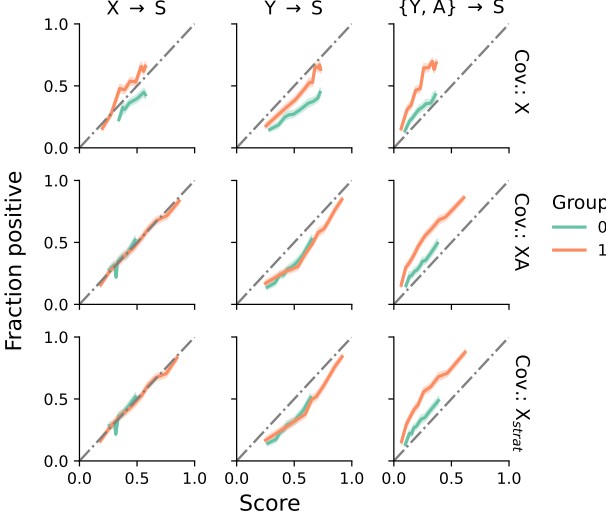

**Supplementary Figure B3: Simulation study: calibration with selection bias**. Plotted are calibration curves for each subgroup with 95% confidence intervals. Models are fit in the selected population and evaluated in the full population without selection. The first row corresponds to subgroup-agnostic prediction, the second row to prediction with $A$ as an additional covariate, and the third row to stratified prediction by $A$.

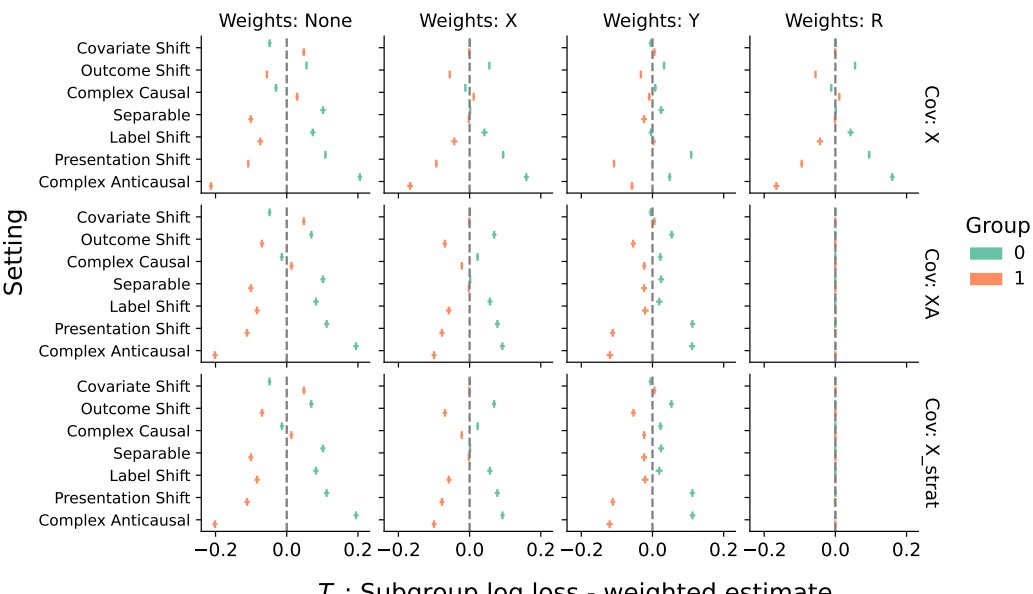

**Supplementary Figure B4: Simulation study: controlled evaluation of log loss.** Plotted are the statistics $T_a$ with 95% confidence intervals, corresponding to differences between the unweighted disaggregated performance with the population performance weighted to match the distribution of $X$, $Y$, or $R$ on the subgroups. The first row corresponds to subgroup-agnostic prediction, the second row to prediction with $A$ as an additional covariate, and the third row to stratified prediction by $A$.

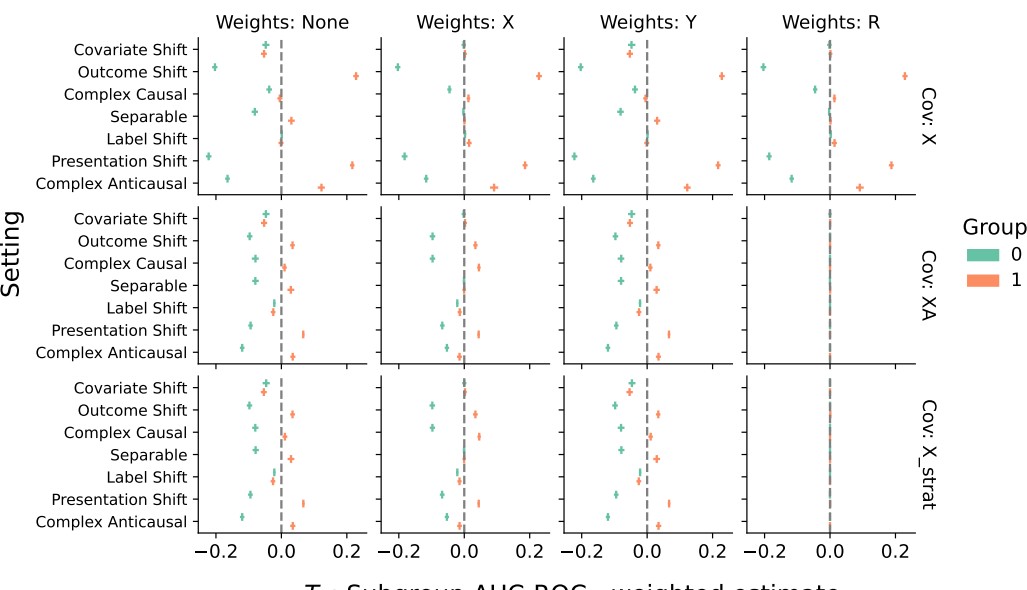

**Supplementary Figure B5: Simulation study: controlled evaluation of AUC-ROC.** Plotted are the statistics $T_a$ with 95% confidence intervals, corresponding to differences between the unweighted disaggregated performance with the population performance weighted to match the distribution of $X$, $Y$, or $R$ on the subgroups. The first row corresponds to subgroup-agnostic prediction, the second row to prediction with $A$ as an additional covariate, and the third row to stratified prediction by $A$.

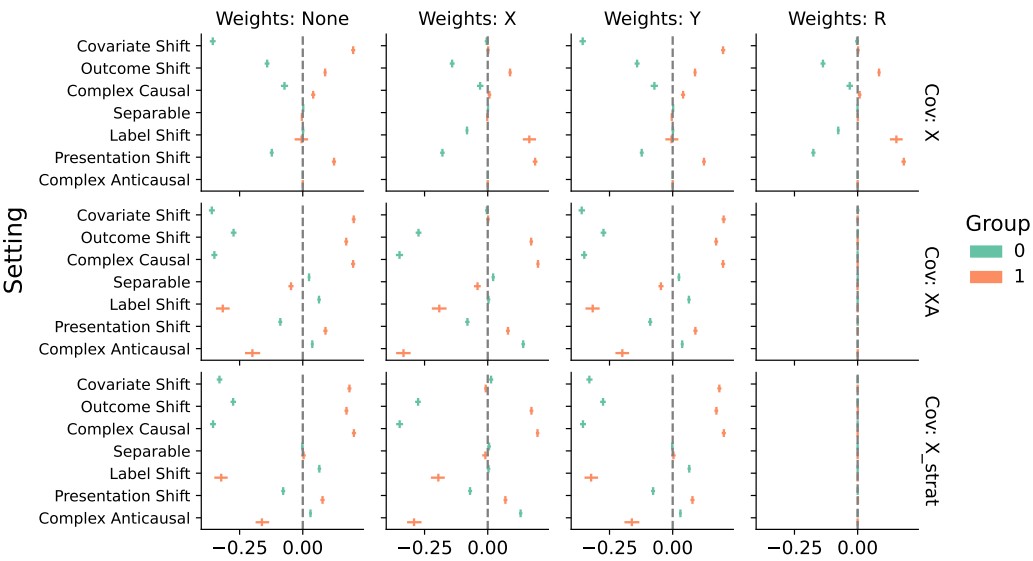

**Supplementary Figure B6: Simulation study: controlled evaluation of sensitivity**. Plotted are the statistics $T_a$ with 95% confidence intervals, corresponding to differences between the unweighted disaggregated performance with the population performance weighted to match the distribution of $X$, $Y$, or $R$ on the subgroups. The first row corresponds to subgroup-agnostic prediction, the second row to prediction with $A$ as an additional covariate, and the third row to stratified prediction by $A$.

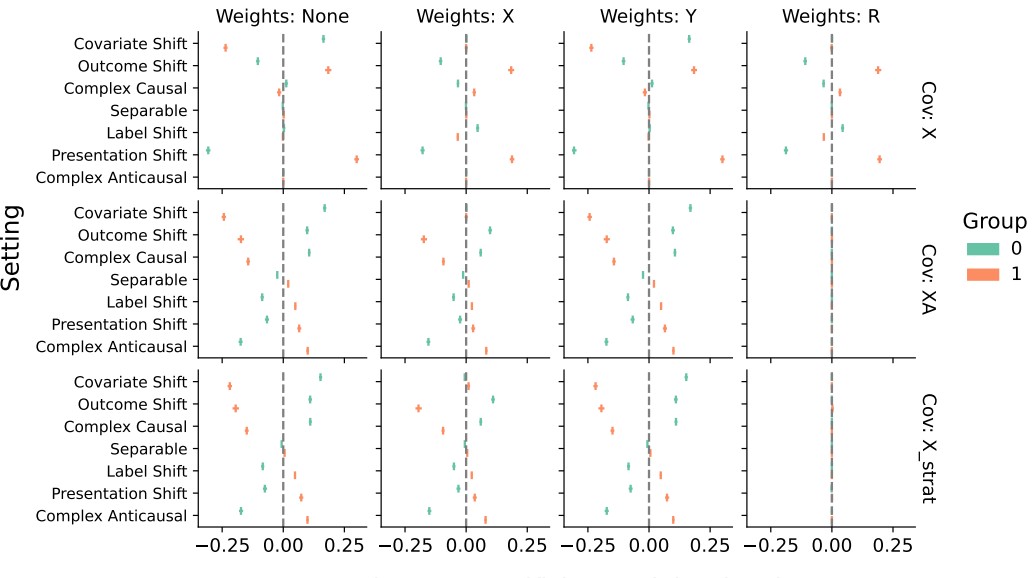

**Supplementary Figure B7: Simulation study: controlled evaluation of specificity**. Plotted are the statistics $T_a$ with 95% confidence intervals, corresponding to differences between the unweighted disaggregated performance with the population performance weighted to match the distribution of $X$, $Y$, or $R$ on the subgroups. The first row corresponds to subgroup-agnostic prediction, the second row to prediction with $A$ as an additional covariate, and the third row to stratified prediction by $A$.

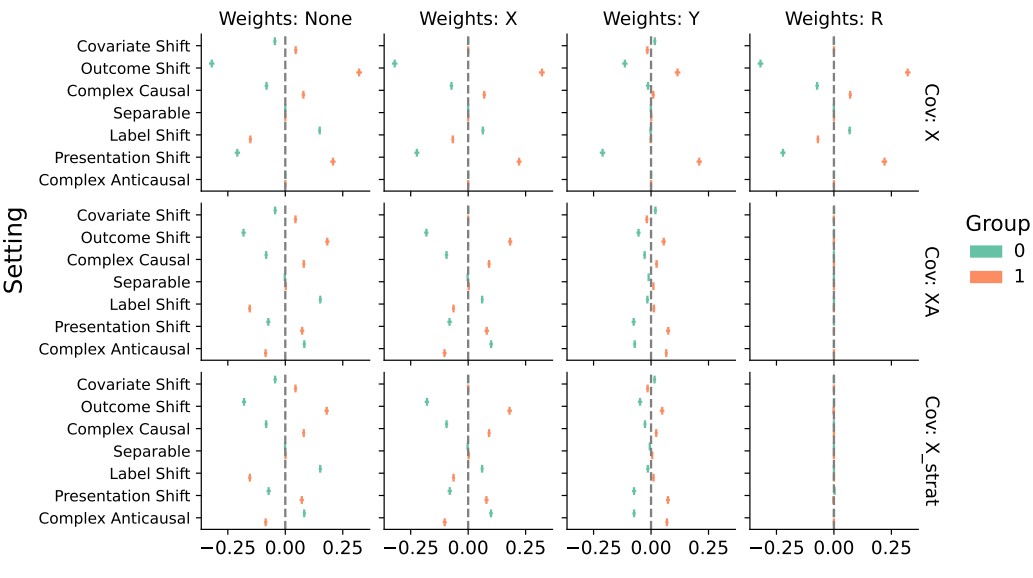

**Supplementary Figure B8: Simulation study: controlled evaluation of net benefit**. Plotted are the statistics $T_a$ with 95% confidence intervals, corresponding to differences between the unweighted disaggregated performance with the population performance weighted to match the distribution of $X$, $Y$, or $R$ on the subgroups. The first row corresponds to subgroup-agnostic prediction, the second row to prediction with $A$ as an additional covariate, and the third row to stratified prediction by $A$.

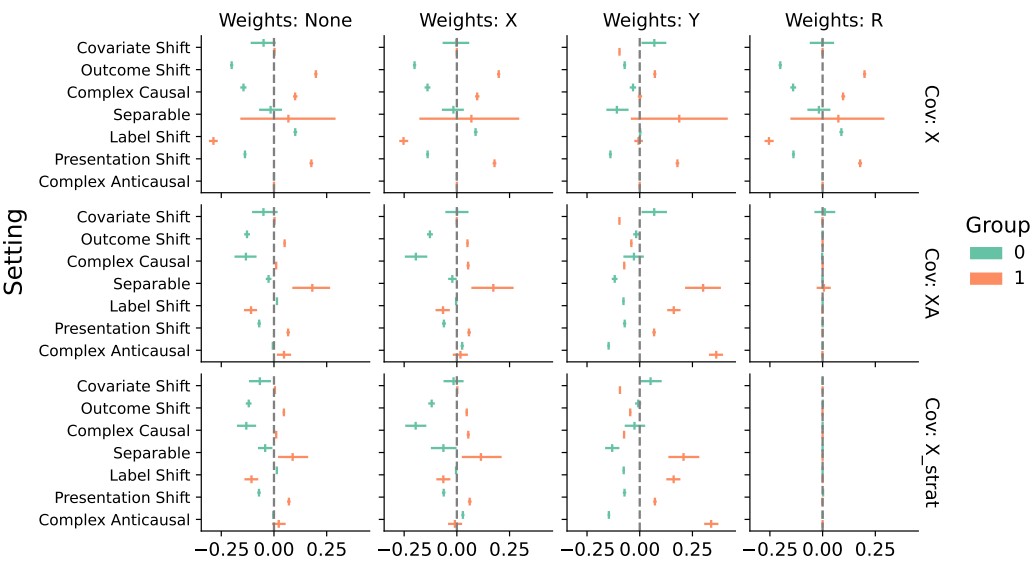

**Supplementary Figure B9: Simulation study: controlled evaluation of precision**. Plotted are the statistics $T_a$ with 95% confidence intervals, corresponding to differences between the unweighted disaggregated performance with the population performance weighted to match the distribution of $X$, $Y$, or $R$ on the subgroups. The first row corresponds to subgroup-agnostic prediction, the second row to prediction with $A$ as an additional covariate, and the third row to stratified prediction by $A$.

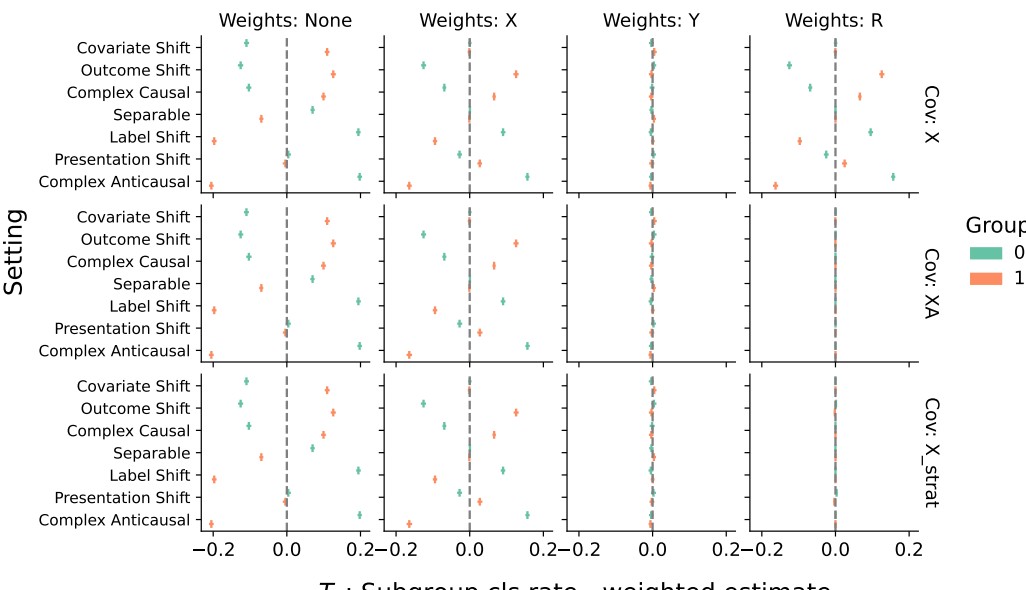

**Supplementary Figure B10: Simulation study: controlled evaluation of classification rate**. Plotted are the statistics $T_a$ with 95% confidence intervals, corresponding to differences between the unweighted disaggregated performance with the population performance weighted to match the distribution of $X$, $Y$, or $R$ on the subgroups. The first row corresponds to subgroup-agnostic prediction, the second row to prediction with $A$ as an additional covariate, and the third row to stratified prediction by $A$.

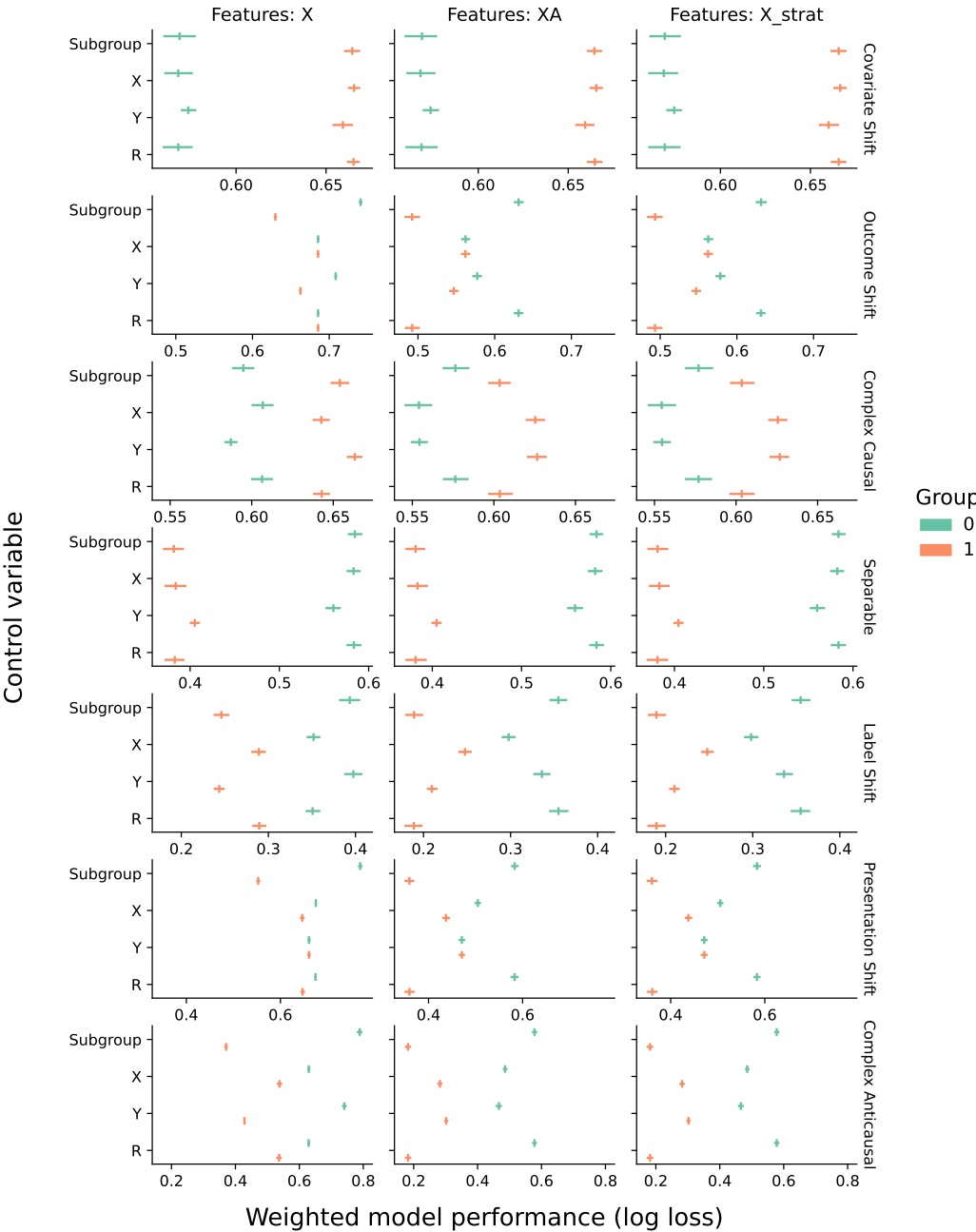

**Supplementary Figure B11: Simulation study: weighted estimation of log loss**. Plotted are the weighted estimates of performance $M_a$ with 95% confidence intervals, corresponding to weighted estimates of population performance weighted to match the distribution of $X$, $Y$, or $R$ for each subgroups. The entry labeled "subgroup" corresponds to the unweighted estimate of subgroup performance. The first column corresponds to subgroup-agnostic prediction, the second column to prediction with $A$ as an additional covariate, and the third column to stratified prediction by $A$.

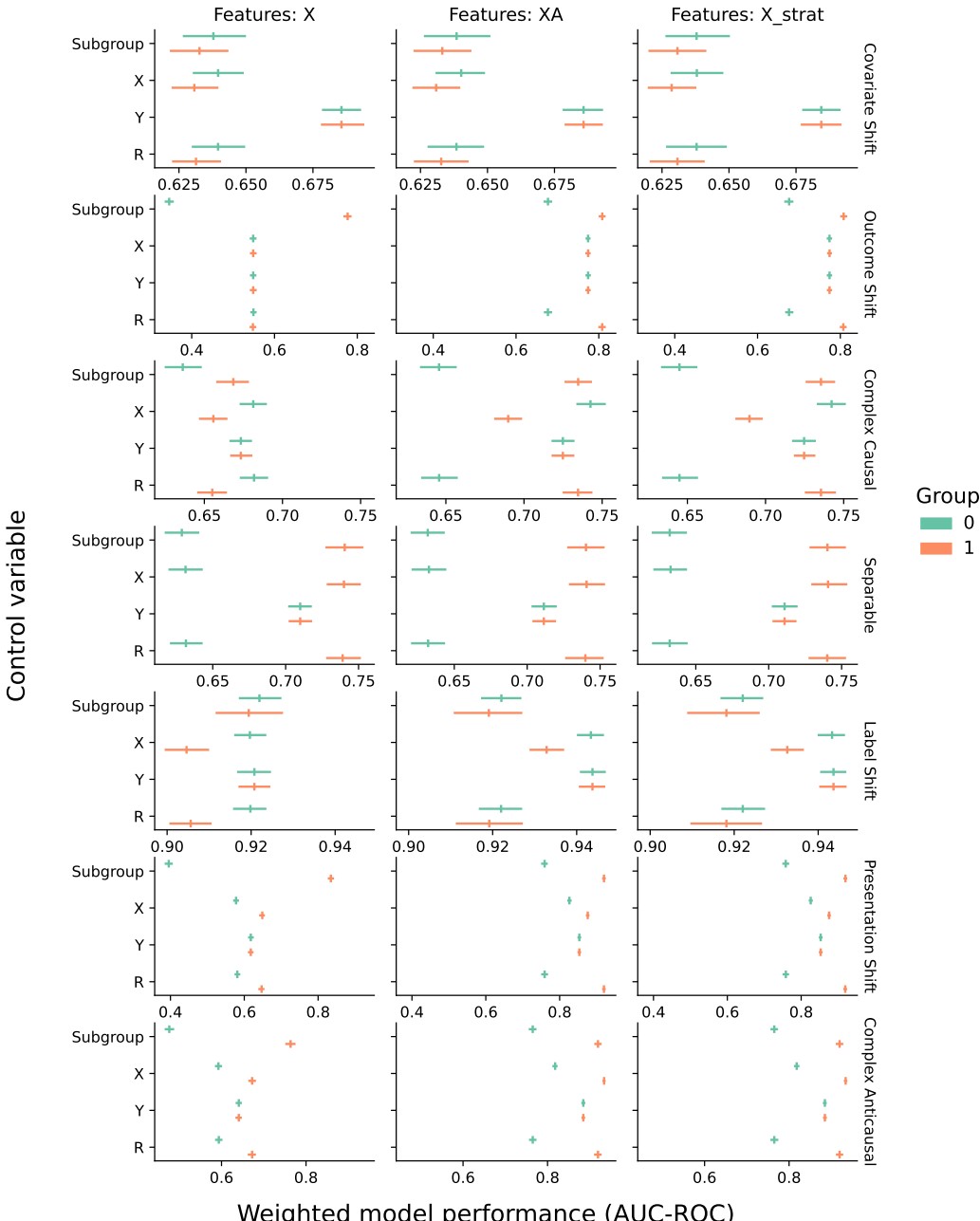

**Supplementary Figure B12: Simulation study: weighted estimation of AUC-ROC**. Plotted are the weighted estimates of performance $M_a$ with 95% confidence intervals, corresponding to weighted estimates of population performance weighted to match the distribution of $X$, $Y$, or $R$ for each subgroups. The entry labeled "subgroup" corresponds to the unweighted estimate of subgroup performance. The first column corresponds to subgroup-agnostic prediction, the second column to prediction with $A$ as an additional covariate, and the third column to stratified prediction by $A$.

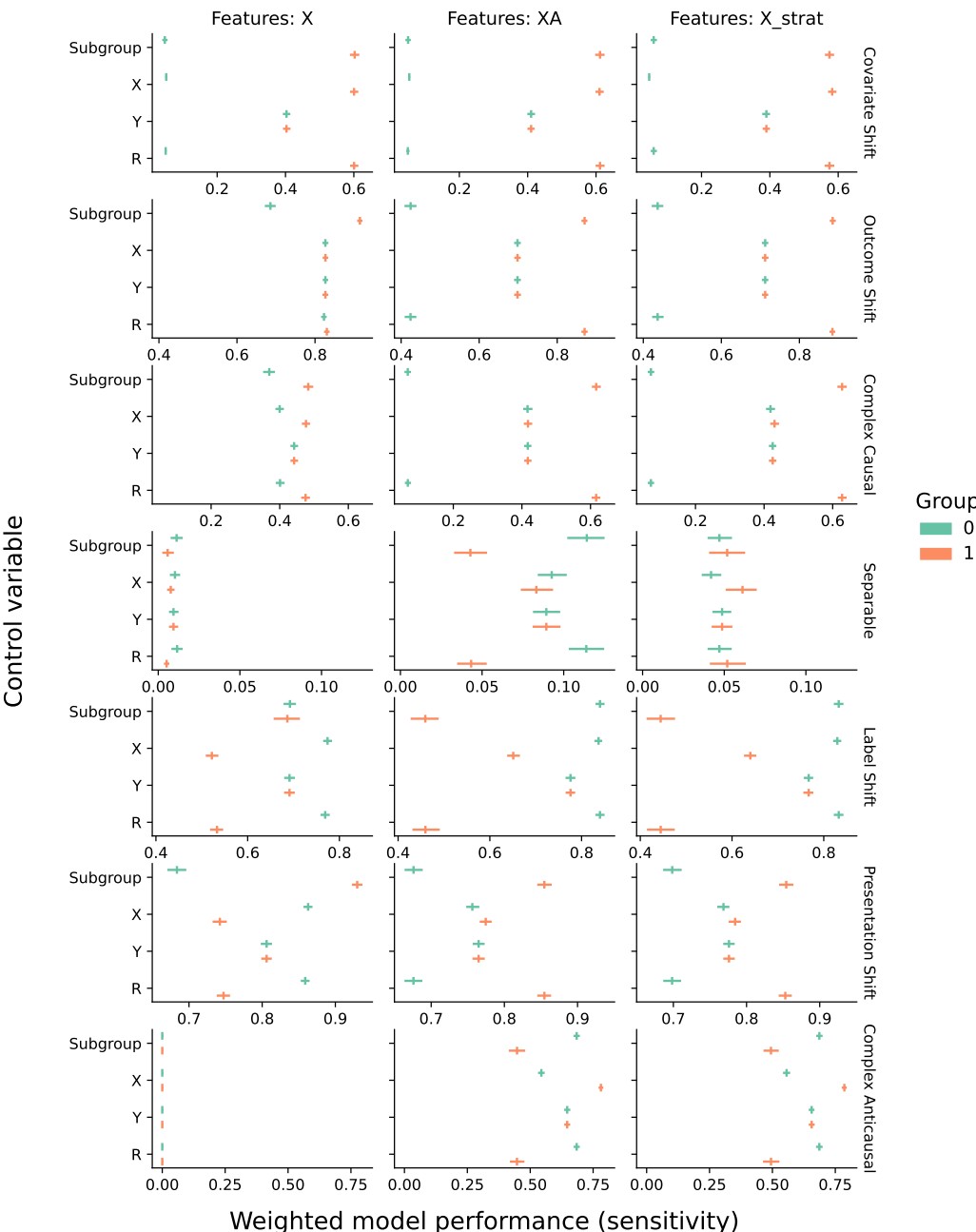

**Supplementary Figure B13: Simulation study: weighted estimation of sensitivity.** Plotted are the weighted estimates of performance $M_a$ with 95% confidence intervals, corresponding to weighted estimates of population performance weighted to match the distribution of $X$, $Y$, or $R$ for each subgroups. The entry labeled "subgroup" corresponds to the unweighted estimate of subgroup performance. The first column corresponds to subgroup-agnostic prediction, the second column to prediction with $A$ as an additional covariate, and the third column to stratified prediction by $A$.

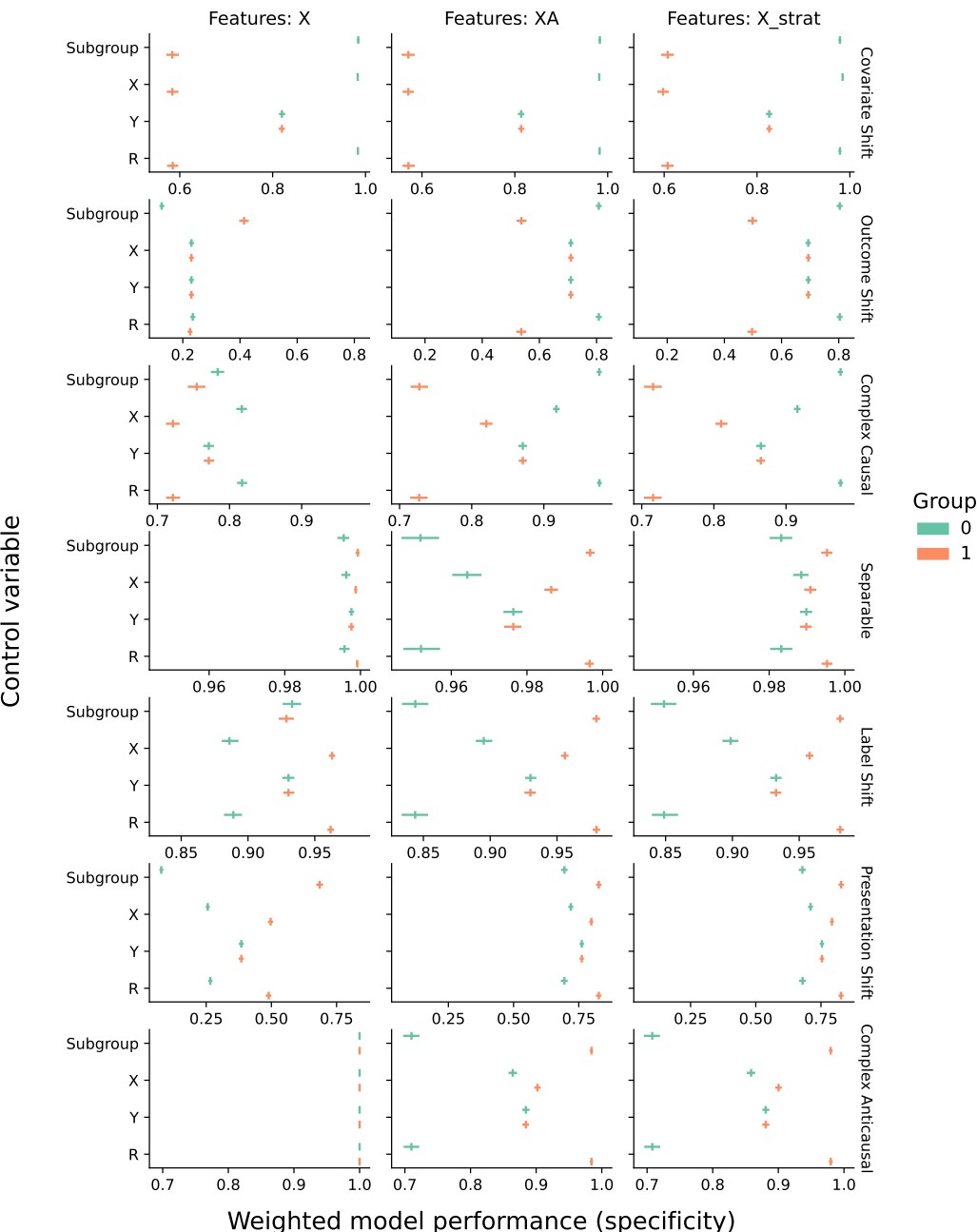

**Supplementary Figure B14: Simulation study: weighted estimation of specificity**. Plotted are the weighted estimates of performance $M_a$ with 95% confidence intervals, corresponding to weighted estimates of population performance weighted to match the distribution of $X$, $Y$, or $R$ for each subgroups. The entry labeled "subgroup" corresponds to the unweighted estimate of subgroup performance. The first column corresponds to subgroup-agnostic prediction, the second column to prediction with $A$ as an additional covariate, and the third column to stratified prediction by $A$.

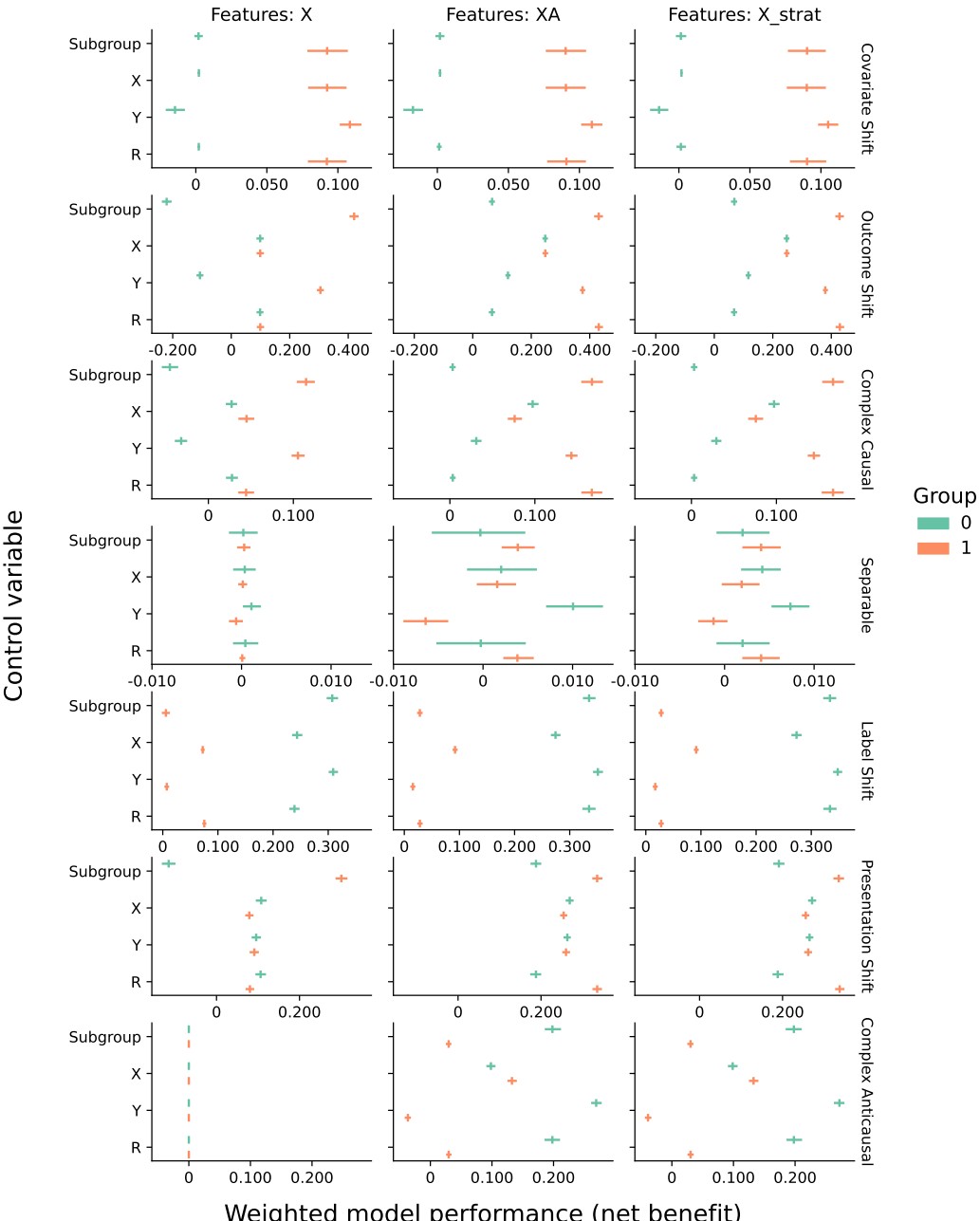

**Supplementary Figure B15: Simulation study: weighted estimation of net benefit**. Plotted are the weighted estimates of performance $M_a$ with 95% confidence intervals, corresponding to weighted estimates of population performance weighted to match the distribution of $X$, $Y$, or $R$ for each subgroups. The entry labeled "subgroup" corresponds to the unweighted estimate of subgroup performance. The first column corresponds to subgroup-agnostic prediction, the second column to prediction with $A$ as an additional covariate, and the third column to stratified prediction by $A$.

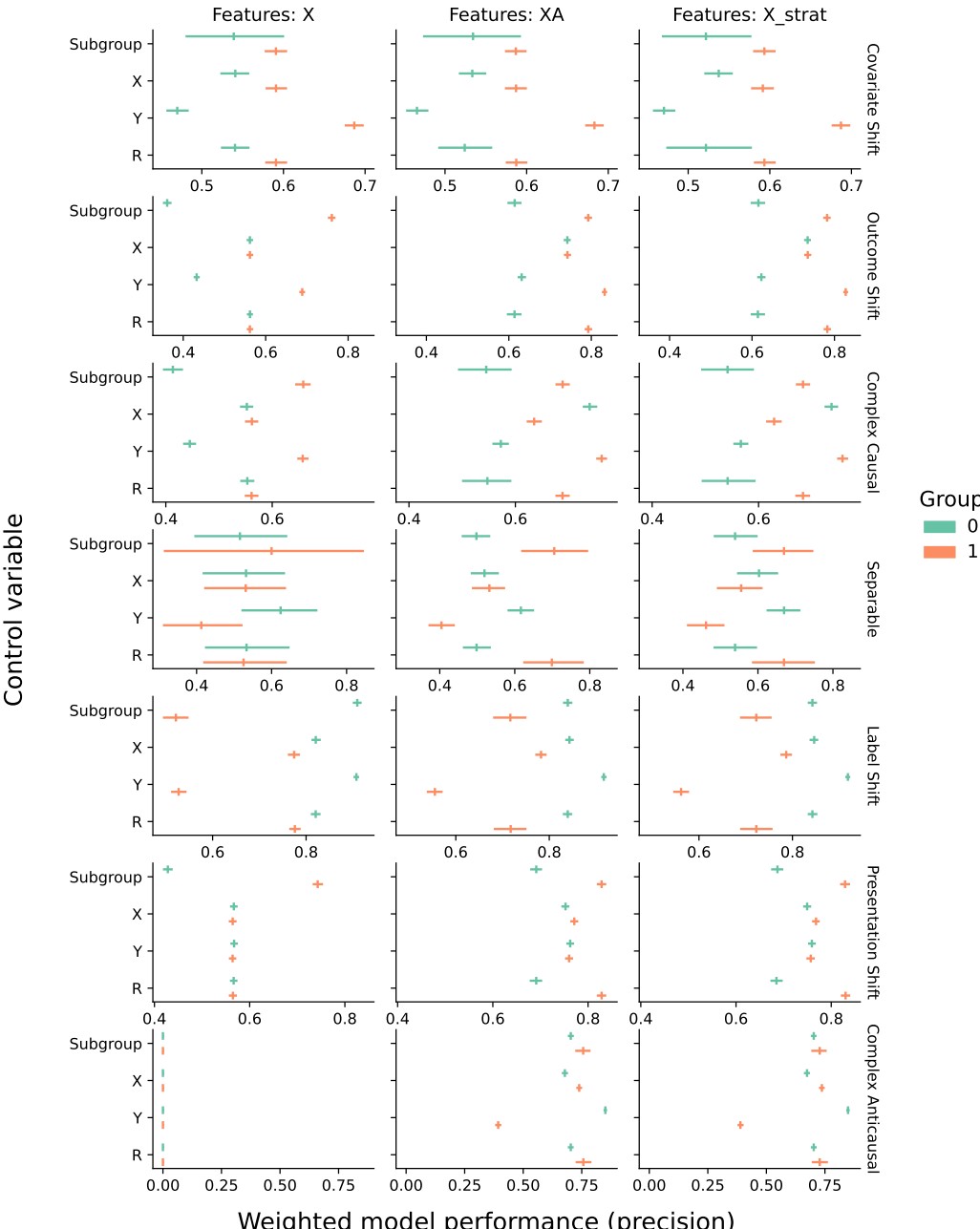

**Supplementary Figure B16: Simulation study: weighted estimation of precision**. Plotted are the weighted estimates of performance $M_a$ with 95% confidence intervals, corresponding to weighted estimates of population performance weighted to match the distribution of $X$, $Y$, or $R$ for each subgroups. The entry labeled "subgroup" corresponds to the unweighted estimate of subgroup performance. The first column corresponds to subgroup-agnostic prediction, the second column to prediction with $A$ as an additional covariate, and the third column to stratified prediction by $A$.

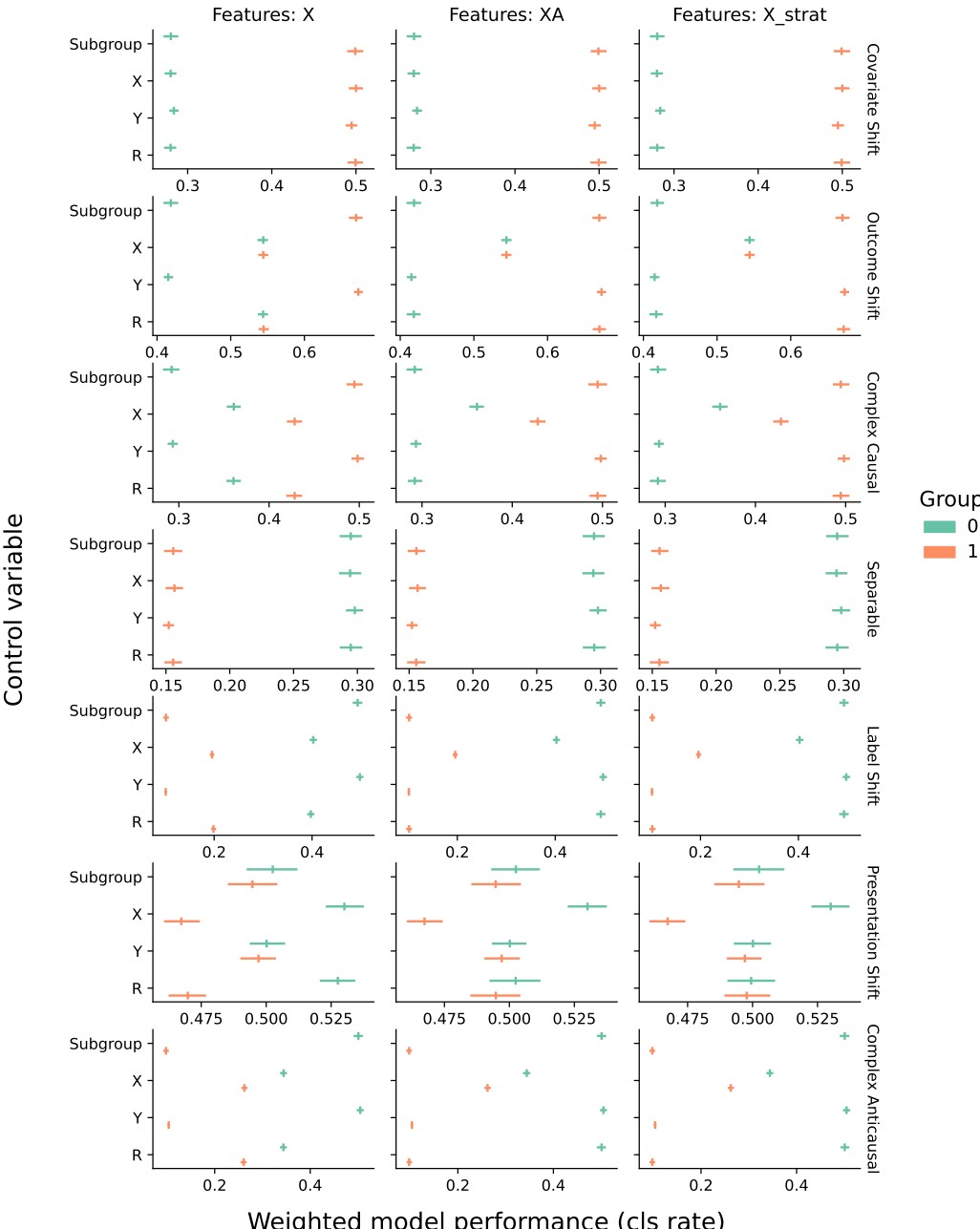

**Supplementary Figure B17: Simulation study: weighted estimation of classification rate**. Plotted are the weighted estimates of performance $M_a$ with 95% confidence intervals, corresponding to weighted estimates of population performance weighted to match the distribution of $X$, $Y$, or $R$ for each subgroups. The entry labeled "subgroup" corresponds to the unweighted estimate of subgroup performance. The first column corresponds to subgroup-agnostic prediction, the second column to prediction with $A$ as an additional covariate, and the third column to stratified prediction by $A$.

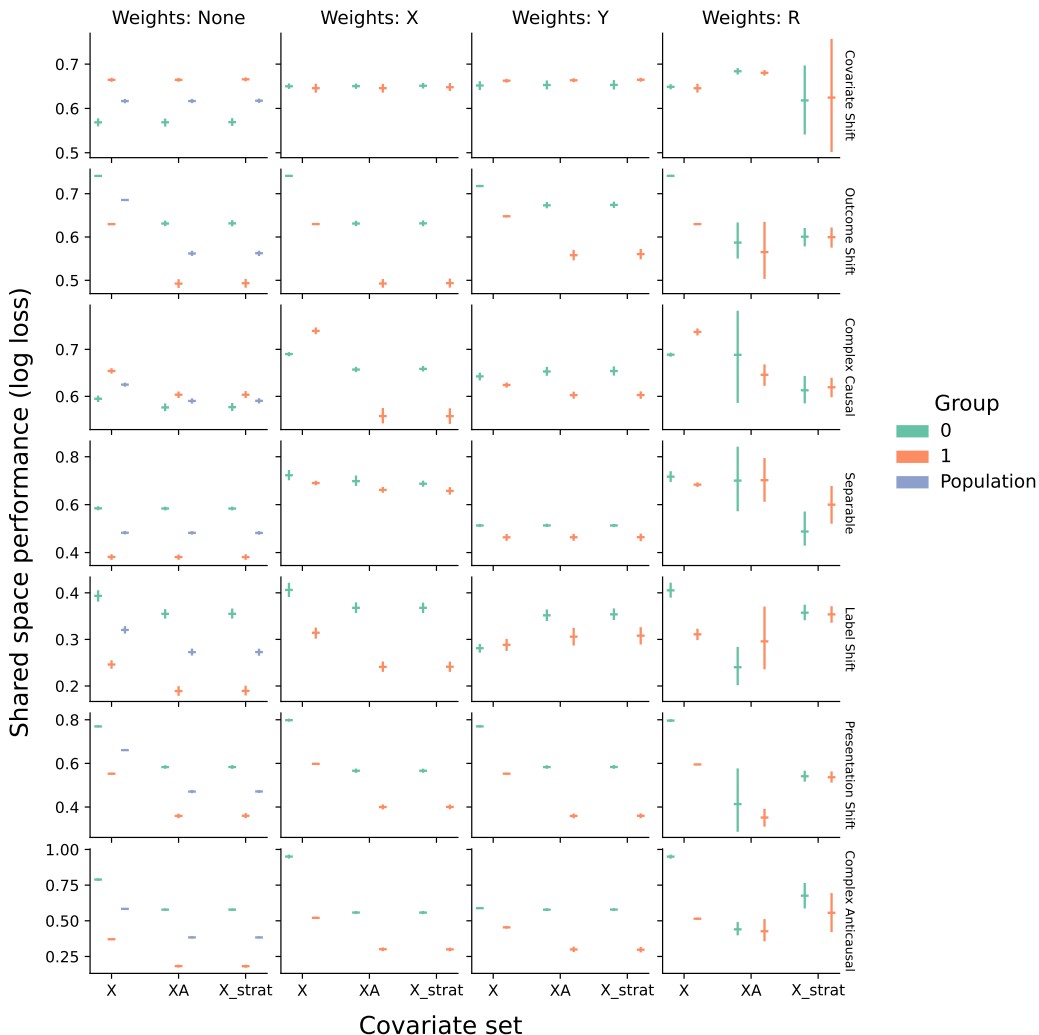

**Supplementary Figure B18: Simulation study: shared space subgroup log loss**. Plotted are the average performance with 95% confidence intervals for subgroup performance following weighting to a shared space, using the approach of Cai et al. [16]. Columns correspond to different conditioning variables used to construct the weights and rows correspond data generating processes.

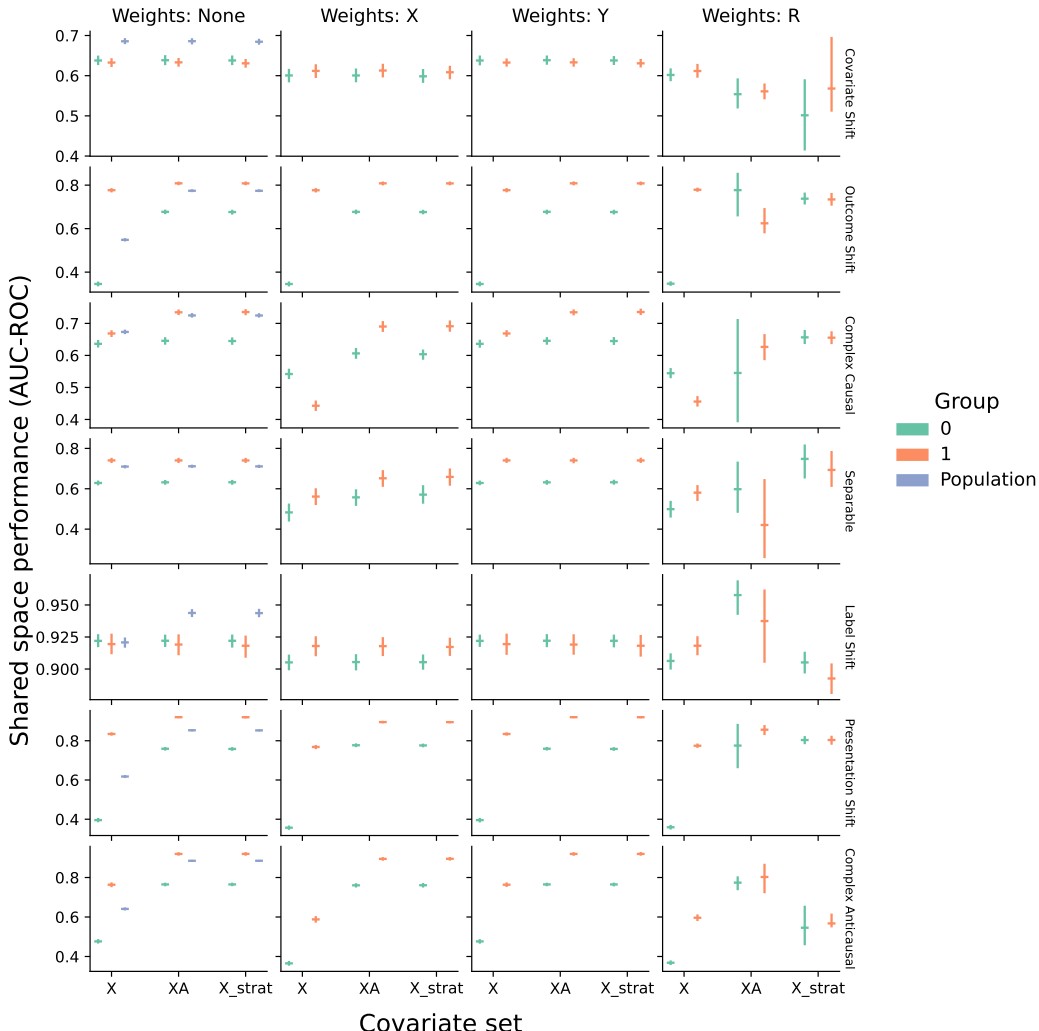

**Supplementary Figure B19: Simulation study: shared space subgroup AUC-ROC**. Plotted are the average performance with 95% confidence intervals for subgroup performance following weighting to a shared space, using the approach of Cai et al. [16]. Columns correspond to different conditioning variables used to construct the weights and rows correspond data generating processes.

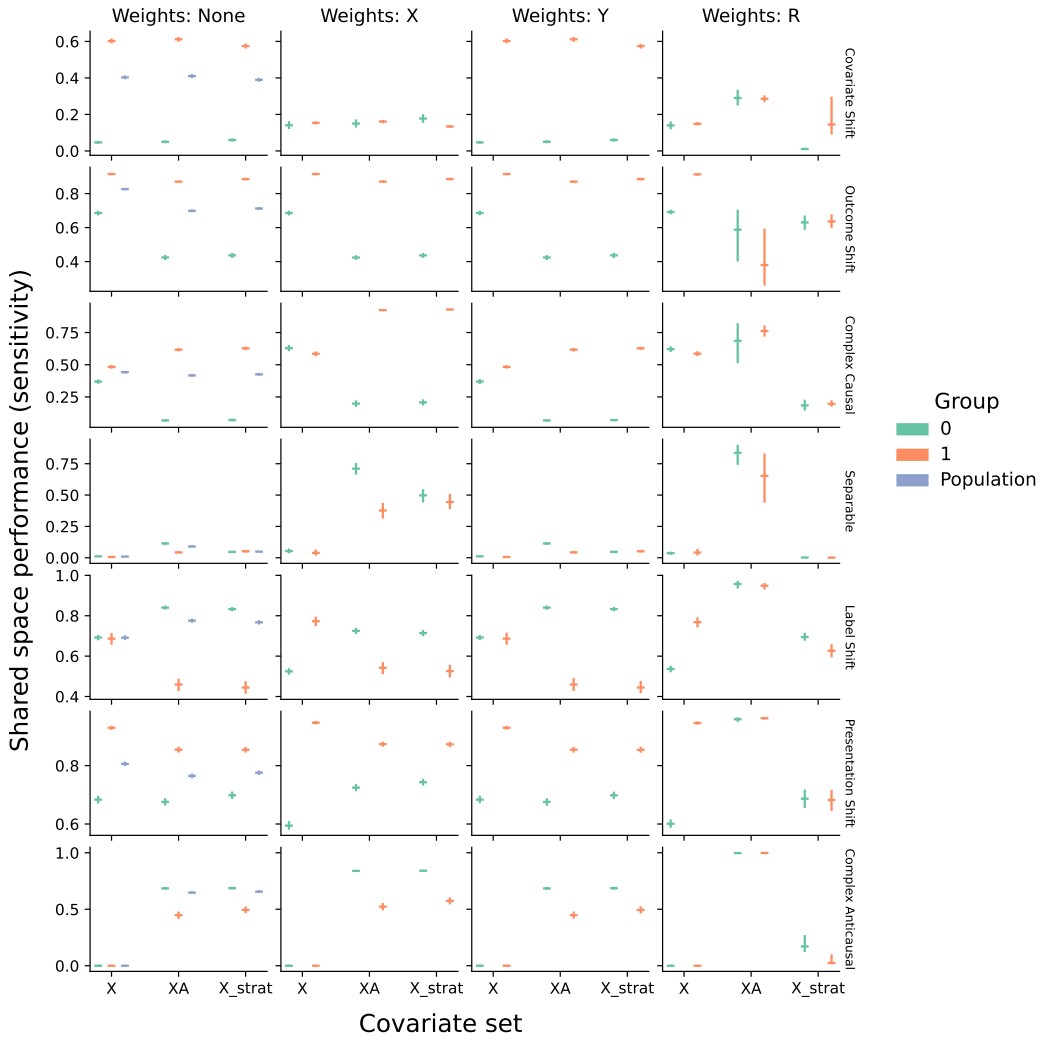

**Supplementary Figure B20: Simulation study: shared space subgroup sensitivity**. Plotted are the average performance with 95% confidence intervals for subgroup performance following weighting to a shared space, using the approach of Cai et al. [16]. Columns correspond to different conditioning variables used to construct the weights and rows correspond data generating processes.

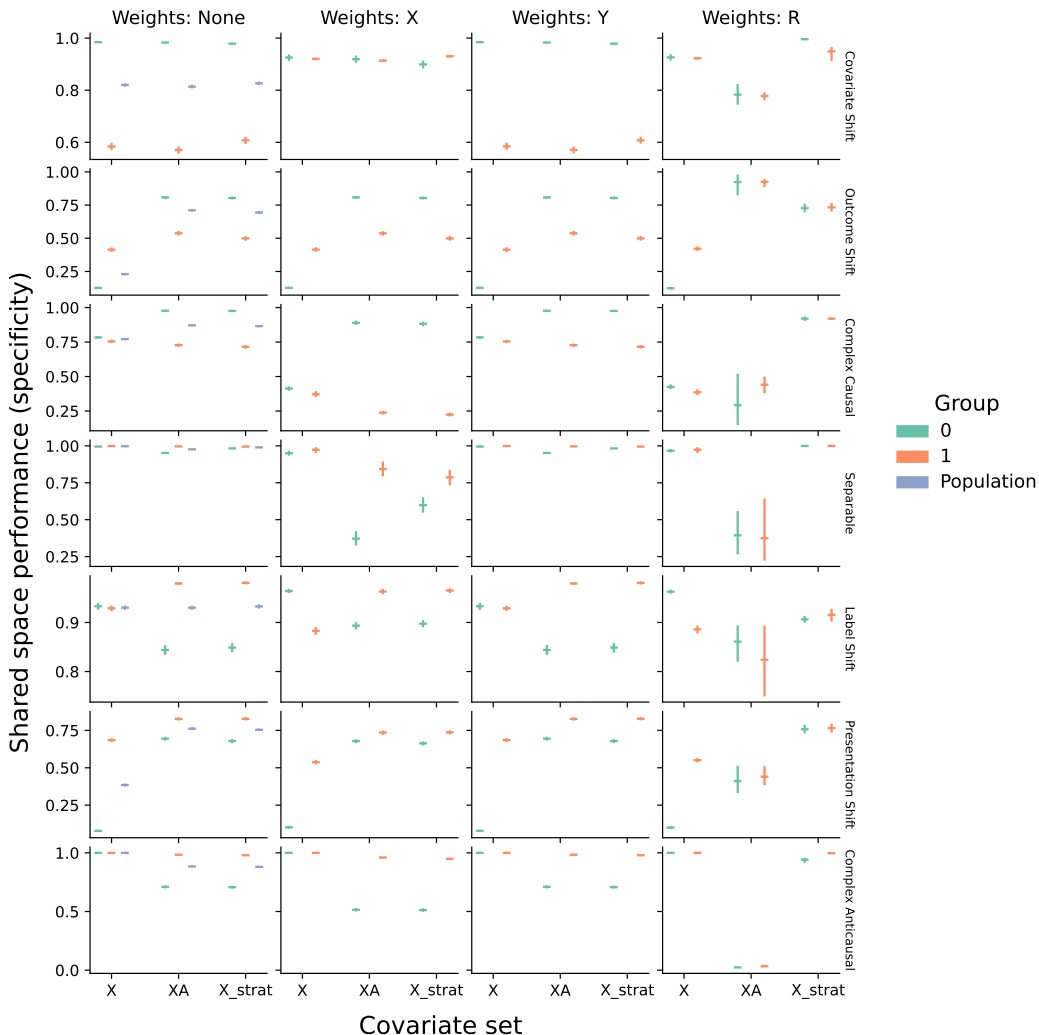

**Supplementary Figure B21: Simulation study: shared space subgroup specificity**. Plotted are the average performance with 95% confidence intervals for subgroup performance following weighting to a shared space, using the approach of Cai et al. [16]. Columns correspond to different conditioning variables used to construct the weights and rows correspond data generating processes.

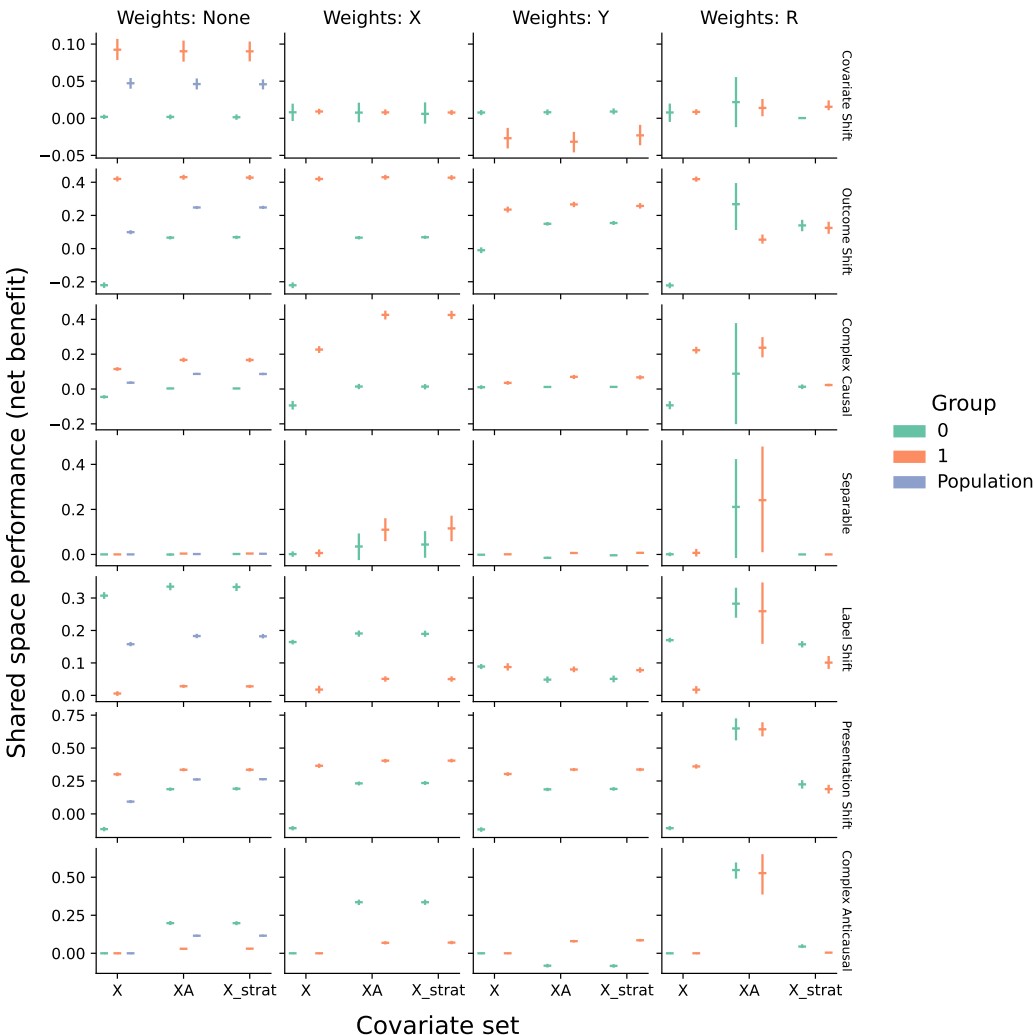

**Supplementary Figure B22: Simulation study: shared space subgroup net benefit**. Plotted are the average performance with 95% confidence intervals for subgroup performance following weighting to a shared space, using the approach of Cai et al. [16]. Columns correspond to different conditioning variables used to construct the weights and rows correspond data generating processes.

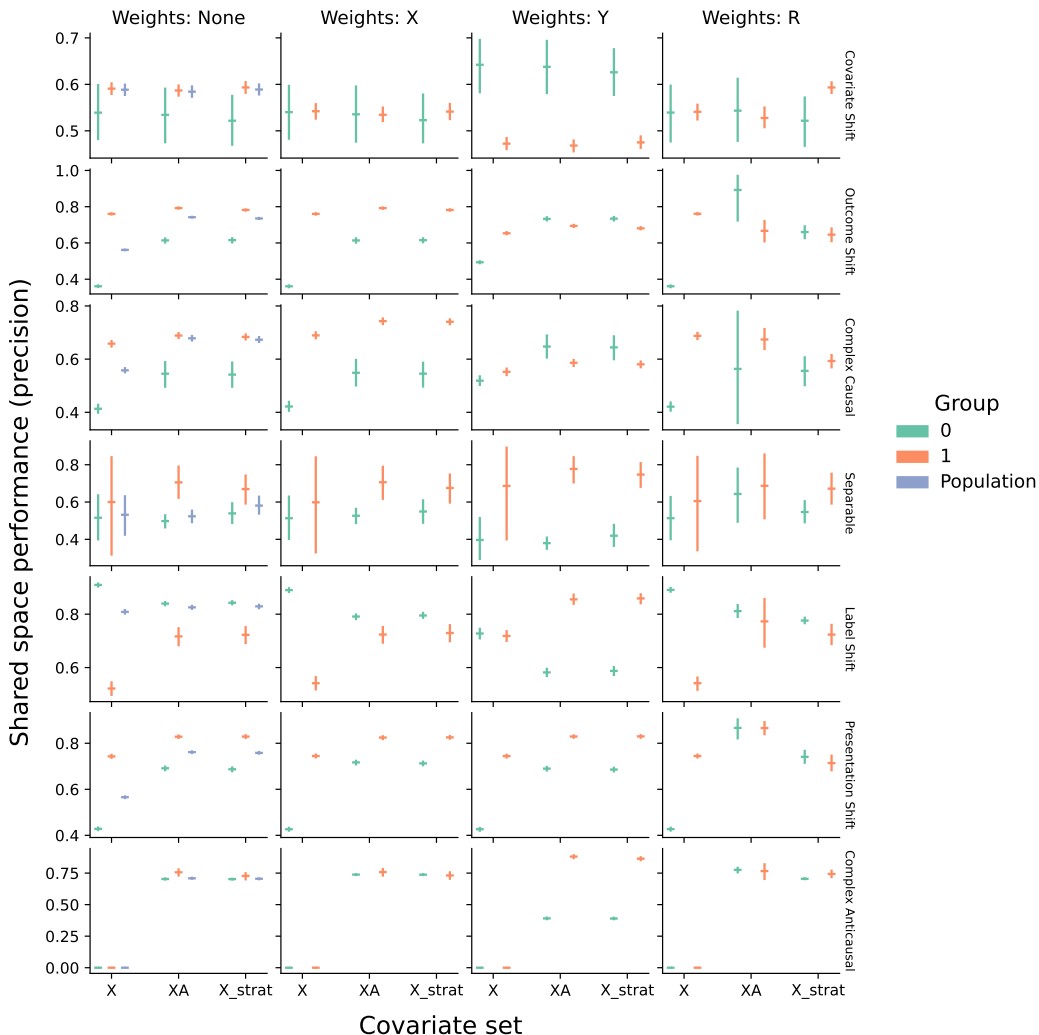

**Supplementary Figure B23: Simulation study: shared space subgroup precision**. Plotted are the average performance with 95% confidence intervals for subgroup performance following weighting to a shared space, using the approach of Cai et al. [16]. Columns correspond to different conditioning variables used to construct the weights and rows correspond data generating processes.

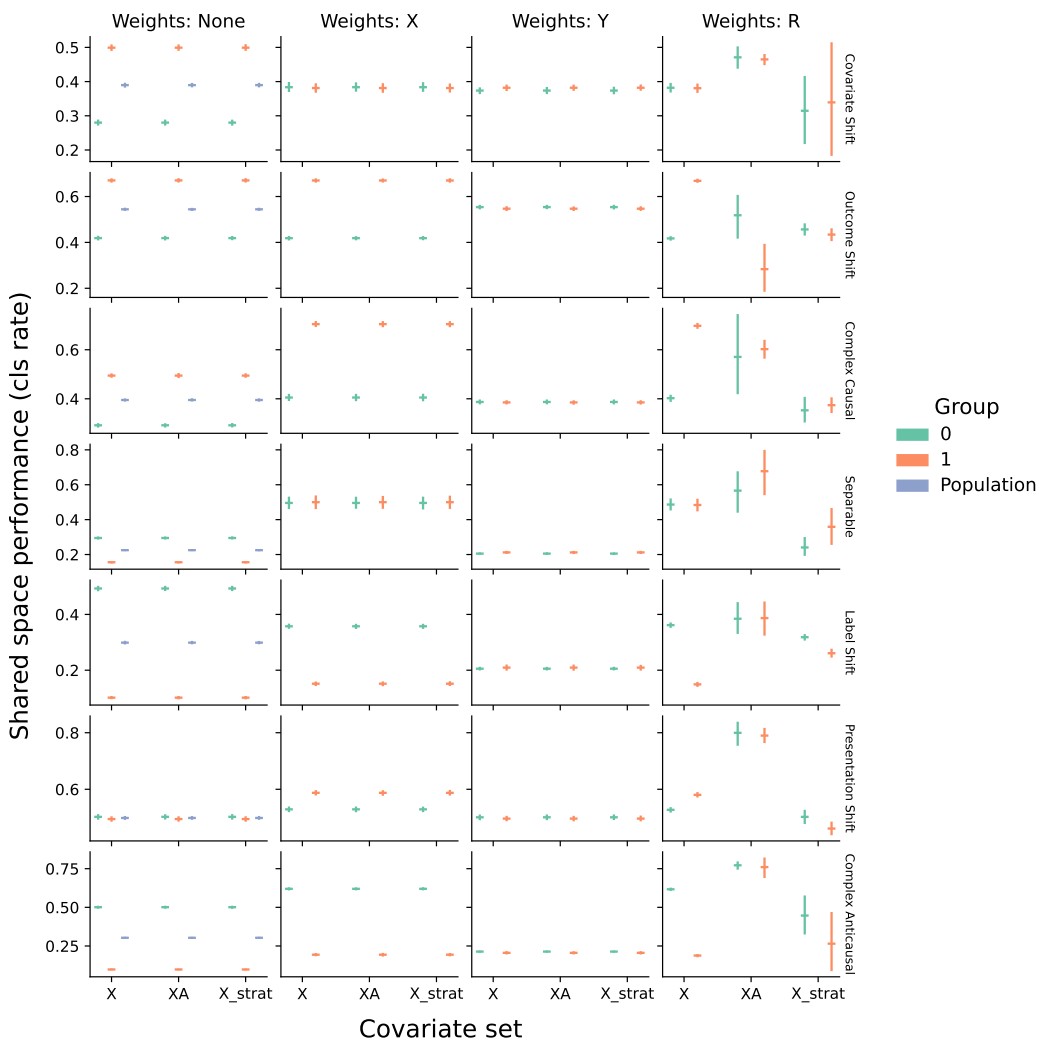

**Supplementary Figure B24: Simulation study: shared space subgroup classification rate**. Plotted are the average performance with 95% confidence intervals for subgroup performance following weighting to a shared space, using the approach of Cai et al. [16]. Columns correspond to different conditioning variables used to construct the weights and rows correspond data generating processes.

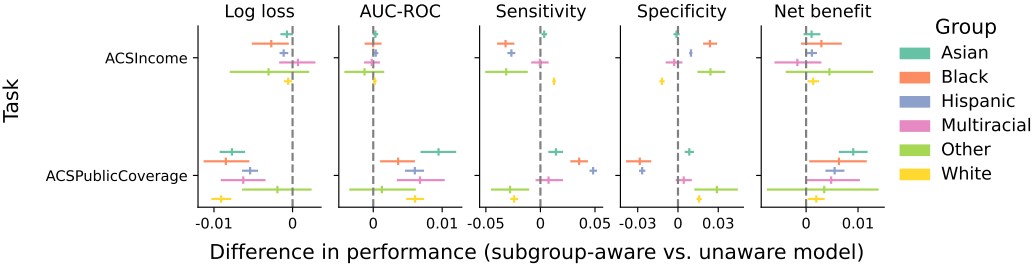

**Supplementary Figure B25: ACS PUMS: the effect of subgroup-aware prediction on model performance**. We report the difference in performance between models that have access to subgroup membership as an additional covariate as compared to those that do not. Plotted are average differences with 95% confidence intervals for each setting and for several performance metrics.

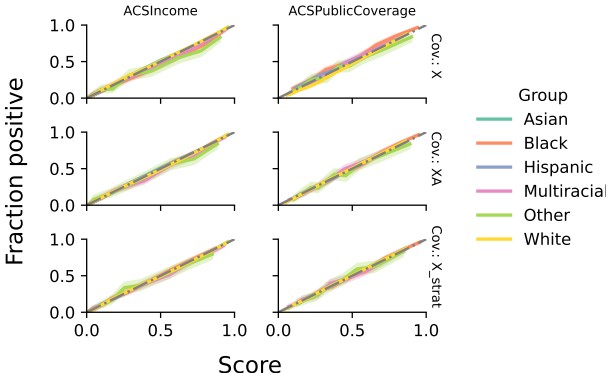

**Supplementary Figure B26: ACS PUMS: calibration curves**. Plotted are calibration curves for each subgroup with 95% confidence intervals. The first row corresponds to subgroup-agnostic prediction, the second row to prediction with $A$ as an additional covariate, and the third row to stratified prediction by $A$.

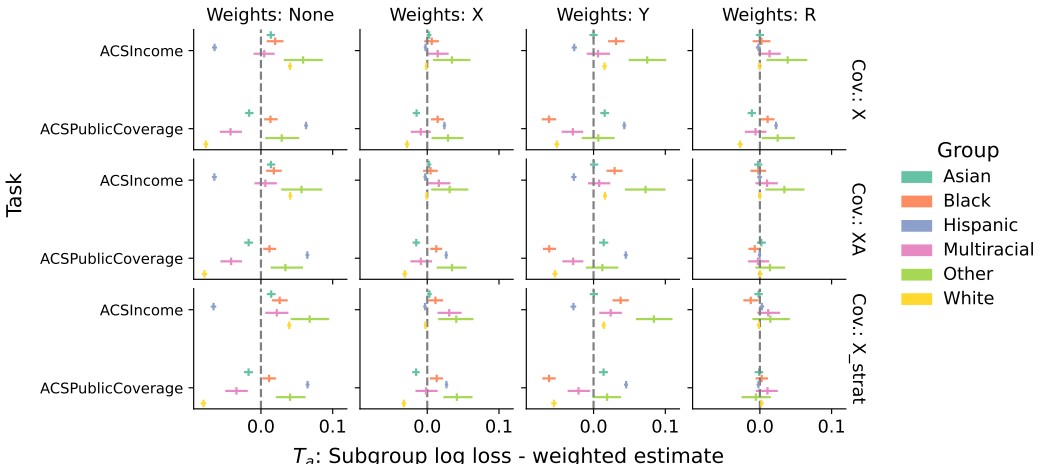

**Supplementary Figure B27: ACS PUMS: controlled evaluation of log loss**. Plotted are the statistics $T_a$ with 95% confidence intervals, corresponding to differences between the unweighted disaggregated performance with the population performance weighted to match the distribution of $X$, $Y$, or $R$ on the subgroups. The first row corresponds to subgroup-agnostic prediction, the second row to prediction with $A$ as an additional covariate, and the third row to stratified prediction by $A$.

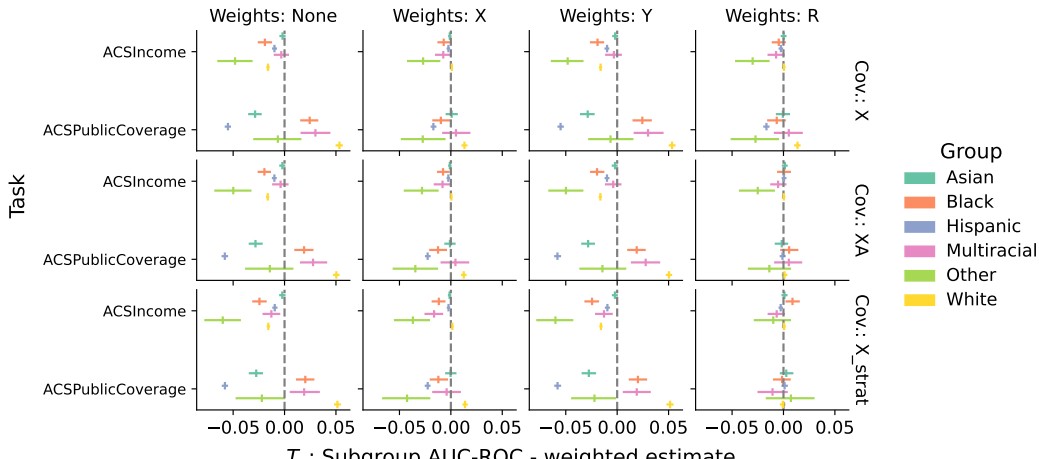

**Supplementary Figure B28: ACS PUMS: controlled evaluation of AUC-ROC**. Plotted are the statistics $T_a$ with 95% confidence intervals, corresponding to differences between the unweighted disaggregated performance with the population performance weighted to match the distribution of $X$, $Y$, or $R$ on the subgroups. The first row corresponds to subgroup-agnostic prediction, the second row to prediction with $A$ as an additional covariate, and the third row to stratified prediction by $A$.

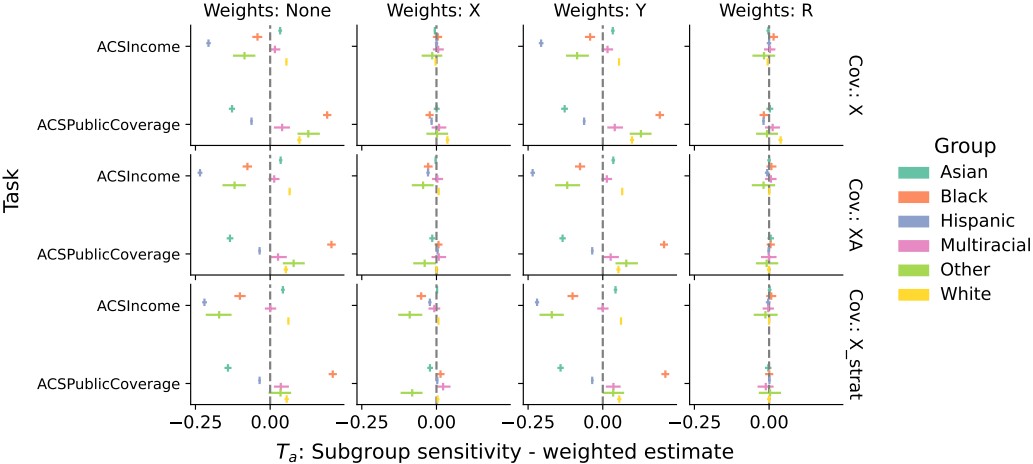

**Supplementary Figure B29: ACS PUMS: controlled evaluation of sensitivity**. Plotted are the statistics $T_a$ with 95% confidence intervals, corresponding to differences between the unweighted disaggregated performance with the population performance weighted to match the distribution of $X$, $Y$, or $R$ on the subgroups. The first row corresponds to subgroup-agnostic prediction, the second row to prediction with $A$ as an additional covariate, and the third row to stratified prediction by $A$.

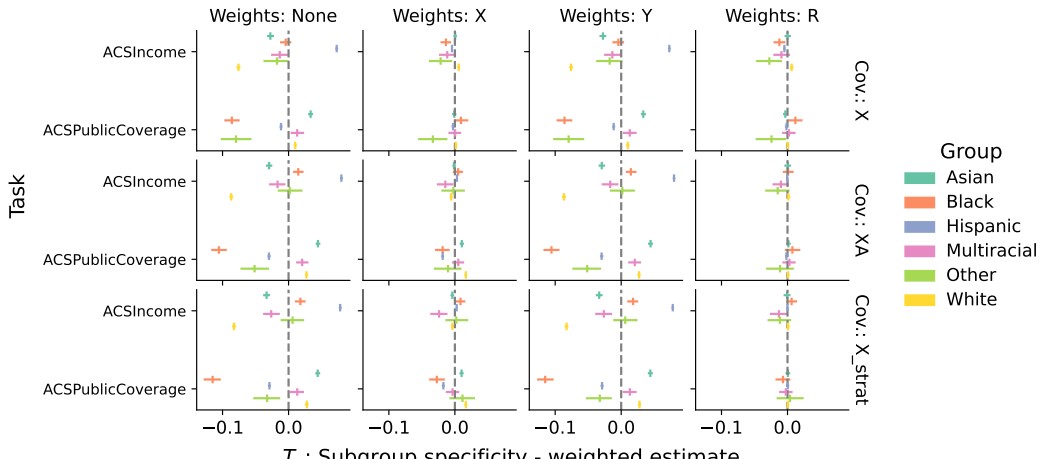

**Supplementary Figure B30: ACS PUMS: controlled evaluation of specificity**. Plotted are the statistics $T_a$ with 95% confidence intervals, corresponding to differences between the unweighted disaggregated performance with the population performance weighted to match the distribution of $X$, $Y$, or $R$ on the subgroups. The first row corresponds to subgroup-agnostic prediction, the second row to prediction with $A$ as an additional covariate, and the third row to stratified prediction by $A$.

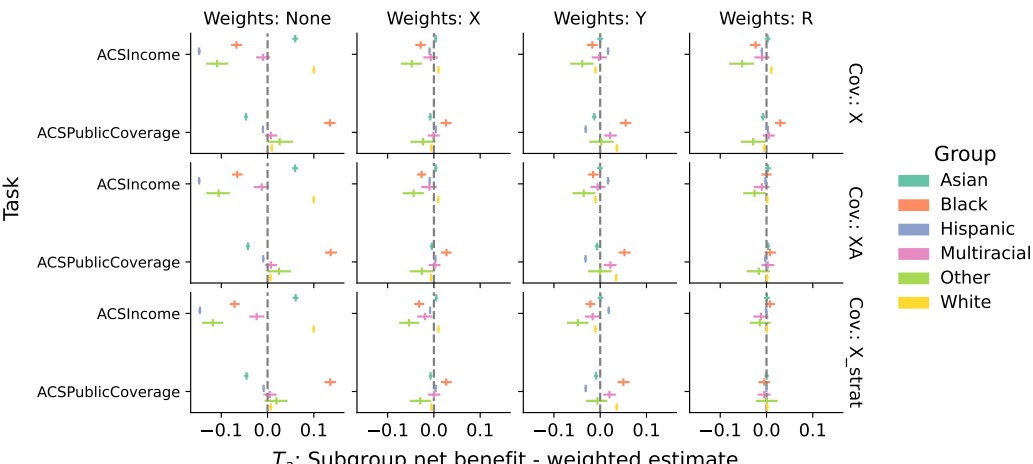

**Supplementary Figure B31: ACS PUMS: controlled evaluation of net benefit**. Plotted are the statistics $T_a$ with 95% confidence intervals, corresponding to differences between the unweighted disaggregated performance with the population performance weighted to match the distribution of $X$, $Y$, or $R$ on the subgroups. The first row corresponds to subgroup-agnostic prediction, the second row to prediction with $A$ as an additional covariate, and the third row to stratified prediction by $A$.

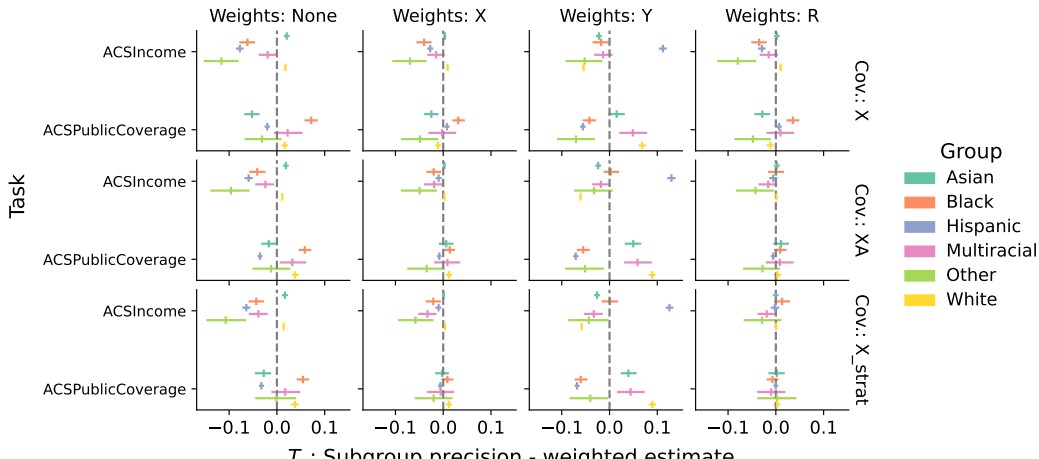

**Supplementary Figure B32: ACS PUMS: controlled evaluation of precision**. Plotted are the statistics $T_a$ with 95% confidence intervals, corresponding to differences between the unweighted disaggregated performance with the population performance weighted to match the distribution of $X$, $Y$, or $R$ on the subgroups. The first row corresponds to subgroup-agnostic prediction, the second row to prediction with $A$ as an additional covariate, and the third row to stratified prediction by $A$.

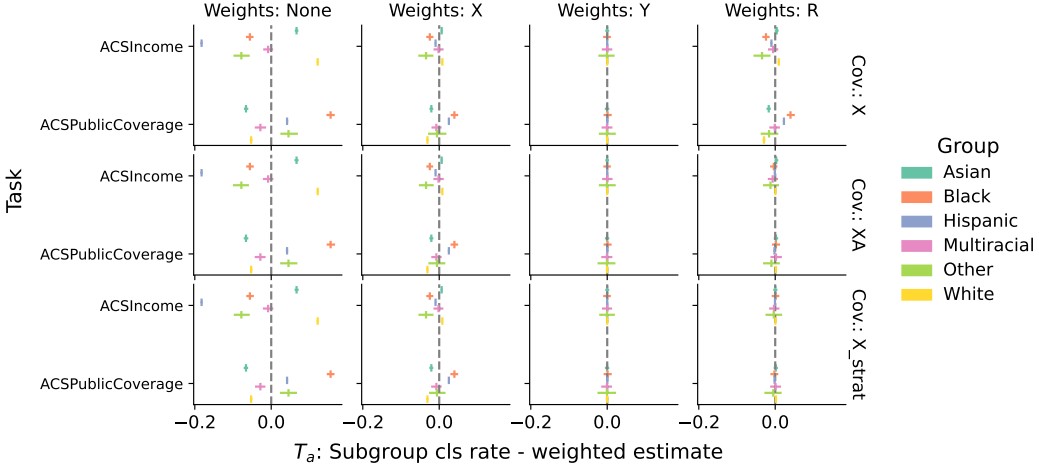

**Supplementary Figure B33: ACS PUMS: controlled evaluation of classification rate**. Plotted are the statistics $T_a$ with 95% confidence intervals, corresponding to differences between the unweighted disaggregated performance with the population performance weighted to match the distribution of $X$, $Y$, or $R$ on the subgroups. The first row corresponds to subgroup-agnostic prediction, the second to row prediction with $A$ as an additional covariate, and the third row to stratified prediction by $A$.

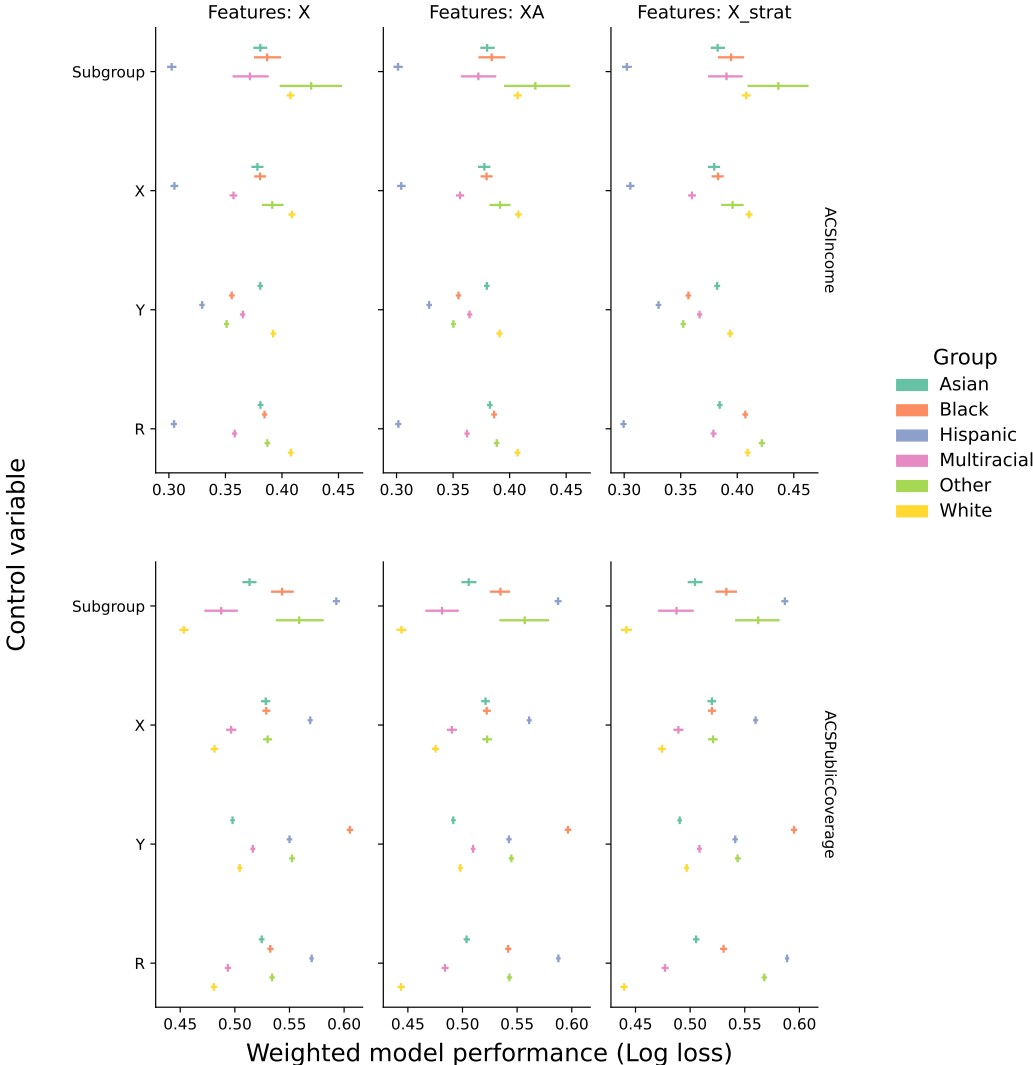

**Supplementary Figure B34: ACS PUMS: weighted estimation of log loss**. Plotted are the weighted estimates of performance $M_a$ with 95% confidence intervals, corresponding to weighted estimates of population performance weighted to match the distribution of $X$, $Y$, or $R$ for each subgroups. The entry labeled "subgroup" corresponds to the unweighted estimate of subgroup performance. The first column corresponds to subgroup-agnostic prediction, the second column to prediction with $A$ as an additional covariate, and the third column to stratified prediction by $A$.

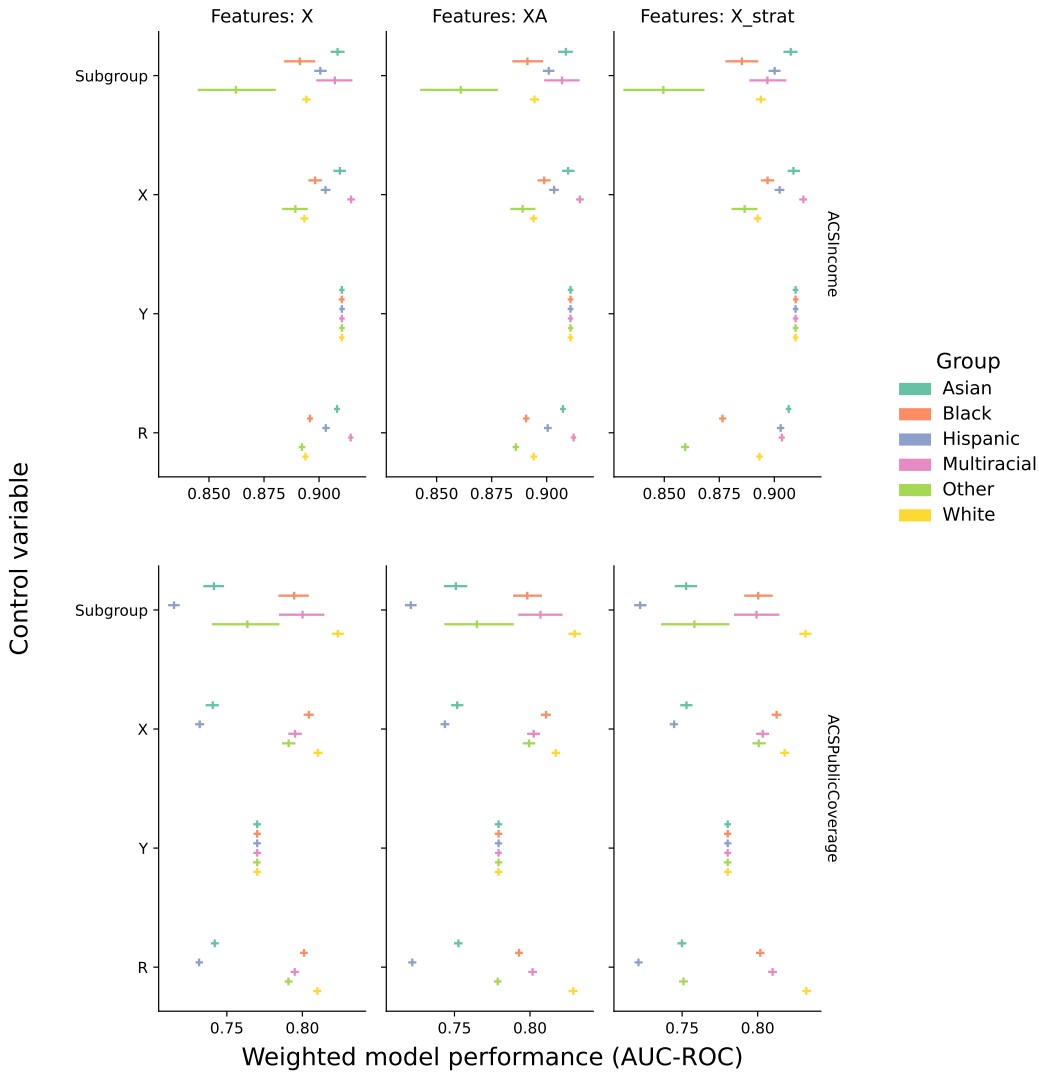

**Supplementary Figure B35: ACS PUMS: weighted estimation of AUC-ROC**. Plotted are the weighted estimates of performance $M_a$ with 95% confidence intervals, corresponding to weighted estimates of population performance weighted to match the distribution of $X$, $Y$, or $R$ for each subgroups. The entry labeled "subgroup" corresponds to the unweighted estimate of subgroup performance. The first column corresponds to subgroup-agnostic prediction, the second column to prediction with $A$ as an additional covariate, and the third column to stratified prediction by $A$.

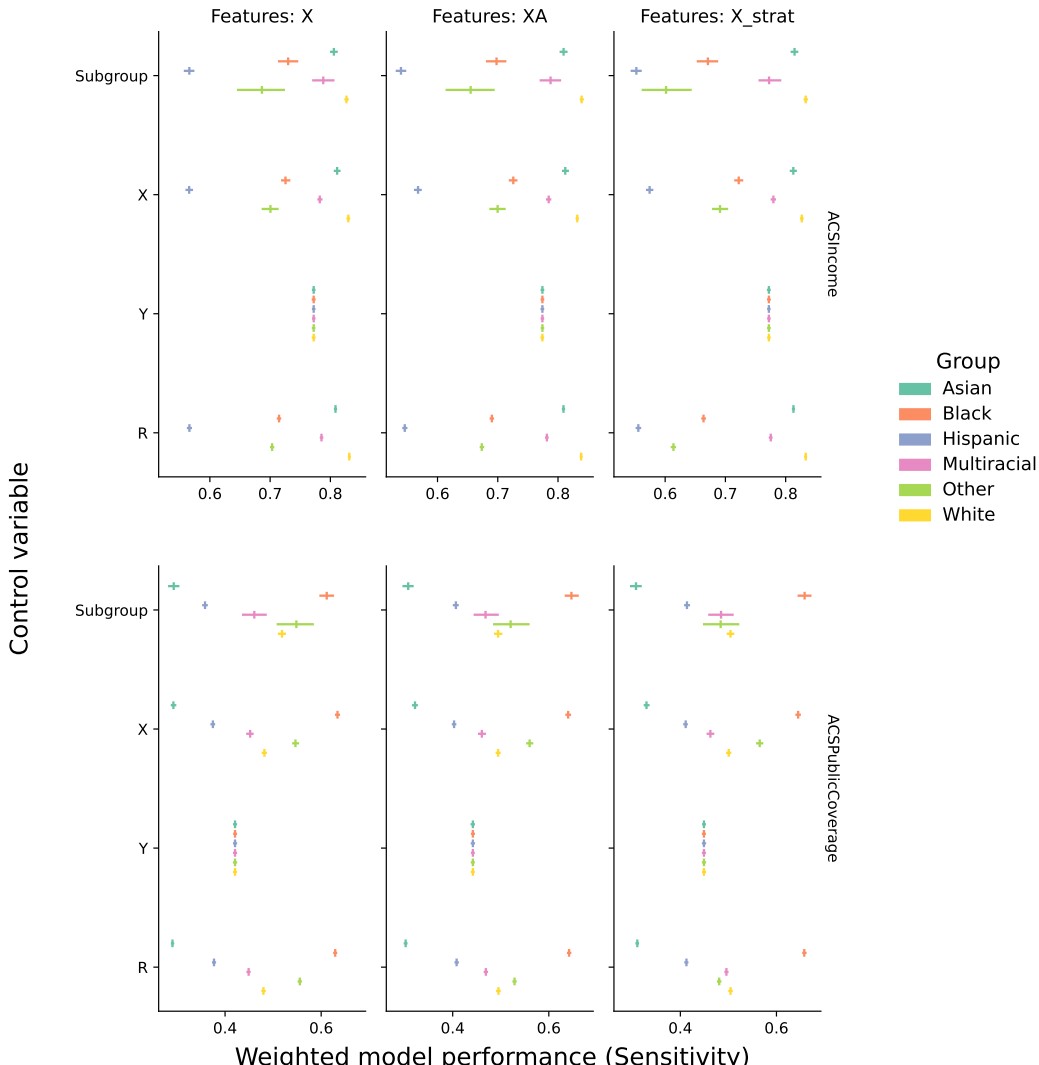

**Supplementary Figure B36: ACS PUMS: weighted estimation of sensitivity**. Plotted are the weighted estimates of performance $M_a$ with 95% confidence intervals, corresponding to weighted estimates of population performance weighted to match the distribution of $X$, $Y$, or $R$ for each subgroups. The entry labeled "subgroup" corresponds to the unweighted estimate of subgroup performance. The first column corresponds to subgroup-agnostic prediction, the second column to prediction with $A$ as an additional covariate, and the third column to stratified prediction by $A$.

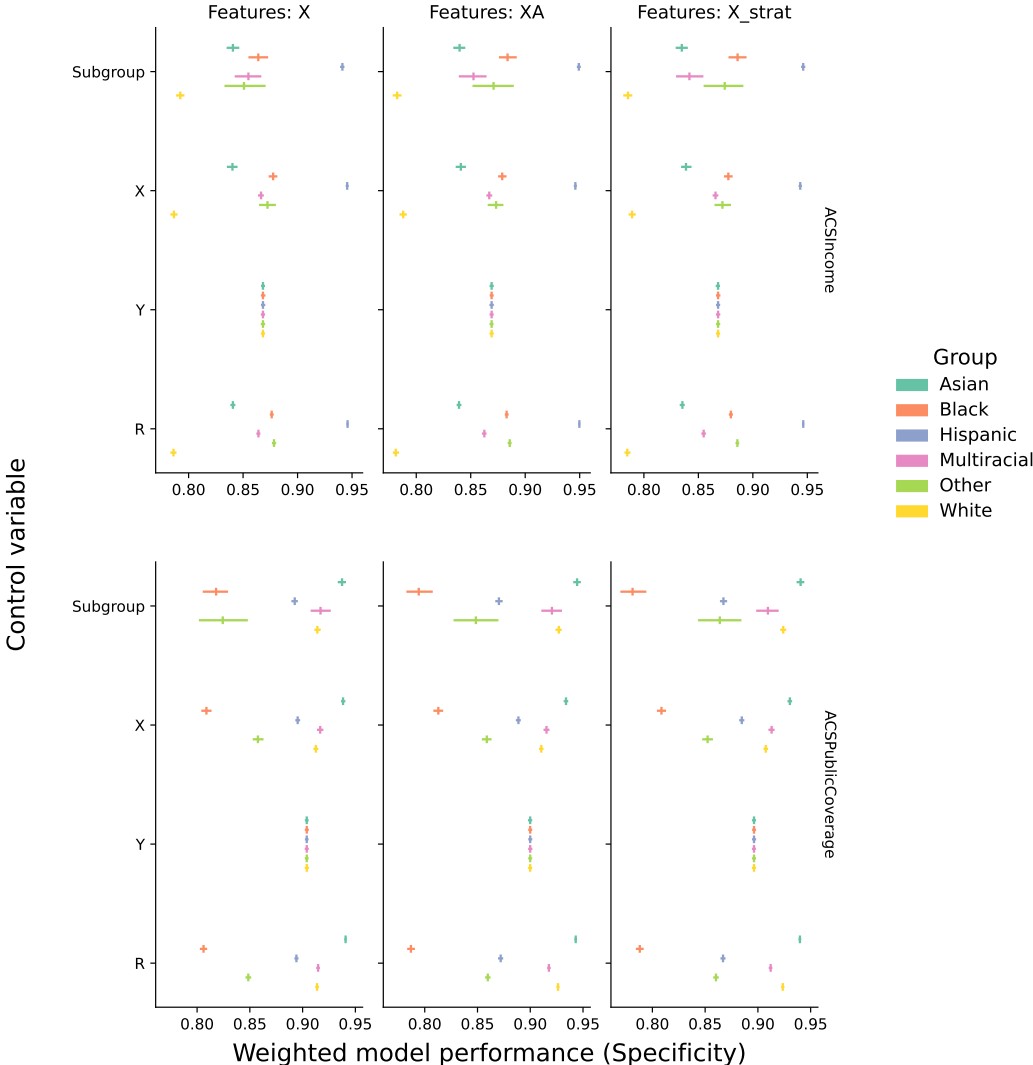

**Supplementary Figure B37: ACS PUMS: weighted estimation of specificity**. Plotted are the weighted estimates of performance $M_a$ with 95% confidence intervals, corresponding to weighted estimates of population performance weighted to match the distribution of $X$, $Y$, or $R$ for each subgroups. The entry labeled "subgroup" corresponds to the unweighted estimate of subgroup performance. The first column corresponds to subgroup-agnostic prediction, the second column to prediction with $A$ as an additional covariate, and the third column to stratified prediction by $A$.

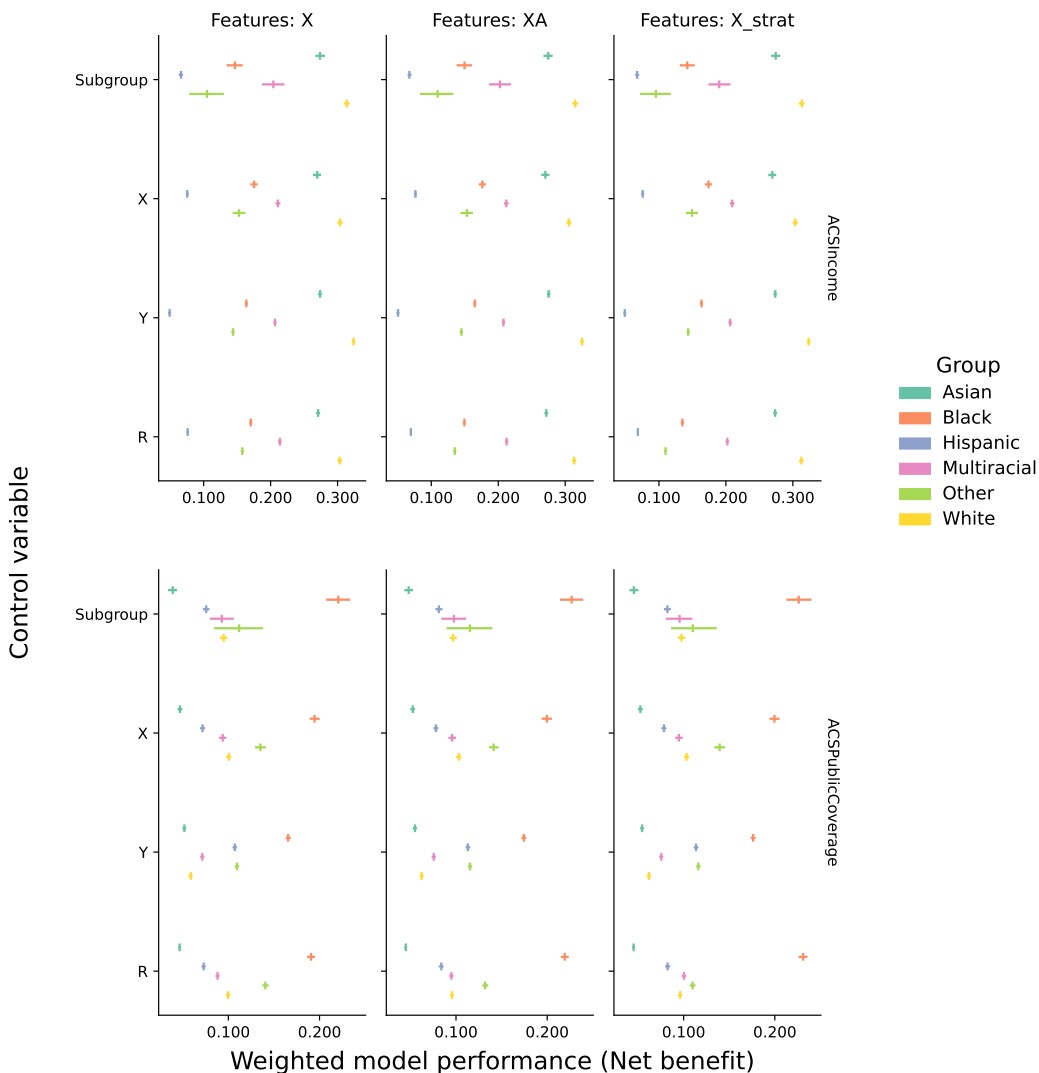

**Supplementary Figure B38: ACS PUMS: weighted estimation of net benefit**. Plotted are the weighted estimates of performance $M_a$ with 95% confidence intervals, corresponding to weighted estimates of population performance weighted to match the distribution of $X$, $Y$, or $R$ for each subgroups. The entry labeled "subgroup" corresponds to the unweighted estimate of subgroup performance. The first column corresponds to subgroup-agnostic prediction, the second column to prediction with $A$ as an additional covariate, and the third column to stratified prediction by $A$.

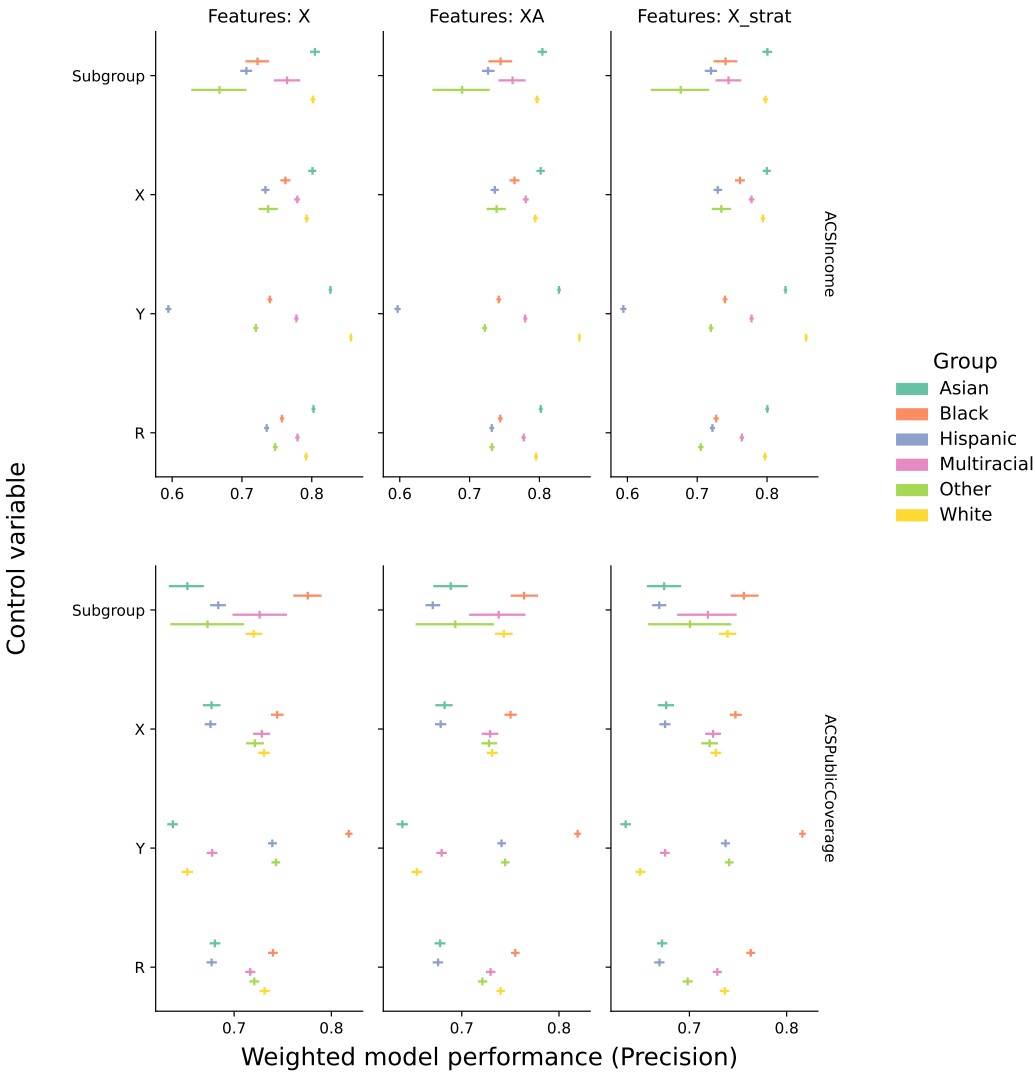

**Supplementary Figure B39: ACS PUMS: weighted estimation of precision**. Plotted are the weighted estimates of performance $M_a$ with 95% confidence intervals, corresponding to weighted estimates of population performance weighted to match the distribution of $X$, $Y$, or $R$ for each subgroups. The entry labeled "subgroup" corresponds to the unweighted estimate of subgroup performance. The first column corresponds to subgroup-agnostic prediction, the second column to prediction with $A$ as an additional covariate, and the third column to stratified prediction by $A$.

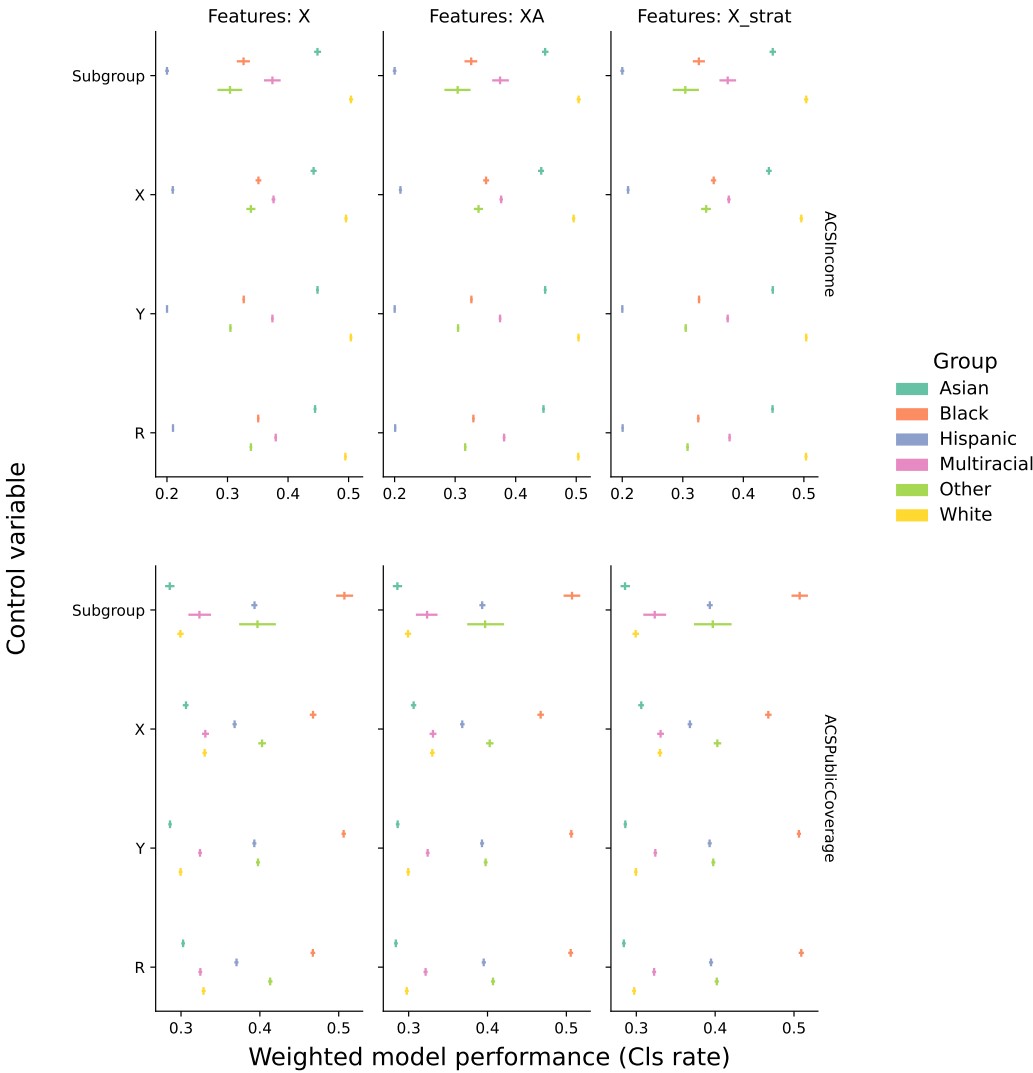

**Supplementary Figure B40: ACS PUMS: weighted estimation of classification rate**. Plotted are the weighted estimates of performance $M_a$ with 95% confidence intervals, corresponding to weighted estimates of population performance weighted to match the distribution of $X$, $Y$, or $R$ for each subgroups. The entry labeled "subgroup" corresponds to the unweighted estimate of subgroup performance. The first column corresponds to subgroup-agnostic prediction, the second column to prediction with $A$ as an additional covariate, and the third column to stratified prediction by $A$.

