# OpenReview forum: "Understanding challenges to the interpretation of disaggregated evaluations of algorithmic fairness"
_NeurIPS.cc/2025/Conference — NeurIPS 2025 poster_

### Official Review · Reviewer_Lqbf · 2025-06-27

**Clarity:** 1
**Significance:** 3
**Originality:** 2
**Rating:** 4
**Confidence:** 3

**Summary:**

The paper investigates why evaluating fairness by comparing model performance across disaggregated groups (e.g., per-race or per-gender accuracy) can be misleading. The authors:

- Analyze multiple causal data-generation processes (e.g., covariate shift) using DAGs to illustrate when disparities reflect underlying population inequities versus algorithmic unfairness.
- Derive conditions under which Bayes-optimal models satisfy fairness notions such as separation or sufficiency.
- Propose a controlled-evaluation method using a control variable $V$, combined with a statistic $T_a$ to test whether observed subgroup disparities are confounded.
- Validate their approach with simulations and experiments on public census data (ACS PUMS), demonstrating that causal structure—rather than raw accuracy differences—often drives conclusions about fairness or unfairness.

**Questions:**

- **Readability and narrative structure**: Could you consider improving the exposition by:
  – Breaking down complex sentences and clarifying transitions?
  – Introducing motivating examples early (e.g., a simple causal scenario described in words and illustrated with a DAG)?
  – Incrementally layering formalism, while moving dense tables or derivations (e.g., conditional independence relationships) to the appendix?
  – Providing a clearer high-level roadmap and smoother transitions across sections?
  – Labeling key results as Theorems or Lemmas to help orient the reader?

- **Robustness under model misspecification**: Given that $T_a$ assumes Bayes-optimality, have the authors considered including a qualitative discussion or empirical illustration of how $T_a$ behaves under model misspecification (e.g., due to limited model capacity, regularization, or optimization error)?

- **Guidance for selecting control variable $V$**: Can the authors provide more concrete guidance or examples on how to choose the control variable $V$ in different causal settings? Additionally, would it be possible to illustrate how $T_a$ varies under different reasonable choices of $V$, and how such variation might affect downstream interpretation?

- **Broader empirical evaluation**: Are there plans to extend the empirical evaluation beyond the ACS dataset? Including additional domains—such as healthcare (e.g., MIMIC) or criminal justice (e.g., COMPAS)—or simulating more complex environments involving multiple confounders could help demonstrate the method’s broader applicability and robustness.

- **Stakeholder communication**: Since many stakeholders expect to see raw subgroup disparities, can the authors suggest how conclusions based on $T_a$ might be presented in practice? For example, could templates or visual aids be proposed that show both raw metrics and $T_a$-adjusted insights, to help convey the causal reasoning to non-technical audiences?

**Ethical Concerns:**

["NO or VERY MINOR ethics concerns only"]

**Final Justification:**

Thank you for the thoughtful and constructive revisions. The added algorithm block, improved clarity, and guidance on selecting V significantly enhance the paper’s accessibility and practical relevance. While the empirical scope is appropriate for the venue, I encourage minor restructuring of the experiment section to make key insights more accessible (e.g., via a summary table). With these changes, I believe the paper is in good shape for acceptance.

**Limitations:**

Yes

**Paper Formatting Concerns:**

No formatting issues detected

**Quality:**

2

**Strengths And Weaknesses:**

**Strengths**

- **Strong conceptual contribution**: The paper thoughtfully clarifies why subgroup performance gaps may not reliably indicate unfairness, grounding its arguments in causal reasoning. This represents a meaningful advancement over more simplistic fairness evaluation approaches.

- **Highlights an important challenge in subgroup evaluations**: Many fairness methods rely on detecting metric differences between subpopulations. This work insightfully demonstrates that such disparities may arise from deeper structural confounders, suggesting that raw subgroup gaps should motivate further causal analysis rather than immediate conclusions about unfairness.

- **Diagnostic framework with practical utility**: The controlled evaluation procedure, which conditions away confounding, provides a principled and implementable approach. Its reliance on existing data and plausible causal assumptions makes it accessible to practitioners.

- **Empirical relevance**: The ACS PUMS evaluation effectively illustrates multiple failure modes in fairness assessments, grounding the theoretical insights in real-world data.

**Weaknesses**

- **Clarity and writing style**: The writing is dense and frequently difficult to follow, particularly in the introduction and theoretical sections. Long, multi-clause sentences combine several ideas without clear transitions or structure. The paper makes strong theoretical points, but the dense use of notation and compact causal arguments can hinder readability. Several key ideas would benefit from more intuitive explanation before being formalized. A clearer high-level roadmap and smoother transitions across sections would improve accessibility. This is, in my view, the main weakness of the paper and currently prevents me from increasing my score. However, I would be open to revising my evaluation if the authors significantly improve the exposition.

- **Model misspecification not empirically explored**: Although the paper warns that controlled evaluation may misinterpret results if models are misspecified, it does not empirically assess how robust $T_a$ is with real (non–Bayes-optimal) models. A simple experiment or sensitivity analysis would make the warning more actionable for practitioners.

- **Control variable ($V$) selection remains somewhat abstract**: The authors advise selecting $V$ based on domain knowledge, but practitioners may find this challenging without more concrete guidance. Including specific examples illustrating how to select $V$ in realistic scenarios would make the method more accessible and actionable.

- **Narrow empirical scope**: While the ACS case study is valuable, it is limited to two predictive tasks. Additional experiments in other domains (e.g., healthcare, criminal justice) would strengthen the paper’s generality.

- **Limited procedural guidance**: While the controlled-evaluation concept is generally clear, the paper would benefit from modular pseudo-code or a step-by-step outline describing data preparation, weight computation, and significance testing for $T_a$ to aid reproducibility.

---

> ### Author Rebuttal · Authors · 2025-07-30
>
> We thank reviewer Lqbf for their constructive feedback. We particularly appreciate the recommendations as to how to improve the readability and clarity of the work. At the end of the response, we include a detailed list of proposed changes to the manuscript to be made by the time of the camera-ready submission.
>
> **Clarity, writing style, and readability**:
> * We propose to make a series of changes to the manuscript to aid in clarity and readability. This includes breaking up long sentences, adding concrete examples, and adding conceptual scaffolding and transitions to guide the reader through the technical content.
> * As mentioned, we include a detailed description of those changes at the end of this response.
> * To address the concern about procedural reproducibility, we will include an algorithm block describing the computation of the weighted statistic $T_a$ and include code to reproduce the experiments.
>
> **Robustness under model misspecification**:
>
> We agree that systematic assessment of the role of model misspecification and estimation error would enhance the work. In future work, we hope to advance the theoretical understanding of the estimators and estimation procedures we propose and systematically evaluate the role of misspecification and estimation error. For this purpose, we expect that techniques from the semiparametric estimation literature that treats the optimal predictive model as a nuisance function will provide insight. Epistemic uncertainty quantification (with e.g., ensembles or Bayesian models) may also prove useful as a practical means of assessing variance in the estimator, as a proxy for closeness to Bayes-optimality. We will update the discussion section to refer to these as promising future directions.
>
> We would also like to clarify that the use of controlled evaluation procedures does not require Bayes-optimality, but interpretation can differ depending on whether the model is Bayes-optimal. In the original manuscript, this is conveyed by the rows of Table 3 that refer to an arbitrary model $f$ that is not Bayes-optimal. We acknowledge that this could be made clearer in the manuscript and we intend to do so through textual revision to sections 3.3 and 3.4. We have also drafted a new table (to be included in the appendix) that details the set of implications and valid conclusions that can be drawn from controlled evaluations based on the configuration of the weights, the covariate set, the observed statistic $T_a$, and whether the model is Bayes-optimal. Comparison of the set of valid conclusions with and without Bayes-optimality clarifies when and how Bayes-optimality is relevant. We include the content of this table in full in the response to reviewer c4Qe.
>
> Finally, we would like to highlight that the experiments with ACS PUMS are plausibly conducted in a practical setting where Bayes-optimality cannot be assumed. As we discuss, interpreting controlled evaluation procedures as conditional independence testing provides the ability to falsify causal assumptions and support understanding of why performance differs across subgroups. In the results section (lines 359-362), we see some deviation from Bayes-optimal behavior through miscalibration of subgroup-aware prediction for the “Other” subgroup in the ACSIncome task.
>
> **Guidance for selecting control variables**:
>
> We agree that clearer recommendations for control variable choice and interpretation would improve the actionability and accessibility of the work. We intend to include a new paragraph at the end of section 3.4 that includes a clear set of recommendations for use and interpretation of each of the controlled evaluation procedures that we discuss (that is, control for $X$, $Y$, or $R$). We detail the recommendations for interpretation in the new table provided in response to reviewer c4Qe (also mentioned above in the discussion of robustness under model misspecification). We also revise lines 366-370 in the discussion section to clarify that the choice of control variable depends on the research question and inference of interest with the interpretation dependent on causal assumptions.
>
> **Broader empirical evaluation**:
>
> We agree that further empirical evaluation would strengthen the work. However, we do believe that the present scope is sufficient to address the immediate research questions and appropriate for the space allotted. In future work, we intend to apply our approach in applied contexts (e.g., in healthcare) and conduct a richer theoretical and empirical investigation of methodological considerations not thoroughly addressed here (e.g., model misspecification and finite sample error; separability in high dimensional data).
>
> **Stakeholder communication**:
>
> Our position is that the controlled disaggregated evaluations we propose are complementary to a broader set of analytic practices, including standard uncontrolled evaluations, assessment of calibration, plotting predictive distributions, characterization of the data distribution, reflection on the appropriateness of the problem formulation and assumptions, and transparent reporting. We believe that further interdisciplinary research (e.g., in HCI, human factors, and data visualization) is needed to understand how to best present this broader collection of analyses and visualizations to stakeholders.
>
> **To address this and related concerns, we concretely propose to**:
> * Add additional discussion regarding implications of this work for the general interpretation of disaggregated evaluation.
> * Emphasize the need for future research to understand how to best present controlled evaluation results alongside other findings.
> * Provide new guidance for use and interpretation of controlled evaluations in practical settings, mentioned above and detailed in the response to review c4Qe.
>
> **Here, we outline in detail the proposed changes to aid in clarity**:
> * Introduction
>   * Introduce running examples that can be reused throughout the exposition in the introduction and the presentation of the causal graphs. To expand upon the reference to the use of algorithmic fairness in healthcare, we intend to refer to examples from cardiovascular risk estimation (CVD) and chest X-ray classification (CXR; see refs cited inline). This can be referred to at the end of the first paragraph, again in paragraph two when discussing interventions, and throughout section 3.1 to ground the initial presentation of the causal graphs.
>   * Third paragraph (lines 28-48):
>     * Break up long sentences, including sentences 2 (lines 28-32) and 3 (lines 32-26).
>     * Split this paragraph in two, separately discussing the issues related to subgroup performance stability and misrepresentative data.
>     * Insert a new second sentence (line 29) that presents a high-level, intuitive description of the problem before additional details.
> * Section 3
>   * Add a new paragraph prior to section 3.1 that provides conceptual scaffolding and roadmap for the technical sections that follow.
>   * In section 3.1, present the assumptions of each causal graph in terms of the use cases introduced above (CVD for causal direction graphs and CXR for anticausal).
>   * Section 3.2
>     * Add new exposition to the beginning of the section that clarifies its purpose in relation to the results in the subsequent sections. This can in part be rectified with the new summary section at the beginning of Section 3.
>     * Split up long sentences (lines 184-185, 196-198, 203-208).
>     * Consider moving Table 1 (conditional independence properties of Bayes-optimal models) to the appendix.
>     * Move section 3.5 (subgroup separability) to be a subsection of 3.2.
>   * Section 3.3
>     * Revise the first sentences to emphasize and clarify the purpose of the section.
>     * Clarify that the key findings presented in Table 3 regarding metric stability do not require Bayes-optimality, as Table 3 includes separate rows for arbitrary (not Bayes-optimal) and Bayes-optimal models. Provide explicit guidance about how to use Table 3 for reasoning in practical settings where neither Bayes-optimality nor complete causal structure can be assumed.
>   * Section 3.4
>     * Include an algorithm block that clearly describes the computation of $T_a$ (to include in the appendix).
>     * Include clear guidance for the design and interpretation of controlled disaggregated evaluation results under different conditions (control variable, set of covariates used, value of $T_a$, and Bayes-optimality) without knowledge of the causal graph. This will be incorporated into a revised version of the current final paragraph of this section. This content will be based on and refer to a new table (to be included in the appendix) covering the same content that is outlined in the response to reviewer c4Qe.
> * Experiments
>   * Make minor textual changes to better emphasize the relevance of the subgroup-aware modeling results in relation to the disaggregated evaluation results.
> * Discussion
>   * In the first paragraph, emphasize key practical takeaways. This includes recommendations for interpretation of standard disaggregated evaluations, the implications of the theoretical analysis and empirical findings, as well as guidance for use and interpretation of the methodology presented here.
>   * Revise lines 366-370 to clarify that the choice of control variable depends on the research question and inference of interest with the interpretation dependent on the causal assumptions.
>   * Include mention of additional areas of future research discussed elsewhere in this rebuttal, including the need for further theoretical and empirical investigation into inference under model misspecification and estimation error, extension to broader types of data (continuous, multi-class, high-dimensional, sparse), extension to forms of data misspecification beyond selection (e.g., measurement error), and empirical study in specific contexts of interest (i.e., healthcare).

---

> > ### Comment · Reviewer_Lqbf · 2025-08-04
> >
> > Thank you for the thorough response and the concrete changes you’ve outlined. Your revisions to improve clarity, add an algorithm block, and provide guidance on selecting $V$ will enhance the paper’s accessibility and practical utility.
> >
> > Regarding empirical scope, I understand your point that the current ACS case study may suffice for this venue. To strengthen reader comprehension, I would still encourage you to refactor the experiment section so that key findings aren’t buried in dense prose (e.g. by summarizing conclusions in a table or boxed highlights).
> >
> > Assuming these revisions are implemented as described, I am happy to raise my score to 4.

---

> > > ### Author Response · Authors · 2025-08-04
> > > **Discussion period**
> > >
> > > Thank you for taking the time to engage with the rebuttal and for the willingness to raise your score to a 4. We will certainly make the changes to the paper as outlined and consider further revision to the text of the experiments section to emphasize the key empirical findings.

---

### Official Review · Reviewer_c4Qe · 2025-07-02

**Clarity:** 2
**Significance:** 3
**Originality:** 4
**Rating:** 5
**Confidence:** 2

**Summary:**

This paper explores theoretical and empirical properties of the disaggregated performance of Bayes-optimal models under different causal model assumptions. The paper lays out a framework for understanding the different types of causal relationships between labels, covariates, and group membership, and elucidates some theoretical properties of models under these different conditions. The overall argument of the work is that disaggregated metrics in a vacuum are not enough to fully understand whether or not a model is “fair” because some group-wise differences might be expected depending on the causal structure of the data generating process. For experiments, the paper shows a simulation study and measures the difference between subgroup-aware and subgroup-agnostic models on two Census Bureau ACS tasks.

**Questions:**

- The experiments show that subgroup-aware models have improved predictive performance over models that do not include subgroup membership information. What implications does this have for disaggregated evaluations? I personally had a hard time following the narrative of what we learn about disaggregated metrics from this difference in the models.

- How much does the Bayes optimality assumption affect the generalizability of the results shown here in practice?

- Are there novel metrics or techniques that the authors would propose to alleviate some of the issues that they find in the work?

**Ethical Concerns:**

["NO or VERY MINOR ethics concerns only"]

**Final Justification:**

After considering the author responses and other reviews, I raised my score to a 5. I think the concerns about clarity that I raised will be sufficiently addressed in the final revision.

**Limitations:**

I have discussed some limitations in the weaknesses above.

**Quality:**

3

**Strengths And Weaknesses:**

The primary strength of this paper is in its motivation and problem framing. I agree with the authors’ main conceit that disaggregated evaluations, on their own, do not necessarily tell the whole story of model performance. The authors have introduced an interesting and rigorous framework for understanding some of the pitfalls that can come from over-reliance on subgroup disaggregation.

The main weakness of this paper, in my opinion, is the clarity of the presentation and the connection between the experimental results and the overall argument that the paper is making. I see from both the empirical and theoretical results that subgroup-aware models tend to perform better than their agnostic counterparts under a variety of metrics in every causal setting except for covariate shift. I personally had a hard time connecting that empirical point back to the introduction and story of the paper regarding disaggregated evaluations. I have some more specific questions below that I hope will illustrate the confusion further.

Another weakness of the work is that I find it hard to take concrete recommendations from the work. I do think some useful mathematical properties are illustrated, but I do not see much discussion in the work of suggestions for future evaluators. I think this could be expanded on for the camera-ready version.

Overall this paper seems like sound work, but could benefit from some improvements in clarity to hammer its points home.

---

> ### Author Rebuttal · Authors · 2025-07-30
>
> We thank reviewer c4Qe for their support of the work and their constructive comments. Below, we respond to each of the questions raised:
>
> **Regarding the relationship between disaggregated evaluation and analysis of the informativeness of subgroup membership**:
>
> Thank you for this feedback, as we agree that the current presentation is not entirely clear on this point. To clarify, despite the apparent disconnect between these two threads of inquiry (disaggregated evaluation vs. capacity of subgroup-aware modeling to improve prediction performance), there is a deep connection between them through the lens of conditional independence testing. This allows us to view both types of analysis as providing distinct views on the same underlying phenomena.
>
> For example, in the Bayes-optimal case both of the following findings are evidence against the same conditional independence $Y \perp A \mid X$ (i.e., the covariate shift assumption): (1) subgroup-aware modeling improves predictive performance over subgroup-agnostic modeling and (2) control for $X$ in disaggregated evaluation of a subgroup-agnostic model is insufficient to explain subgroup performance differences. Similarly, when we are in a setting where that covariate shift assumption is violated, we expect subgroup-agnostic prediction to induce sufficiency violation and miscalibration for subgroups, and this can be directly assessed with controlled evaluation on the basis of $R$. Notably, such findings with regards to calibration and sufficiency are also a signal that provides signal post-hoc that the model could be improved for subgroups for which it is miscalibrated.
>
> We intend to make a number of minor changes to the manuscript to aid in clarity. Please see the response to reviewer Lqbf for a detailed point-by-point description of those changes. Relevant to the points you raise, we intend to add additional conceptual scaffolding to the header of section 3 and to the individual subsections in order to better clarify the rhetorical purpose of each section with respect to the aims of the paper as a whole. In the experiments section, we also intend to make minor textual changes to better emphasize the relevance of the subgroup-aware modeling results in relation to the disaggregated evaluation results.
>
> **Regarding the role of Bayes-optimality**:
>
> We would like to clarify that the use of controlled evaluation procedures does not require Bayes-optimality, but interpretation can differ depending on whether the model is Bayes-optimal. In the original manuscript, this is conveyed by the rows of Table 3 that refer to an arbitrary model $f$ that is not Bayes-optimal. We acknowledge that this could be made clearer in the manuscript and we intend to do so through textual revision to sections 3.3 and 3.4. We have also drafted a new table (to be included in the appendix) that details the set of implications and valid conclusions that can be drawn from controlled evaluations based on the configuration of the weights, the covariate set, the observed statistic $T_a$, and whether the model is Bayes-optimal. Comparison of the set of valid conclusions with and without Bayes-optimality clarifies when and how Bayes-optimality is relevant. At the end of this response, we outline the content of that table.
>
> **Novel metrics and techniques**:
>
> In future work, we hope to advance the theoretical understanding of the estimators and estimation procedures we propose and systematically evaluate the role of misspecification and estimation error. For this purpose, we expect that techniques from the semiparametric estimation literature that treats the optimal predictive model as a nuisance function will provide insight. Epistemic uncertainty quantification (with e.g., ensembles or Bayesian models) may also prove useful as a practical means of assessing variance in the estimator, as a proxy for closeness to Bayes-optimality. We will update the discussion section to refer to these as promising future directions.
>
> **On practical guidance for practitioners**:
>
> We intend to make textual revision to sections 3.4 and to the discussion to provide further recommendations for practitioners. The detailed table of guidance regarding interpretation of controlled evaluation provided at the end of this response also serves this purpose. The revisions to the discussion will include both general guidance regarding interpretation of standard uncontrolled disaggregated evaluations as well as the controlled evaluation procedures we propose.
>
>
> **Content for a new table (to be added to the appendix) containing guidance regarding interpretation of controlled evaluations.** Each header corresponds to a setting of control variable, choice of covariates, controlled evaluation finding, and assumption regarding Bayes-optimality. Bulleted content under the headers are the set of implications and conclusions implied by the setting.
>
> Control: $X$, Covariates: $X$, Reject($T_a=0$): Yes, Bayes-optimal: Yes
> * Subgroup-dependent error structure conditioned on $X$ is present
> * Evidence that data not generated under covariate shift
> * Sufficiency violation likely to be present
> * Subgroup performances differences not explained by covariate shift
> * Possible to improve performance with subgroup-aware prediction
> * Improved subgroup-agnostic prediction is not possible
>
> Control: $X$, Covariates: $X$, Reject($T_a=0$): Yes, Bayes-optimal: No
> * Same as the Bayes-optimal case, with the exception that some performance differences may be mitigated through improved subgroup-agnostic modeling or data collection
>
> Control: $X$, Covariates: $X$, Reject($T_a=0$): No, Bayes-optimal: Yes
> * Results consistent with (i.e., cannot reject) the hypothesis of no subgroup-dependent error structure conditioned on $X$
> * Results consistent with (i.e., cannot reject) the hypothesis that data generated under covariate shift
> * Results consistent with (i.e., cannot reject) the hypothesis that all performance differences explained by covariate shift
>
> Control: $X$, Covariates: $X$, Reject($T_a=0$): No, Bayes-optimal: No
> * Results consistent with (i.e., cannot reject) the hypothesis of no subgroup-dependent error structure conditioned on $X$
> * Results consistent with (i.e., cannot reject) the hypothesis that data generated under covariate shift
> * Cannot rule out hypothesis that performance differences explained by systematic underperformance across subgroups
>
> Control: $Y$, Covariates: $X$, Reject($T_a=0$): Yes, Bayes-optimal: Yes
> * Subgroup-dependent error structure conditioned on $Y$ is present
> * Evidence that data not generated under label shift
>
> Control: $Y$, Covariates: $X$, Reject($T_a=0$): Yes, Bayes-optimal: No
> * Same as the Bayes-optimal case
>
> Control: $Y$, Covariates: $X$, Reject($T_a=0$): No, Bayes-optimal: Yes
> * Results consistent with (i.e., cannot reject) the hypothesis of no subgroup-dependent error structure conditioned on $Y$
> * Results consistent with (i.e., cannot reject) the hypothesis that data generated under label shift
> * Results consistent with (i.e., cannot reject) the hypothesis that the separation and equalized odds criteria hold
> * If label shift holds, properties implied by covariate shift violation may hold (i.e., sufficiency violation and informativeness of subgroup membership)
>
> Control: $Y$, Covariates: $X$, Reject($T_a=0$): No, Bayes-optimal: No
> * Same as the Bayes-optimal case
>
> Control: $Y$, Covariates: $\{X, A\}$, Reject($T_a=0$): Yes, Bayes-optimal: Yes
> * Subgroup-dependent error structure conditioned on $Y$ is present
>
> Control: $Y$, Covariates: $\{X, A\}$, Reject($T_a=0$): Yes, Bayes-optimal: No
> * Same as the Bayes-optimal case
>
> Control: $Y$, Covariates: $\{X, A\}$, Reject($T_a=0$): No, Bayes-optimal: Yes
> * Cannot reject hypothesis of no subgroup-dependent error structure conditioned on $Y$
>
> Control: $Y$, Covariates: $\{X, A\}$, Reject($T_a=0$): No, Bayes-optimal: No
> * Same as the Bayes-optimal case
>
> Control: $R$, Covariates: $X$, Reject($T_a=0$): Yes, Bayes-optimal: Yes
> * Evidence of sufficiency violation
> * Evidence that data not generated under covariate shift
> * Possible to improve performance with subgroup-aware prediction
>
> Control: $R$, Covariates: $X$, Reject($T_a=0$): Yes, Bayes-optimal: No
> * Evidence of sufficiency violation
> * Subgroup-aware prediction likely to improve performance
> * Evidence that data either not generated under covariate shift or model underperforms
>
> Control: $R$, Covariates: $X$, Reject($T_a=0$): No, Bayes-optimal: Yes
> * Results consistent with (i.e., cannot reject) the hypothesis that sufficiency is satisfied
> * Results consistent with (i.e., cannot reject) the hypothesis that data generated under covariate shift
>
> Control: $R$, Covariates: $X$, Reject($T_a=0$): No, Bayes-optimal: No
> * Results consistent with (i.e., cannot reject) the hypothesis that sufficiency is satisfied
>
> Control: $R$, Covariates: $\{X, A\}$, Reject($T_a=0$): Yes, Bayes-optimal: Yes
> * This outcome should not be possible if the weights are accurately estimated
>
> Control: $R$, Covariates: $\{X, A\}$, Reject($T_a=0$): Yes, Bayes-optimal: No
> * Evidence of sufficiency violation
> * Indicates that the model could be improved for one or more subgroups
>
> Control: $R$, Covariates: $\{X, A\}$, Reject($T_a=0$): No, Bayes-optimal: Yes
> * Results consistent with (i.e., cannot reject) the hypothesis that sufficiency is satisfied
>
> Control: $R$, Covariates: $\{X, A\}$, Reject($T_a=0$): No, Bayes-optimal: No
> * Same the Bayes-optimal case

---

> > ### Comment · Reviewer_c4Qe · 2025-08-05
> >
> > Thank you for the clarifications! After considering your response, assuming the proposed changes are made, I will raise my score to a 5.

---

### Official Review · Reviewer_oWko · 2025-07-03

**Clarity:** 3
**Significance:** 3
**Originality:** 3
**Rating:** 5
**Confidence:** 5

**Summary:**

This paper presents a theoretical framework for evaluating the limitations of disaggregated performance metrics in fairness analysis. Using causal graphical models, particularly DAGs, the authors characterize when performance disparities across subgroups arise from structural data differences versus from model bias. They formalize a taxonomy of distribution shift types (e.g., covariate, outcome, label, presentation), derive conditional independence properties under each, and propose a controlled evaluation procedure using importance weighting to isolate confounding.
The theoretical contributions are clear: even Bayes-optimal models can yield group-wise disparities under plausible data-generating assumptions, and common fairness metrics may be misleading without consideration of underlying causal structure. Synthetic and real-world (ACS PUMS) experiments empirically support these findings.

**Questions:**

- Given that real-world practitioners typically cannot observe or validate the causal structure, what practical guidance can you offer for choosing among the causal assumptions in your taxonomy? Are there diagnostics or heuristics to help evaluate which DAG is plausible?
- Is there a path forward for extending this framework to multi-class or continuous outcomes? If so, what new assumptions would be required, and what challenges arise?
- How sensitive is the framework (particularly the weighting-based controlled evaluation) to models that are miscalibrated or underfit? Can misleading conclusions arise in such settings?
- What are the empirical characteristics of the importance weighting method under data sparsity or high dimensionality? How does the variance of the estimator behave for small subgroups?
-  For teams working with real-world data, what minimal conditions must be met (in terms of data quality, model fidelity, or covariate richness) to apply this framework responsibly?

**Ethical Concerns:**

["NO or VERY MINOR ethics concerns only"]

**Limitations:**

The paper touches on some of these issues but could do more to caution against misapplication. Controlled eval procedures can create false confidence if the model is poorly fit or the causal assumptions are incorrect. Practitioners without causal expertise could be misled by surface-level stability.

**Quality:**

3

**Strengths And Weaknesses:**

First, let's start with the strengths:
The causal analysis is rigorous, and the paper provides a clear set of sufficient conditions under which group-level disparities do or do not reflect bias. I'm also a fan of the use of DAGs to formalize distribution shift types (e.g., covariate vs. outcome shift) to add clarity to fairness evaluation debates and help unify previously fragmented observations in the literature. The visualization is quite strong. The tables summarizing which fairness criteria are expected to hold under different assumptions offer a usable diagnostic framework for readers with sufficient causal knowledge. Lastly, the focus on Bayes-optimality enables tractable and interpretable theoretical results.

Now on to the weaknesses in no particular order:
1. The framework presumes knowledge of the data-generating process or, at a minimum, a defensible causal DAG. In most real-world settings, this is a strong and often unjustifiable assumption. The practical relevance of the framework is directly tied to the feasibility of accurate causal discovery.

2. All theoretical and empirical analysis is restricted to binary classification. Extending to multi-class or continuous outcomes would likely require fundamentally different assumptions and analytical tools- not just technical extensions.

3. While Bayes-optimality enables clean theory, it is an unrealistic assumption in most deployment contexts. In its current form, the framework does not account for model capacity limitations, data scarcity, or learning instability.

4. The empirical evaluation is narrow with only two tabular prediction tasks and does not explore how the framework performs when causal assumptions are violated. Even within those tasks, the framework is not fully stress-tested under common real-world uncertainties.

---

> ### Author Rebuttal · Authors · 2025-07-30
>
> We thank reviewer oWko for their engaged and enthusiastic review. In light of their comments and questions, we would like to emphasize the following.
>
> **Regarding the need for causal knowledge of the data-generating-process**:
>
> We clarify that the use of our framework does not necessarily require an analyst to have full knowledge of the data generating process.
>
> In general our core results are of the form, “if a given causal structure and set of conditions hold, then the property of interest (typically conditional independence) holds”. This type of result immediately implies a diagnostic tool that serves as a form of causal discovery: if we observe evidence against the property of interest then we have evidence towards falsification of the causal structure and set of conditions that imply the property of interest. This follows from the equivalence of a logical statement “If P, then Q” with the contrapositive “If not Q, then not P”. Note that this does not provide the means to confirm any particular hypothesis regarding the underlying causal structure, but it does allow for rejection of a causal graph when we have evidence of violation of a conditional independence property that the causal graph in question satisfies.
>
> A key example of this in our work is reasoning about evidence against the hypothesis that the data were generated under the covariate shift in a setting without selection (which encodes the assumption $Y \perp A \mid X$). We show that both (1) evaluation of a subgroup-agnostic predictor with control for the covariates $X$ and (2) an analysis of whether subgroup-aware prediction improves over subgroup-agnostic prediction can be used as evidence to falsify the assumption that the data were generated under the covariate shift graph. In either case, we have evidence that the covariate shift assumption is violated but not confirmation that the data were generated from any of the other graphs considered. Fortunately, the other graphs that violate the covariate shift assumption (label shift, outcome shift, presentation shift, complex shift) behave similarly with respect to the properties that we discuss related to disaggregated evaluation and the informativeness of subgroup membership. The label shift graph is an exception, but we are again fortunate to have the ability to falsify the assumptions of the label shift setting with observations on the basis of evidence against the conditional independence $X \perp A \mid Y$ (which corresponds to a controlled evaluation of a subgroup-agnostic model that controls for $Y$).
>
> We also would like to emphasize that one rhetorical purpose of describing the properties of models fit to data faithful to this set of specific causal graphs is to serve as a reasonably comprehensive set of vignettes demonstrating how subgroup performance instability arises under common causal assumptions regarding heterogeneity across subgroups. The experiments with simulated data serve to verify that the properties of interest hold as expected. The experiments with real-world data serve to demonstrate how the framework can enhance fairness analyses in practical settings where the causal graph is not necessarily known and we cannot assume the model is Bayes-optimal.
>
> A major caveat to the above arguments is that in the case that selection is present, it is necessary to at least have some knowledge of the data generating process for valid inference. This is not a limitation unique to this study, but rather a fundamental inability to perform inference in the absence of assumptions regarding the relationship between the data and the target population or parameter or interest.
>
> **Regarding multi-class or continuous outcomes**:
>
> We agree that extending the approach to handle multi-class or continuous outcomes may require fundamentally different assumptions and analytical tools. We will update the discussion section to include this as an important area of future research.
>
> To briefly speculate, we expect that the multi-class classification context is a more straightforward translation from binary classification than the continuous outcome case is. For multi-class classification, the Bayes-optimal multi-class probabilistic predictor fully characterizes the categorical distribution $P(Y | X)$ in the same way that the Bayes-optimal binary probabilistic classifier fully characterizes Bernoulli $P(Y | X)$. As a result, we suspect many of the core definitions and findings to extend to the multi-class setting.
>
> In the continuous case, the conditional mean estimator $\mathbb{E}[Y \mid X]$ is not sufficient to describe arbitrary $P(Y \mid X)$. As a result, the Bayes-optimal conditional mean regressor does not immediately satisfy the same set of conditional independences satisfied in classification contexts (e.g., sufficiency). However, if we make  assumptions on the data generating process (e.g., if $Y$ is conditionally Gaussian or Poisson), the parameters fully characterizing those distributions can be estimated via maximum likelihood. The implications for disaggregated evaluations may also differ under different noise structures. For example, if the true noise model is homoskedastic (across levels of $X$), the Bayes-optimal conditional mean regressor attains equal subgroup mean squared error under the covariate shift graph, even if this is not true for binary classification or under heteroskedasticity.
>
> **Regarding departures from Bayes-optimality**:
>
> We would like to clarify that while some of the properties that we describe are specific to the Bayes-optimal case, our framework can still be informative under departures from Bayes-optimality. Please refer to the response to reviewer c4Qe for more details. Please also refer to the response to reviewer Lqbf for further discussion of potential directions for further theoretical and empirical study of the role of model misspecification and estimation error.
>
> Regarding weighting with data sparsity and high dimensionality:
> We concur that further work would be required to comprehensively extend our evaluation framework to high dimensional and sparse datasets. Both high-dimensionality and sparsity affect our approach by plausibly violating overlap (see e.g., D’Amour 2021), such that one or more regions of the covariate space are uniquely associated with exactly one subgroup. In the present version of the work, we do conduct a brief investigation of the issues that arise in the setting in section 3.5 titled “Subgroup separability” and the corresponding set of simulation experiments for the setting titled “Separable” (e.g., in Figure 1). The primary implication for our approach is the inability to detect covariate shift violation in the separable setting because it is not possible to compare $Y \mid X$ across subgroups for levels of $X$ that are only observed for a single subgroup. This is a fundamental issue and not a unique limitation of our approach.
>
> In the causal inference literature, overlap violations often yield weighting estimators with high variance, typically due to the presence of weights with small denominators. Intuitively, this occurs when a weighting estimator projects a distribution onto a region of low probability. We mitigate this issue by projecting the aggregate population distribution onto the distribution of each subgroup. Common support is guaranteed in this context because a subgroup is a subset of the aggregate population.
>
> Our approach yields weights proportional to $P(A \mid V)$, which is bounded between 0 and 1, and does not induce the same sort of high variance estimates. The limitation of this approach, however, is that under full separability (i.e., $P(A \mid V)$ = 0 or 1 for all $V$), the weighted population estimate is identical to the unweighted subgroup estimate regardless of the causal structure, implying no ability to compare the subgroups.
>
> In the present paper, we discuss this in supplementary methods section A.1 and do observe some examples of high variance estimation when applying the alternative weighting approach of Cai et al (Supplementary Figures B18-B24).
>
> D’Amour, Alexander, et al. "Overlap in observational studies with high-dimensional covariates." Journal of Econometrics 221.2 (2021): 644-654.
>
> **Regarding minimal conditions for use with real-world data**:
>
> For the setting that the majority of this work is conducted, it is necessary to assume that the data used is high-quality and representative of the relevant target population and that the prediction task is well-formulated, such that increases in predictive performance are consistent with increased utility of the model. Arguably, these assumptions underpin most applied uses of machine learning, regardless of whether they are made explicit or are appropriate. One strength of our approach is that we provide a structured way to reason about deviations from this ideal through explicit specification of the causal structure of selection mechanisms.
> * Regarding data quality and richness, similar to our approach taken to reasoning about selection, it is plausible that the causal graphs we present could be augmented to explicitly represent assumptions regarding measurement mechanisms that consider the observed data to be noisy or incomplete views of unobserved constructs of interest. This would allow characterization of model performance and fairness properties under various assumptions regarding those mechanisms and may suggest approaches to testing. We can update the discussion section to mention this as future directions.
> * Regarding model fidelity, please refer to the earlier points about deviations from Bayes-optimality.
> * We intend to update the discussion section to emphasize key practical takeaways, including recommendations for the interpretation of standard disaggregated evaluation, implications of the theoretical analysis and empirical findings, as well as guidance for appropriate use and interpretation of the methodology presented here.

---

> > ### Comment · Reviewer_oWko · 2025-08-09
> >
> > Thank you for your thoughtful rebuttal. Below are my observations on your responses:
> >
> > Causal Knowledge and Data-Generating Process:
> > The clarification that your framework does not require full knowledge of the data-generating process is helpful. However, the reliance on causal assumptions for diagnostic purposes still requires careful consideration, especially when analysts may struggle to accurately identify the causal structure in complex scenarios. More specific guidance on how to validate or assess plausible causal assumptions would enhance the practical use of your framework in real-world settings, where these assumptions are often difficult to confirm.
> >
> > Extension to Multi-Class or Continuous Outcomes:
> > The discussion on extending the framework to multi-class or continuous outcomes is valuable, but as you noted, the assumptions and tools required would differ significantly from binary classification. The challenges posed by continuous outcomes, particularly in the case of the conditional mean estimator, are non-trivial. Further elaboration on how your approach would scale in these settings, including any changes to assumptions or methods, would clarify its broader applicability.
> >
> > Departures from Bayes-Optimality:
> > Your response about the framework's applicability even with departures from Bayes-optimality is reassuring. Still, the limitations you acknowledge, such as model misspecification, are crucial. A more thorough discussion of how the framework can adapt to these issues would be useful. Specifically, examples or strategies for handling less-than-ideal models in practice would strengthen the robustness of your approach.
> >
> > Sparsity and High-Dimensionality Challenges:
> > I appreciate the discussion on sparsity and high-dimensionality, particularly regarding the limitations in overlap violations. The method of projecting distributions onto subgroups is a good starting point, but high variance in sparse settings remains a critical concern. Expanding on the implications of this high variance, especially how it might affect real-world applications with large, complex datasets, would improve understanding of the method’s practical constraints.
> >
> > Minimal Conditions for Use with Real-World Data:
> > While you emphasize the need for high-quality, representative data, the definition of "high-quality" is still somewhat vague. In particular, more clarity on the type of causal assumptions needed for practical applications would help. Additionally, offering guidance on how to handle noisy or incomplete data, especially in the context of causal structure, would make the framework more robust and applicable to real-world scenarios.

---

### Note · Authors · 2025-08-13

We thank the reviewers for their support for the work and constructive feedback.

For this response, we highlight the recurring themes that arose during the review and rebuttal process:

**The role of Bayes-optimality and the applicability of our findings in practical settings**

We clarified during the review process that while some of the properties that we describe are specific to the Bayes-optimal case, our framework can still be informative under departures from Bayes-optimality. In particular, we clarified that the use of controlled evaluation procedures does *not* require Bayes-optimality, even if interpretation can differ depending on whether the model is Bayes-optimal. To make this clearer in the manuscript, we proposed textual changes to sections 3.3 and 3.4 and drafted a new table that details the set of implications and valid conclusions that can be drawn from controlled evaluations based on the configuration of the weights, the covariate set, the observed statistic T_a, and whether the model is Bayes-optimal. Please see the response to reviewer c4Qe for more details.

Several reviewers raised the point that further empirical and theoretical characterization of the role of model misspecification and estimation error would enhance the work. In the camera-ready version, we intend to discuss promising directions for future research that extends this work in these directions.

**Recommendations for practitioners**

The reviewers raised the point that practical recommendations could be presented more clearly. In the camera-ready version, we intend to revise section 3.4 and the discussion section to include clearer guidance for the design and interpretation of both standard and controlled disaggregated evaluation under different conditions without knowledge of the causal graph. We believe that the new table mentioned above in response to reviewer c4Qe helps to address this point.

**Other improvements in clarity**

The reviewers offered several helpful suggestions for how to improve the clarity of the manuscript. We detail our proposed changes in the response to reviewer Lqbf. These include further exposition and conceptual scaffolding throughout the manuscript, minor textual revision to break up long sentences, an algorithm block, clarification of the relationship between seemingly disparate sets of results (subgroup-awareness vs. metric stability), clearer recommendations for practitioners, and clearer communication of limitations and scope.

---

### Decision · Program_Chairs · 2025-09-17

**Decision:**

Accept (poster)

**Comment:**

This paper uses causal models to explore how various settings (such as different types of distribution shift) impact classifiers, and in particular whether Bayes-optimality remains a sufficient condition for the fairness criteria of separation and sufficiency.

Reviewers agreed the paper is well-motivated. They had some concerns about whether it provides practical recommendations, and the authors answered their concerns satisfactorily with proposed revisions.

Reviewers questioned whether focusing on Bayes-optimal classifiers was too narrow a focus and authors agreed to give some additional guidance for the non-optimal case. The paper would also be strengthened by considering criteria other than sufficiency and separation. For example, since causal models have already been introduced, it might not require much additional work to explore a causal fairness definition. This is not a requirement, only an example suggestion. But the current focus on sufficiency and separation is fairly narrow. The fairness literature contains other prominent criteria which are known to be (generally) incompatible with sufficiency/separation, and including any of these would broaden the potential audience for this paper.

The caption for Figure 1 mentions confidence intervals but I only see points. This must be corrected in the revision.